# Server-Proximal Aggregation for Federated Domain-Incremental Learning under Partial Participation: Task-Uniform Convergence and Backward Transfer

Longtao Xu [1]    Jian Li [1]

## Abstract

Real-world federated systems seldom operate on static data: input distributions drift while privacy rules forbid raw data sharing. We study Federated Domain-Incremental Learning (FDIL), where (i) clients are heterogeneous, (ii) tasks arrive sequentially with shifting domains, and (iii) the label space remains fixed. Two theoretical pillars remain missing for FDIL under partial participation: a guarantee of backward knowledge transfer (BKT) and a convergence rate that holds *uniformly across the task sequence*. We introduce SPECIAL (Server-Proximal Efficient Continual Aggregation for Learning), a simple, replay-free FDIL algorithm that adds a single server-side "anchor" to FedAvg: in each round, the server aggregates updates from a uniformly sampled subset of clients and then blends the result with the previous global model via a lightweight proximal step. This anchor curbs cumulative drift without replay buffers, synthetic data, or task-specific heads, leaving communication cost and model size unchanged. Our theory shows that SPECIAL (i) *preserves earlier tasks*: a BKT bound caps any increase in earlier-task loss by a drift-controlled term that shrinks with more rounds, local epochs, and participating clients; and (ii) *achieves task-uniform, communication-efficient convergence* for non-convex FDIL with partial participation: $\mathcal{O}\big(\sqrt{E/(NT)}\big)$ in expected gradient norm, with $E$ local epochs, $T$ rounds, and $N$ participating clients, while explicitly separating optimization variance from inter-task drift. Experiments on standard FDIL benchmarks corroborate the theory.

[1]Stony Brook University, New York. Correspondence to: Longtao Xu <longtao.xu@stonybrook.edu>, Jian Li <jian.li.3@stonybrook.edu>.

*Proceedings of the 43rd International Conference on Machine Learning*, Seoul, South Korea. PMLR 306, 2026. Copyright 2026 by the author(s).

## 1. Introduction

Modern learning systems face *temporal non-stationarity*: data arrive as a sequence of tasks whose distributions evolve over time. Continual learning (CL), also called incremental learning (IL), therefore develops algorithms that acquire new task knowledge efficiently, retain earlier performance to mitigate catastrophic forgetting, and exploit forward and backward transfer (Lopez-Paz & Ranzato, 2017; Chen & Liu, 2018). In parallel, Federated Learning (FL) enables collaborative training across clients that hold heterogeneous data without sharing raw samples (McMahan et al., 2017). FL introduces its own difficulties such as client heterogeneity, limited communication, and privacy constraints. When the *temporal* shifts of CL meet the *spatial* heterogeneity of FL, naive combinations fail: centralized CL violates privacy, while standard FL presumes stationarity and offers no retention guarantees. Federated Continual Learning (FCL) (Yoon et al., 2021) emerges as the principled fusion addressing *both* kinds of non-stationarity.

Most FCL studies implicitly target class-incremental scenarios (FCIL) in which new classes appear. Yet many real deployments exhibit *domain shifts under a fixed label set*, namely the federated domain-incremental learning (FDIL) regime. For example, a hospital consortium refines a diagnostic model while scanners, protocols, or demographics drift; labels stay fixed and raw images remain private. FDIL is therefore not a niche corner case but a common practical regime that demands methods tailored to domain drift *and* federated privacy constraints.

Despite a flurry of algorithms such as memory replay (Rebuffi et al., 2017; Hou et al., 2019; Wu et al., 2019), regularization/proximal (Zenke et al., 2017; Li & Hoiem, 2017a), parameter isolation/expansion (Javed & White, 2019), and meta-learning (Finn et al., 2017; Riemer et al., 2019), *two theoretical gaps persist in FDIL under simultaneous client heterogeneity and temporal domain shift*: (i) a *backward knowledge transfer* (BKT) guarantee that quantifies when training on later tasks improves or at least preserves earlier tasks without storing past data; and (ii) a *task-uniform* convergence rate that holds across the entire sequence of tasks rather than only the last one, while accounting for stochastic noise, intra- and inter-client variance, and cumulative

domain drift. These gaps are amplified under *partial participation*, the prevailing deployment mode in which only a subset of clients is selected each round. To close these theoretical gaps, the primary question we seek to address is

*Can we design a simple FDIL framework that handles spatial heterogeneity, temporal non-stationarity, and partial participation, yet guarantees BKT and an all-task convergence rate?*

We answer this question with SPECIAL (*Server-Proximal Efficient Continual Aggregation for Learning*), a lightweight, replay-free framework for FDIL with partial participation. In each communication round, the server uniformly samples $N$ of $M$ clients (without replacement), aggregates only their updates, and then applies a single *server-side proximal anchor* that blends the aggregated model with the previous global model via a small quadratic step. This one-line change to FedAvg curbs cumulative parameter drift, letting the model adapt to new domains while implicitly preserving earlier knowledge, *without* replay buffers, synthetic data, or task-specific heads, so both communication cost and model size remain unchanged. Unlike replay- or expansion-based FDIL methods (Li et al., 2024c; Qi et al., 2023), SPECIAL keeps the footprint constant; unlike client-side regularizers (Li et al., 2024b), the proximal interaction occurs *only at the server*, and our analysis targets continual, *task-uniform* guarantees. We deliver two key theoretical contributions:

● **Backward knowledge transfer bound under partial participation.** After partially or fully training on a new task, the loss on any earlier task is provably no greater than its previous value plus a drift-controlled term that *shrinks with more rounds, more local epochs, and larger per-round participation $N$*. The bound explicitly captures stochastic variance and client heterogeneity.

● **First task-uniform convergence rate for FDIL with partial participation.** We establish, to our knowledge, the **first** task-uniform non-convex convergence guarantee for FDIL under partial participation. With suitable local and global learning rates, the expected gradient norm of SPECIAL decays at $\mathcal{O}(\sqrt{E/(NT)})$, where $E$ is the number of local epochs, $N$ the number of participating clients per round, and $T$ the number of rounds. The rate matches the communication efficiency of FedAvg on a single stationary task when expressed per participating update, holds simultaneously across all tasks, is independent of the total client pool size $M$, and includes an explicit inter-task drift term that quantifies how domain shift accumulates and how it can be suppressed by tuning the proximal weight and the round/epoch trade-off.

Together, these results connect continual-learning retention theory with federated optimization under realistic sampling,

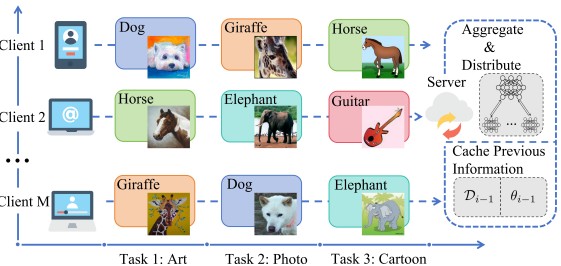

*Figure 1.* Illustration of FDIL on the PACS dataset.

demonstrating that a single server-side proximal anchor can simultaneously curb cumulative drift and enable *provably efficient, task-uniform* learning over long sequences of evolving domains with partial participation.

## 2. Problem Formulation

### 2.1. Federated Domain-Incremental Learning

In the CL setting, a model observes a sequence of tasks $\mathcal{T} = \{\mathcal{T}_1, \ldots, \mathcal{T}_K\}$. Task $\mathcal{T}_i = \{(x_n^i, y_n^i)\}_{n=1}^{N_i}$ contains $N_i$ data samples with inputs $x_n^i \in \mathcal{X}^i$ and labels $y_n^i \in \mathcal{Y}^i$. In domain-incremental learning (DIL), all tasks share the same label space ($\mathcal{Y}^i = \mathcal{Y}$ for every task $i$), while their input domains differ ($\mathcal{X}^i \neq \mathcal{X}^j$ for some $i \neq j$). We extend DIL to a federated setting with a central server and $M$ clients. Client $m$ can access only its own local data for each task. At task $K$, the goal of FDIL is to find parameters $\theta_K \in \mathbb{R}^d$ that minimize the cumulative loss across all $K$ tasks:

$$\min_{\theta_K \in \mathbb{R}^d} f_{1:K}(\theta_K) = \sum_{i=1}^K f_i(\theta_K) = \frac{1}{M} \sum_{i=1}^K \sum_{m=1}^M f_{i,m}(\theta_K),$$
(1)

where $f_{i,m}(\theta) = \mathbb{E}_{(\boldsymbol{x},y) \sim \mathcal{T}_{i,m}}[f_{i,m}(\theta; \boldsymbol{x}, y)]$ is the possibly non-convex loss of client $m$ on task $i$. Figure 1 illustrates the FDIL workflow. Eq. (1) can be decomposed into the loss $f_K$ on the current task $K$ and the cumulative loss $f_{1:K-1}$ on previous tasks:

$$f_{1:K}(\theta_K) = \underbrace{f_K(\theta_K)}_{\text{current}} + \underbrace{f_{1:K-1}(\theta_K)}_{\text{previous}}.$$
(2)

Because privacy regulations and storage limits prevent the server or clients from keeping raw data from earlier tasks, $f_{1:K-1}(\theta_K)$ cannot be evaluated exactly. It can, however, be approximated without full access to past data samples. Experience-replay approaches retain a small, representative subset of earlier data to stand in for the missing objective. In our setting, *we instead treat the previous global model $\theta_{K-1}$, the parameter vector obtained after finishing task $K-1$, as a compact summary of prior knowledge.* Although it stores no raw samples, $\theta_{K-1}$ *encodes enough information to regularize learning on the new task and protect performance on earlier ones.*

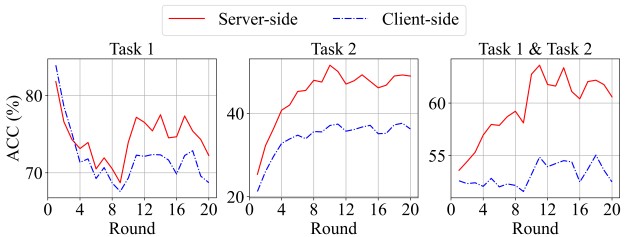

*Figure 2.* Test accuracy on Task 1, Task 2, and their union for Digit-10 under client-side vs. server-side proximal terms.

At the start of task $K$, we initialize the model with the parameters learned on task $K-1$, that is, $\theta_K^0 = \theta_{K-1}$. Inspired by regularization-based continual-learning methods, we add a proximal term to the objective to discourage large departures from $\theta_{K-1}$. This lightweight anchor curbs catastrophic forgetting by keeping optimization centered on the previous solution while still allowing the model to absorb new information from task $K$. Consequently, we optimize the following regularized surrogate for Eq. (1):

$$\theta_K = \arg\min_{\theta \in \mathbb{R}^d} \Big\{ f_{1:K}(\theta) = \underbrace{f_K(\theta)}_{\text{current}} + \underbrace{\lambda \|\theta - \theta_{K-1}\|^2}_{\text{previous}} \Big\}.$$

$$(3)$$

### 2.2. Client- vs. Server-Side Proximal Terms

Most regularization-based approaches keep prior knowledge in play during local training (Li et al., 2020b; 2024b; Zhang et al., 2023). In the training process on task $i, i \geq 2$, each client $m$ solves

$$\theta_{i,m} = \arg\min_{\theta \in \mathbb{R}^d} \Big\{ f_{i,m}(\theta) + \lambda \|\theta - \theta_{i-1}\|^2 \Big\}, \quad (4)$$

so every local update is pulled towards the previous global model $\theta_{i-1}$ in each local epoch. While this continual "review" helps preserve earlier knowledge, it can also hinder adaptation to the new task, known as the classic stability-plasticity dilemma (Parisi et al., 2019; Grossberg, 2013).

We illustrate the effect on the first two tasks of Digit-10 (see details in Appendix F). As Figure 2 shows, the client-side proximal term protects task 1 accuracy during the initial rounds of task 2 training but learns task 2 more slowly. In contrast, a server-side proximal term lets clients update freely and applies the anchor only at aggregation, striking a better balance: it reaches higher accuracy on the new task without extra forgetting and even exhibits backward transfers, with task 1 accuracy increasing after 10 rounds on task 2, yielding the best overall performance. This observation motivates our server-side regularization design, which reconciles stability and plasticity more effectively than client-side regularizers (see Section 5 for numerical evaluations).

---

**Algorithm 1** The SPECIAL Algorithm

---

1: Initialize $\theta_0$
2: **for** task $i = 1$ to $K$ **do**
3:     Set task initial point: $\theta_i^0 = \theta_{i-1}$
4:     **for** round $t = 0$ to $T - 1$ **do**
5:         The server randomly selects a set $\mathcal{S}_i^t$ of $N$ clients
6:         **for** client $m \in \mathcal{S}_i^t$ **in parallel do**
7:             Download from server: $\theta_{i,m}^{t,0} = \theta_i^t$
8:             **for** epoch $e = 0$ to $E - 1$ **do**
9:                 Calculate: $g_{i,m}^{t,e} = \nabla f_{i,m}\big(\theta_{i,m}^{t,e}, \xi_{i,m}^{t,e}\big)$
10:                 Local update: $\theta_{i,m}^{t,e+1} = \theta_{i,m}^{t,e} - \gamma_L g_{i,m}^{t,e}$
11:             **end for**
12:             $\Delta_{i,m}^t = \theta_{i,m}^{t,E} - \theta_{i,m}^{t,0} = -\gamma_L \sum_{e=0}^{E-1} g_{i,m}^{t,e}$
13:             Send $\Delta_{i,m}^t$ to server
14:         **end for**
15:         Server receives all $\Delta_{i,m}^t$
16:         Compute: $\Delta_i^t = \frac{1}{N} \sum_{m \in \mathcal{S}_i^t} \Delta_{i,m}^t$
17:         Intermediate update: $\bar{\theta}_i^{t+1} = \theta_i^t + \gamma_G \Delta_i^t$
18:         Server update according to Eq. (6)
19:     **end for**
20:     Set final model of task $i$: $\theta_i = \theta_i^T$
21: **end for**

---

## 3. The SPECIAL Algorithm

SPECIAL augments FedAvg with a *server-side proximal anchor* that links each round's update to the global model obtained at the end of the *previous* task. The anchor functions as a compact memory: it curbs catastrophic drift without changing the client workflow or enlarging communication.

**Federated skeleton.** At the start of each communication round for task $i$, the server broadcasts the current global model $\theta_i^t$ to a subset of $N$ out of $M$ clients. We denote the selected index set by $\mathcal{S}_i^t, i \in [K], t \in [T]$. Each selected client downloads the model and performs $E$ epochs of stochastic-gradient descent (SGD) on *only* its own data from task $i$. At the $e$-th epoch, we assume the gradient of client $m$ estimated on the local data sample $\xi_{i,m}^{t,e}$ is an unbiased estimator, which is denoted as $g_{i,m}^{t,e} = \nabla f_{i,m}\big(\theta_{i,m}^{t,e}, \xi_{i,m}^{t,e}\big)$. After local optimization, the client transmits the update vector $\Delta_{i,m}^t = \theta_{i,m}^{t,E} - \theta_i^t$ rather than the full model back to the server. This vector summarizes all information the client has extracted from its previous data during the round.

After collecting the difference vectors $\Delta_{i,m}^t$ from all participating clients, the server computes their average, exactly the aggregation step used by FedAvg. Applying this average, scaled by a global learning rate $\gamma_G$, yields an intermediate model $\bar{\theta}_i^{t+1} = \theta_i^t + \gamma_G \Delta_i^t$ that reflects what the current task has taught the federation so far, and $\bar{\theta}_i^{t+1}$ is the aggregation result for round $t$ when the current task is the first task of the sequence. Otherwise, the server blends $\bar{\theta}_i^{t+1}$ with the

final model from the previous task, $\theta_{i-1}$, via a lightweight proximal step before the round ends (detailed below). The resulting model $\theta_i^{t+1}$ is broadcast at the start of next round.

**Server-side proximal anchor.** Because clients optimize only the current-task loss, naively applying FedAvg may rapidly forget features useful to earlier tasks. To counter this, the server solves the quadratic problem

$$\theta_i^{t+1} = \underset{u \in \mathbb{R}^d}{\arg\min} \left\{ \left\| u - \bar{\theta}_i^{t+1} \right\|^2 + \lambda \left\| u - \theta_{i-1} \right\|^2 \right\}, \quad (5)$$

whose closed-form solution is

$$\theta_i^{t+1} = \frac{1}{1+\lambda} \bar{\theta}_i^{t+1} + \frac{\lambda}{1+\lambda} \theta_{i-1}. \quad (6)$$

Here, $\bar{\theta}_i^{t+1}$ captures information from the current task, whereas $\theta_{i-1}$ embodies all knowledge accumulated thus far. Their weighted average preserves earlier skills while still adapting to the new domain. The coefficient $\lambda > 0$ is SPECIAL 's *only* new hyperparameter and governs the stability–plasticity trade-off: $\lambda = 0$ reduces to plain FedAvg, whereas larger $\lambda$ assigns more weight to retention. The complete procedure is summarized in Algorithm 1.

## 4. Theoretical Analysis

We first state technical assumptions used throughout. We then present two main results: (i) a BKT guarantee showing that training on a new task improves or at least preserves the loss on each earlier task, and (ii) a task-uniform, communication-efficient convergence rate for SPECIAL that holds for non-convex objectives and non-IID data under partial participation. Full proofs appear in Appendix E.

### 4.1. Assumption

**Assumption 4.1** (Bounded Gradients). For every task $i \in [K]$ and client $m \in [M]$, the stochastic gradient is uniformly bounded, i.e., $\|\nabla f_{i,m}(\theta, \xi)\| \leq B$, for $\theta \in \mathbb{R}^d$, $\xi \in \mathcal{T}_{i,m}$, and a constant $B$.

**Assumption 4.2** ($L$-Smooth). For every task $i \in [K]$, client $m \in [M]$, and all parameter vectors $\theta, \theta' \in \mathbb{R}^d$, the local objective $f_{i,m}(\theta)$ is $L$-Smooth, i.e., $\|\nabla f_{i,m}(\theta) - \nabla f_{i,m}(\theta')\| \leq L \|\theta - \theta'\|$.

**Assumption 4.3** (Unbiased Local Gradient estimator). For every task $i \in [K]$, client $m \in [M]$, and communication round $t \in [T]$, the local stochastic gradient computed on a random sample $\xi_{i,m}^t$ is an unbiased estimate of the true local gradient, i.e., $\mathbb{E}\left[\nabla f_{i,m}\left(\theta_{i,m}^t, \xi_{i,m}^t\right)\right] = \nabla f_{i,m}\left(\theta_{i,m}^t\right)$.

**Assumption 4.4** (Bounded Local Gradient Difference). There exists a constant $\sigma_L > 0$, such that for every task $i \in [K]$, client $m \in [M]$, and round $t \in [T]$, the variance of stochastic gradients in client $m$ at task $i$ is bounded by $\mathbb{E}\left\|\nabla f_{i,m}\left(\theta_{i,m}^t, \xi_{i,m}^t\right) - \nabla f_{i,m}\left(\theta_{i,m}^t\right)\right\|^2 \leq \sigma_L^2$.

**Assumption 4.5** (Bounded Intra-task Gradient Difference). There exists a constant $\sigma_G > 0$, such that for every task $i \in [K]$, client $m \in [M]$, and round $t \in [T]$, the global variability of local gradient from each client's loss function can be bounded by $\|\nabla f_{i,m}(\theta_i^t) - \nabla f_i(\theta_i^t)\|^2 \leq \sigma_G^2$.

**Assumption 4.6** (Bounded Inter-task Gradient Difference). There exists a constant $\sigma_T > 0$, such that for every pair of tasks $i, j \in [K]$ and every parameter vector $\theta \in \mathbb{R}^d$, the difference of their global gradients can be bounded by $\|\nabla f_i(\theta) - \nabla f_j(\theta)\|^2 \leq \sigma_T^2$.

Assumption 4.1 is routinely used to control magnitude of stochastic gradients in distributed optimization (Lin et al., 2022; Li et al., 2020c; Reddi et al., 2021). Assumption 4.2 is standard for non-convex analysis. Assumptions 4.3 and 4.4 bound, respectively, the sampling bias and variance introduced by per-client minibatches. Assumption 4.5 is the usual way to quantify *client heterogeneity* in FL. For inter-task drift we additionally assume a bounded gradient discrepancy across tasks (Assumption 4.6); this *does not* require tasks to be similar. The constant plays the role of a variation/drift budget common in non-stationary optimization and continual learning: our bounds degrade monotonically with the task-drift level, recovering the stationary single-task setting when the drift is zero. This mirrors "positive correlation" or path-length style conditions used to formalize task relatedness in CL (Lin et al., 2022). A formal connection is given in Corollary D.1 in Appendix D.

### 4.2. Backward Knowledge Transfer Analysis

We first analyze backward knowledge transfer, namely the phenomenon that training on a new task can *improve* performance on earlier tasks (Benavides-Prado & Riddle, 2022).

**Theorem 4.7** (BKT under partial participation). *Let $\theta_K^t$ denote the global model after round $t$ of task $K$, and set $\theta_K^0 = \theta_{K-1}$. In each round $\tau$, the server samples a subset $\mathcal{S}_K^\tau \subseteq [M]$ of size $N$ uniformly without replacement and aggregates only those clients. Suppose for every $\tau \in [t]$, local epoch $e \in [E]$, and client $m \in [M]$, the gradients are $\epsilon$-aligned with earlier-task gradient at task-$K$ start, i.e., $\left\langle \nabla f_{1:K-1}\left(\theta_K^0\right), \nabla f_{K,m}\left(\theta_{K,m}^{\tau,e}\right) \right\rangle \geq \epsilon \left\| \nabla f_{1:K-1}\left(\theta_K^0\right) \right\| \cdot \left\| \nabla f_{K,m}\left(\theta_{K,m}^{\tau,e}\right) \right\|, \epsilon \in (0,1)$, and local and global learning rates satisfy $\gamma_L \leq \frac{2\epsilon \|\nabla f_{1:K-1}(\theta_K^0)\|}{BLEt\sqrt{ME \cdot \left(\frac{1}{\lambda^2+2\lambda}\right)}}$ and $\gamma_G \leq \frac{1}{\sqrt{N}(K-1)}$. Then, for every $t \geq 1$, the server-updated model $\theta_K^t$ generated by SPECIAL satisfies:*

$$\mathbb{E}_t\left[f_{1:K-1}\left(\theta_K^t\right)\right] \leq f_{1:K-1}\left(\theta_K^0\right)$$
$$+ \frac{2\epsilon^2 \sigma_L^2 \left\|\nabla f_{1:K-1}\left(\theta_K^0\right)\right\|^2}{(K-1)\, tEMNLB^2}, \quad (7)$$

*where the expectation is over client sampling and data stochasticity.*

**Implications.** The bound has two parts. The first term is the *initial sub-optimality* on the earlier tasks at the start of task $K$ (since $\theta_K^0 = \theta_{K-1}$). It reflects how well the model already performed on the earlier tasks before seeing any data from task $K$. The second term is a *vanishing correction* that decays with more rounds $t$, more local epochs $E$, and larger per-round participation $N$. Its magnitude scales with $\|\nabla f_{1:K-1}(\theta_K^0)\|^2$ (harder when the earlier-task gradient is large) and with the local variance $\sigma_L^2$, and it is modulated by the server-side proximal weight $\lambda$ via the step-size constraint. When $N = M$, we recover the full-participation case as a corollary (see Corollary D.2 in Appendix D); when $N$ decreases, the $1/N$ factor makes the decay slower, matching intuition under partial participation.

*Remark* 4.8 (**Positioning relative to prior BKT theory**). Lin et al. (2022) establish backward transfer in a *centralized* CL setting, showing that under a cosine-alignment condition, training on a new task can reduce earlier-task loss at any epoch. In practice, large distribution gaps at task boundaries often trigger an initial spike in the global loss, which obscures early-epoch improvements. Our bound in Eq. (7) makes this behavior explicit via the local-data variance term $\sigma_L^2$: it expresses the expected earlier-task loss at $\theta_K^t$ as the starting loss plus a *vanishing correction* that contracts with more communication rounds, more local epochs, and broader effective participation. This quantifies when and how fast loss recovery occurs after a sharp shift. The structure parallels Lemma 3.2 of Shi & Wang (2023), which separates retained loss from a prediction/shift error, but our analysis is *federated*, allows non-IID clients and round-by-round subsampling, and relies on a *server-side* proximal anchor rather than replay or architectural expansion. The alignment parameter $\epsilon$ in Eq. (7) plays the role of a cosine-similarity lower bound: better alignment between gradients of the new task and the aggregate of earlier tasks yields a faster decay of the correction term. The constants attached to $\sigma_L^2$ make the dependence on stochasticity and heterogeneity transparent. Taken together, Eq. (7) extends centralized BKT theory to FDIL under partial participation without replay buffers or task-growing memory, and complements our task-uniform convergence result by certifying *loss-space* retention while the model continues to optimize on the new domain.

### 4.3. Task-Uniform Convergence of SPECIAL

Most prior FCL analyses establish convergence only on the *final* task (Keshri et al., 2025; Han et al., 2023a), which is insufficient for FDIL: with heterogeneous tasks, converging on $f_K$ need not imply progress on $f_{1:K-1}$. We instead target the *global* objective $f_{1:K} \triangleq \sum_{i=1}^{K} f_i$ and

prove a task-uniform rate under partial participation. The proof has two ingredients: (i) Lemma 4.9 gives a *uniform* within-task drift bound $\|\theta_i^t - \theta_i^0\|^2 \leq \gamma_G^2 \gamma_L^2 E^2 B^2/\lambda^2$ that is independent of round $t$ and task $i$; and (ii) Theorem 4.10 converts this control into a convergence guarantee for $\min_{t \in [T]} \mathbb{E}\|\nabla f_{1:K}(\theta_K^t)\|^2$, yielding the communication-efficient rate $\mathcal{O}(\sqrt{E/(NT)})$ with explicit separation of stochastic noise, client heterogeneity, and inter-task drift $\sigma_T^2$. The result holds for non-convex, non-IID FDIL and is independent of the total client pool size $M$, highlighting the role of server-side proximal anchor and step-size choice in stabilizing progress across the entire task sequence.

Compared with a centralized setting, convergence in the federated setting is complicated by *client drift*: local updates amplify stochastic gradient noise and repeated rounds can inflate parameter variance. The difficulty is sharper in FDIL, where each task starts from a checkpoint that already encodes prior tasks. If the deviation $\|\theta_i^t - \theta_i^0\|^2$ grows unchecked, progress on the joint objective slows or can even stall. SPECIAL limits this growth via a server-side proximal anchor, leading to the following drift bound.

**Lemma 4.9** (Uniform within-task drift)**.** *Assume the bounded-gradient condition in Assumption 4.1. For every task $i \in [K]$ and round $t \in [T]$, the sequence $\{\theta_i^t\}$ produced by SPECIAL satisfies*

$$\mathbb{E}_t \left\| \theta_i^t - \theta_i^0 \right\|^2 \leq \frac{\gamma_G^2 \gamma_L^2 E^2 B^2}{\lambda^2}. \tag{8}$$

**Implications.** The squared distance from the task-initial point is governed jointly by the global and local learning rates ($\gamma_G, \gamma_L$), the number of local epochs ($E$), the gradient bound ($B$), and crucially the proximal weight $\lambda$. A larger $\lambda$ tightens the anchor to $\theta_i^0$ and suppresses drift. The bound is independent of the round $t$ and task $i$, implying a uniform deviation cap across all rounds and tasks.

We are now ready to state a convergence guarantee for the cumulative objective $f_{1:K}$ that holds uniformly across the entire task sequence.

**Theorem 4.10** (Task-uniform convergence of SPECIAL)**.** *Let the local learning rate $\gamma_L$ and global learning rate $\gamma_G$ satisfy $\gamma_G \leq \frac{1}{K-1}$, $\gamma_L \leq \frac{1}{8EL}$ and $\gamma_G \gamma_L \leq \frac{1+\lambda}{3EL}$. In each round, the server samples $N$ of $M$ clients uniformly without replacement and aggregates only their updates. Then the sequence $\{\theta_K^t\}$ generated by SPECIAL satisfies:*

$$\min_{t \in [T]} \mathbb{E} \left\| \nabla f_{1:K}\left(\theta_K^t\right) \right\|^2 \leq \frac{f_{1:K}^0 - f_{1:K}^*}{\frac{1-\frac{1}{K}}{2(1+\lambda)} E \gamma_G \gamma_L T} + \Psi, \tag{9}$$

*where $f_{1:K}^0 \triangleq f_{1:K}(\theta_K^0)$, $f_{1:K}^* \triangleq f_{1:K}(\theta_K^*)$, $\theta_K^*$ is an optimal point for task sequence $1:K$, the expectation is*

*over client sampling and data stochasticity, and*

$$
\begin{aligned}
\Psi =& \frac{2}{1-\frac{1}{K}} \left[ \frac{12\gamma_G\gamma_L (M-N) EKL\sigma_G^2}{(1+\lambda) N (M-1)} \right. \\
&+ \left( \frac{\gamma_L^2 E^2 L^2}{\lambda^2} + K + \frac{3\gamma_G\gamma_L (M-N) EKL}{(1+\lambda) N (M-1)} \right) B^2 \\
&+ \left( 5\gamma_L^2 KEL^2 + \frac{60\gamma_G\gamma_L^3 (M-N) E^2 KL^3}{(1+\lambda) N (M-1)} \right) \\
&\cdot \left( \sigma_L^2 + 6E\sigma_G^2 \right) + \frac{3\gamma_G\gamma_L L\sigma_L^2}{2N (1+\lambda)} + \left\| \nabla f_{1:K-1} \left( \theta_K^0 \right) \right\|^2 \\
&+ \left. \frac{(K-1)^2 E}{K} \cdot \frac{3\gamma_G\gamma_L L}{1+\lambda} \left( \frac{1}{2} + \frac{4 (M-N)}{N (M-1)} \right) \sigma_T^2 \right].
\end{aligned}
$$

**Implications.** The right-hand side of Eq. (9) consists of a *vanishing* term and a *constant* term $\Psi$. Under the step-size conditions of Theorem 4.10, the vanishing term scales like $\frac{f_{1:K}^0 - f_{1:K}^*}{c\left(1-\frac{1}{K}\right)ET}$, for a positive constant $c$. Thus the rate accelerates with more rounds $T$ and more local work $E$, but slows as $K$ grows, reflecting that longer histories require more communication to reach the same stationary level.

The residual $\Psi$ is scaled by $\frac{2}{1-\frac{1}{K}}$, which approaches 2 as $K$ increases, and separates five effects: (i) the within-task drift term from Lemma 4.9, which grows linearly with $K$; (ii) additional client-drift terms induced by partial participation, which vanish as $N$ approaches $M$; (iii) accumulated stochasticity $(\sigma_L^2, \sigma_G^2)$ across $K$ tasks, suggesting $\gamma_L = \mathcal{O}(1/(\sqrt{K}E))$ to keep this component small; (iv) an explicit task-heterogeneity penalty $\sigma_T^2$, amplified roughly by $\frac{(K-1)^2}{K}$, which motivates $\gamma_G = \mathcal{O}(1/(K-1))$; and (v) the initial gradient norm $\|\nabla f_{1:K-1}(\theta_K^0)\|^2$, which tightens when earlier tasks are already well fit, echoing Theorem 4.7. Setting $N = M$ recovers the full-participation case as a corollary (see Corollary D.3 in Appendix D).

*Remark* 4.11 (**Novelty and relation to prior analyses**). Partial-participation FL results for non-convex objectives (Yang et al., 2021) control stationarity for a *single*, stationary task and do not model task-to-task drift, so they cannot yield guarantees for $\sum_{i=1}^{K} f_i$. Measures of task relatedness based on gradient inner products (Han et al., 2023b) or sufficient projection/positive correlation (Lin et al., 2022) provide useful notions of similarity but stop short of a *task-uniform* rate under non-IID clients with round-wise subsampling. Decentralized continual-learning analyses such as Choudhary et al. (2023) study different communication graphs and often assume IID data across clients, making them non-transferable to our federated, non-IID setting. Theorem 4.10 closes this gap for FDIL with partial participation: it establishes a task-uniform stationarity bound in the presence of client heterogeneity, and it introduces an explicit inter-task drift term $\sigma_T^2$ that quantifies how domain shift scales with $K$. The server-side proximal anchor is essential, via the

uniform drift bound in Lemma 4.9, to control within-task deviation and propagate stability across tasks *without* replay buffers or architectural expansion.

**Corollary 4.12.** *Let* $\gamma_L = \frac{\lambda}{\sqrt{KT}EL}$, $\gamma_G = \frac{\sqrt{NE}}{(K-1)\lambda L}$, *and* $\left\| \nabla f_{1:K-1} \left( \theta_K^0 \right) \right\|^2 \leq C$, *where $C$ is a constant. While training on task $K$, the task-uniform convergence rate of* SPECIAL *satisfies*

$$
\min_{t \in [T]} \mathbb{E} \left\| \nabla f_{1:K} \left( \theta_K^t \right) \right\|^2 = \mathcal{O} \left( \sqrt{E/(NT)} \right).
$$

*Remark* 4.13 (Trade-off between computation and communication). The rate improves with larger $T$ and $N$, matching intuition from standard FL, and it highlights the classic tension between local computation and communication (Stich, 2019; Li et al., 2020c; Yang et al., 2021). If $E$ is too large, each client over-optimizes its own data and the global iterate $\theta_K^t$ drifts toward the minimizer of the last task $f_K$ instead of the joint objective $f_{1:K}$. If $E$ is too small, many more rounds are required to reach comparable stationarity.

### 4.4. Intuitions and Proof Sketch

We now highlight key ideas and challenges behind the **task-uniform** convergence of SPECIAL. Compared with convergence proofs in FL, the key distinction lies in characterizing the relation across tasks. Following Assumption 4.2 and decomposing $\nabla f_{1:K}$, the loss function can be expanded as:

$$
\begin{aligned}
\mathbb{E}_t \left[ f_{1:K} \left( \theta_K^{t+1} \right) \right] \leq& f_{1:K} \left( \theta_K^t \right) - \frac{\gamma_G\gamma_L E}{1+\lambda} \underbrace{\left\| \nabla f_K \left( \theta_K^t \right) \right\|^2}_{B_1} \\
&- \frac{\gamma_G\gamma_L E}{1+\lambda} \underbrace{\left\langle \nabla f_{1:K-1} \left( \theta_K^t \right), \nabla f_K \left( \theta_K^t \right) \right\rangle}_{B_2} \\
&+ \underbrace{\left\langle \nabla f_{1:K} \left( \theta_K^t \right), \mathbb{E}_t \left[ \theta_K^{t+1} - \theta_K^t \right] + \frac{\gamma_G\gamma_L E}{1+\lambda} \nabla f_K \left( \theta_K^t \right) \right\rangle}_{B_3} \\
&+ \frac{KL}{2} \underbrace{\mathbb{E}_t \left\| \theta_K^{t+1} - \theta_K^t \right\|^2}_{B_4},
\end{aligned}
$$

where $B_1$ represents the convergence rate of the single task, $B_2$ measures the alignment of $\nabla f_{1:K-1}$ and $\nabla f_K$, and the last two terms characterize the magnitude of the model update in a single communication round. Below, we highlight the key differences: 1) *Multiple gradients.* The expansion involves two types of gradients: the single-task gradient $\|\nabla f_K (\theta_K^t)\|^2$ and task-uniform gradient $\|\nabla f_{1:K} (\theta_K^t)\|^2$, where $\|\nabla f_K (\theta_K^t)\|^2$ should be canceled out based on its relation with $\|\nabla f_{1:K} (\theta_K^t)\|^2$ to address the task-uniform convergence rate. 2) *Partial participation.* Terms $B_3$ and $B_4$ both involve the one-round deviation $\theta_K^{t+1} - \theta_K^t$, and the deviation in $B_3$ is equivalent to the deviation under full

*Table 1.* **Main FDIL results under partial participation.** Average accuracy (higher is better) and backward transfer (less negative/higher is better) on Digit-10, VLCS, PACS, and DN4IL with $M = 8$ clients and $N = 4$ sampled per round. We compare standard-FL, replay-based, and replay-free FCL baselines. SPECIAL achieves the best ACC on all datasets while maintaining competitive BWT among replay-free methods. Within each column, darker cell colors indicate better performance.

| Type | Method | Digit-10 | | VLCS | | PACS | | DN4IL | |
|---|---|---|---|---|---|---|---|---|---|
| | | ACC (%) ↑ | BWT (%) ↑ | ACC (%) ↑ | BWT (%) ↑ | ACC (%) ↑ | BWT (%) ↑ | ACC (%) ↑ | BWT (%) ↑ |
| Standard-FL | FedAvg | $58.69 \pm 2.06$ | $-40.44 \pm 2.07$ | $51.22 \pm 1.65$ | $-9.22 \pm 3.78$ | $37.79 \pm 0.81$ | $-22.32 \pm 3.48$ | $16.47 \pm 0.16$ | $-33.91 \pm 0.11$ |
| | FedProx | $58.99 \pm 2.51$ | $-30.19 \pm 2.07$ | $48.24 \pm 1.53$ | $-3.86 \pm 1.11$ | $42.99 \pm 0.97$ | $-19.48 \pm 2.01$ | $21.17 \pm 0.21$ | $-21.40 \pm 0.34$ |
| Replay-based | FedCIL | $50.95 \pm 1.66$ | $-27.56 \pm 4.97$ | $48.84 \pm 0.69$ | $-8.88 \pm 3.55$ | $35.11 \pm 0.77$ | $-10.30 \pm 2.08$ | $14.29 \pm 0.41$ | $-23.01 \pm 0.15$ |
| | MFCL | $61.15 \pm 1.43$ | $-19.58 \pm 2.02$ | $42.89 \pm 5.95$ | $-11.03 \pm 1.99$ | $34.68 \pm 1.05$ | $-14.84 \pm 4.21$ | $19.00 \pm 1.06$ | $-21.78 \pm 1.20$ |
| | SR-FDIL | $61.63 \pm 1.47$ | $-36.38 \pm 2.64$ | $49.38 \pm 2.47$ | $-3.22 \pm 7.26$ | $44.92 \pm 0.92$ | $-16.05 \pm 1.58$ | $21.40 \pm 0.25$ | $-33.38 \pm 0.57$ |
| Replay-free | pFedDIL | $46.17 \pm 2.16$ | $-20.21 \pm 0.31$ | $52.44 \pm 1.93$ | $-8.71 \pm 5.87$ | $39.76 \pm 1.98$ | $-21.85 \pm 0.86$ | $12.47 \pm 0.30$ | $\mathbf{-4.63 \pm 0.14}$ |
| | FLwF-2T | $47.39 \pm 5.35$ | $-33.02 \pm 4.94$ | $53.24 \pm 1.43$ | $-0.32 \pm 6.25$ | $43.04 \pm 2.49$ | $\mathbf{-8.12 \pm 5.24}$ | $23.21 \pm 0.49$ | $-16.29 \pm 0.15$ |
| | SPECIAL-C | $50.61 \pm 2.89$ | $\mathbf{-16.42 \pm 0.57}$ | $51.17 \pm 0.44$ | $-8.98 \pm 4.13$ | $43.30 \pm 0.84$ | $-16.40 \pm 1.06$ | $19.88 \pm 0.34$ | $-23.21 \pm 0.43$ |
| | **SPECIAL** (ours) | $\mathbf{62.12 \pm 0.18}$ | $-21.78 \pm 2.94$ | $\mathbf{54.92 \pm 1.60}$ | $\mathbf{-0.11 \pm 2.51}$ | $\mathbf{45.29 \pm 0.92}$ | $-11.66 \pm 1.13$ | $\mathbf{24.30 \pm 0.15}$ | $-19.50 \pm 0.41$ |

participation in expectation, while $B_4$ is analyzed under partial participation since the deviation is embedded within the $\ell_2$-norm, so we introduced Lemma C.6 and Lemma C.7 to bridge the two terms.

## 5. Experiments

We now evaluate SPECIAL. Full setup and hyperparameters appear in Appendix F.2.

**Datasets and model.** We use four domain-shift benchmarks: Digit-10, VLCS (Torralba & Efros, 2011), PACS (Li et al., 2017), and DN4IL (Gowda et al., 2023), all with a ResNet-18 backbone (He et al., 2016).

**Baselines.** We compare against two FCL families: (i) *Replay-based*: FedCIL (Qi et al., 2023), MFCL (Babakniya et al., 2023a), SR-FDIL (Li et al., 2024c). (ii) *Replay-free*: pFedDIL (Li et al., 2024b), FLwF-2T (Usmanova et al., 2022), and SPECIAL-C (client-side variant). We also include *FedAvg* (McMahan et al., 2017) and *FedProx* (Li et al., 2020a) as conventional FL anchors.

**Configurations.** Unless noted, local epochs $E = 5$. We run $T = 20$ rounds on Digit-10 and VLCS, and $T = 30$ on PACS and DN4IL. Non-IID partitions use Dirichlet sampling with $\alpha = 0.1$ (Digit-10), $\alpha = 0.3$ (VLCS, PACS), and $\alpha = 5$ (DN4IL). Local batch size is 32 with learning rate $10^{-3}$; the global step decays with task index, $\gamma_G = 1/i$. We use $M = 8$ clients and sample $N = 4$ per round. Implementations are in PyTorch (Python 3) on two NVIDIA Quadro RTX 8000 GPUs. Results are averaged over seeds $\{25, 225, 2025\}$. See Appendix F.2 for method-specific grids and selected values.

**Evaluation metrics.** Following Lin et al. (2022), we report

$$\text{ACC} = \frac{1}{K}\sum_{i=1}^{K} A_{K,i}, \quad \text{BWT} = \frac{1}{K-1}\sum_{i=1}^{K-1}(A_{K,i} - A_{i,i}). \quad (10)$$

Here $A_{i,j}$ is the test accuracy on task $j$ after finishing task $i$.

**Main results.** Table 1 shows that SPECIAL attains the

*Table 2.* Task-wise final accuracies on VLCS ($\alpha = 0.3$), comparing SPECIAL to the best-performing baseline on VLCS (FLwF-2T). Higher is better.

| Method | Caltech101 | LabelMe | SUN09 | VOC2007 | ACC |
|---|---|---|---|---|---|
| FLwF-2T (best baseline) | 0.6933 | 0.5229 | 0.4532 | 0.4604 | 0.5324 |
| **SPECIAL** (ours) | **0.7056** | **0.5385** | **0.4813** | **0.4715** | **0.5492** |

highest ACC on all datasets and competitive BWT among *replay-free* methods. Replay-based approaches (SR-FDIL, FedCIL, MFCL) can approach SPECIAL on ACC but require replay or generators. Note that BWT is *relative* by Eq. (10): methods that underfit current tasks can appear to forget less (less negative BWT) because $A_{i,i}$ may already be low and $A_{K,i}$ changes little, reflecting stability at the expense of plasticity. SPECIAL pairs strong ACC with comparatively high BWT among high-ACC baselines, indicating effective forgetting mitigation without impeding adaptation.

*Where the proximal term is applied matters:* Table 1 and Figure 2 together show that the proximal anchor's location is critical. On Digit-10, whose tasks are most correlated, SPECIAL-C (client-side proximal) performs "repeated reviewing" at every local step, yielding slightly lower ACC but higher BWT than server-side SPECIAL, which helps preserve earlier knowledge during the first rounds of next task. When task correlation weakens (VLCS, PACS, DN4IL), continual client-side reviewing slows adaptation to the new domain and does not improve backward transfer; SPECIAL-C trails SPECIAL on both ACC and BWT. This pattern is consistent with our theory: the server-side anchor controls *between-task* drift without throttling *within-task* plasticity.

*Why FedAvg can look strong:* It is expected that *FedAvg* occasionally matches or exceeds some continual baselines on ACC in short horizons because it directly optimizes the current task objective $f_K$ with no mechanism to control inter-task drift. Such wins typically coincide with more negative backward transfer and larger drops on earlier tasks in the per-task matrix $A_{K,i}$ (Appendix F.3); for completeness we also

*Table 3.* **Communication and runtime cost.** *Rounds-to-best* reports the sum (over tasks) of the number of communication rounds needed to reach each task's best performance; *Time/round* is the average wall-clock time per round across tasks. Lower is better for both metrics. SPECIAL matches a FedAvg-like footprint (no extra client training or communication) while reaching best performance in fewer rounds.

| | Method | Digit-10 | | VLCS | | PACS | | DN4IL | |
|---|---|---|---|---|---|---|---|---|---|
| | | Rounds(r) | Time(s) | Rounds(r) | Time(s) | Rounds(r) | Time(s) | Rounds(r) | Time(min) |
| Standard-FL | FedAvg | 83 | **191.27** | 77 | **267.46** | 84 | **25.44** | 171 | **15.87** |
| | FedProx | 72 | 483.19 | 68 | 290.42 | 77 | 55.28 | 157 | 17.04 |
| Replay-based | FedCIL | 68 | 523.16 | 65 | 382.20 | 81 | 38.24 | 165 | 17.85 |
| | MFCL | 77 | 969.01 | 71 | 491.46 | 83 | 129.13 | 160 | 38.77 |
| | SR-FDIL | 71 | 357.34 | 62 | 313.21 | 76 | 50.64 | 161 | 18.42 |
| Replay-free | pFedDIL | 76 | 699.43 | 69 | 519.31 | 84 | 75.69 | 167 | 30.71 |
| | FLwF-2T | 67 | 247.90 | 59 | 291.40 | 76 | 43.43 | 156 | 19.33 |
| | SPECIAL-C | 74 | 518.26 | 61 | 283.86 | 75 | 91.98 | 159 | 19.43 |
| | **SPECIAL** (ours) | **63** | 200.66 | **55** | 294.76 | **67** | 26.93 | **145** | 16.23 |

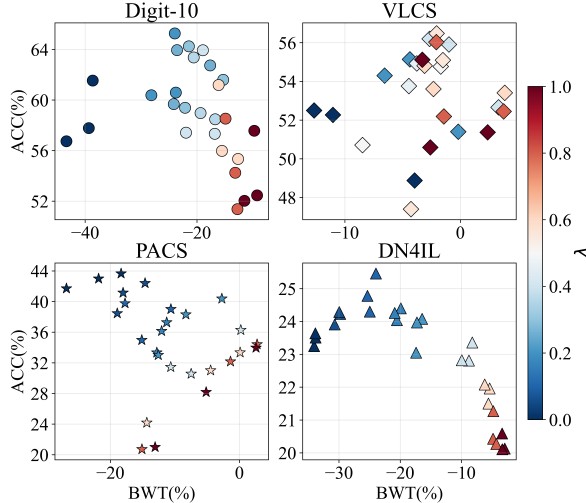

*Figure 3.* **Plasticity-stability trade-off controlled by the server proximal weight $\lambda$.** Each point shows (ACC, BWT) for a fixed $\lambda$; markers denote datasets and color encodes $\lambda$ (low $\rightarrow$ high). Larger $\lambda$ increases stability (improving BWT) but over-anchoring can reduce plasticity (lowering ACC), yielding best joint performance at intermediate $\lambda$.

report the worst earlier-task drop $\min_{i \leq K-1}(A_{K,i} - A_{i,i})$ in Table 8.

**Per-task performance.** Table 2 reports task-wise final accuracies on VLCS, comparing SPECIAL to the strongest baseline on this dataset (FLwF-2T). SPECIAL improves accuracy on each domain. Full per-task results (all datasets and baselines) are provided in Appendix F.3.

**Influence of the proximal weight $\lambda$.** Figure 3 visualizes the ACC-BWT trade-off as $\lambda$ varies (color encodes $\lambda$, low to high). Recall that $\lambda$ balances *current-task plasticity* against *past-task stability* when the server combines the intermediate model with the anchor from the previous task. Three consistent patterns emerge. (i) *BWT increases with $\lambda$.* Larger $\lambda$ strengthens the server anchor and reduces between-task

drift, so forgetting decreases. This matches Lemma 4.9, where the within-task deviation bound scales as $1/\lambda^2$. (ii) *ACC is unimodal.* Very small $\lambda$ favors plasticity and yields high adaptation on the current task but severe forgetting; very large $\lambda$ over-anchors to the previous model and suppresses learning on the new domain, which depresses ACC. Theorem 4.10 reflects this balance: the leading descent term contains $(1+\lambda)^{-1} E \gamma_G \gamma_L T$, so increasing $\lambda$ tightens stability but effectively slows progress on $f_K$ when step sizes are fixed. (iii) *Best joint performance occurs at mid-range $\lambda$.* Points near the upper-right of the plot correspond to intermediate anchors that balance stability and plasticity, consistent with the backward-forward transfer trade-off discussed in Zhang et al. (2023).

In practice, the optimal $\lambda$ is data dependent. We select it by a small sweep and then fix the value across all tasks and seeds for a dataset (Appendix F.2). The chosen values are $\lambda = 0.25$ (Digit-10), $0.40$ (VLCS), $0.05$ (PACS), and $0.10$ (DN4IL). Although $\lambda$ is selected per dataset, the performance is not always knife-edge sensitive. For example, on Digit-10, setting $\lambda = 0.2$ degrades ACC by only $0.06\%$ compared with the selected value, while $\lambda = 0.3$ leads to only a $0.6\%$ degradation. This suggests that strong performance can be obtained over a local band of mid-range $\lambda$ values.

**Computational complexity and communication cost.** Table 3 reports the total rounds-to-best (summed over tasks) and the average wall-clock time per round. Across all four datasets, SPECIAL reaches its best performance in the fewest rounds. Compared with methods with knowledge-transfer mechanism, it also achieves the best per-round time on three datasets and the second-best on VLCS, showing that SPECIAL is both communication-efficient and runtime-competitive. These trends align with algorithmic overheads. Replay-based methods incur extra cost beyond local SGD: FedCIL/MFCL train an additional generative model (on clients or the server), and SR-FDIL performs additional scoring/selection to maintain an exemplar subset. Among replay-

*Table 4.* Results of DistilBERT-based model on the NLP task sequence ($\alpha = 1.0$).

| Method | ACC (%) ↑ | BWT (%) ↑ |
|---|---|---|
| FedAvg | $65.19 \pm 0.26$ | $-7.00 \pm 0.62$ |
| FedProx | $58.91 \pm 1.31$ | $-16.80 \pm 0.98$ |
| pFedDIL | $66.13 \pm 3.24$ | $-6.80 \pm 2.06$ |
| FLwF-2T | $69.15 \pm 0.52$ | $-5.50 \pm 0.68$ |
| SPECIAL-C | $61.37 \pm 1.89$ | $-6.64 \pm 1.25$ |
| **SPECIAL** | $\mathbf{74.70 \pm 0.78}$ | $\mathbf{-4.03 \pm 2.93}$ |

free baselines, pFedDIL introduces an auxiliary classifier and extra optimization objectives, while other regularization-based approaches typically add penalties to the *client-side* loss, which can slow or complicate local optimization. In contrast, SPECIAL keeps client training identical to FedAvg (plain SGD) and applies the proximal anchor *only at the server* via a closed-form blend with the previous global model. Thus, it adds negligible overhead per round and no extra communication, matching a FedAvg-like footprint while achieving faster communication progress, consistent with Corollary 4.12. More discussions and numerical comparisons about the efficiency are available in Appendix F.7.

**Performance on language tasks.** To evaluate the adaptability of SPECIAL beyond vision tasks, we use a pre-trained DistilBERT-based model on a GLUE task sequence consisting of SST-2, QQP, and QNLI. The configuration is kept consistent with the vision experiments, except that we set $\alpha = 1.0$ for the non-IID partition and use AdamW for local optimization. We set $\lambda = 0.45$ to balance plasticity and stability. Since the three replay-based methods are primarily designed for vision-based continual learning and rely on image-oriented mechanisms, we compare against the remaining applicable baselines. Table 4 reports the results. SPECIAL achieves the best ACC and strongest BWT among these baselines, suggesting that the proposed server-side anchoring mechanism can extend beyond vision tasks.

## 6. Conclusion and Limitations

We proposed SPECIAL as a lightweight replay-free algorithm for FDIL under partial participation. The method introduces a server-side proximal anchor that preserves a FedAvg-like client workflow while controlling inter-task drift. We established two theoretical results: a backward knowledge transfer guarantee and a task-uniform convergence guarantee for non-convex FDIL under partial participation. Our theory assumes bounded gradients, $L$-smoothness, and synchronous partial participation; it does not model stragglers, stale updates, or other system-level constraints. Empirically, we evaluated vision FDIL with a

fixed ResNet-18 backbone and language FDIL with a pre-trained DistilBERT-based backbone. The proximal weight $\lambda$ was tuned per dataset. Future work will relax the technical assumptions, extend SPECIAL to asynchronous and drop-tolerant aggregation, develop self-tuning rules for $\lambda$, and broaden evaluation to richer continual streams.

## Acknowledgements

This work was supported in part by the National Science Foundation (NSF) grants 2148309, 2315614 and 2337914, and the National Institutes of Health (NIH) grant 1R01HL184139-01, and was supported in part by funds from OUSD R&E, NIST, and industry partners as specified in the Resilient & Intelligent NextG Systems (RINGS) program. Any opinions, findings, and conclusions or recommendations expressed in this material are those of the authors and do not necessarily reflect the views of the funding agencies.

## Impact Statement

This paper presents work whose goal is to advance the field of Machine Learning. There are many potential societal consequences of our work, none which we feel must be specifically highlighted here.

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

# A. Related work

## A.1. Continual Learning

Continual learning aims to train a model on a sequence of tasks under restriction to access to previous data with the goal of mitigating catastrophic forgetting (Wang et al., 2024; Chen & Liu, 2018; Schwarz et al., 2018; Li et al., 2019). CL methods are typically categorized into two main types: (i) experience replay methods retain or generate synthetic samples from previous tasks to be combined with current task data to be trained (Rebuffi et al., 2017; Wu et al., 2018); (ii) regularization-based methods constrain parameter updates by introducing proximal terms or distillation terms into the loss function to preserver previous information (Li & Hoiem, 2017b; Aich, 2021; Aljundi et al., 2018). From the theoretical perspective, Lin et al. (2022) analyzed backward transfer and inter-task relations with convergence guarantees, while Han et al. (2023a) introduced a decomposition of the loss to separately study overfitting and forgetting under non-convex settings. In this work, we extend CL to the federated setting and build on existing FL theoretical frameworks to establish stronger convergence guarantees for Federated Continual Learning (FCL).

## A.2. Federated Continual Learning

The essence of federated continual learning lies in enabling a model to sequentially learn from distributed tasks across clients while mitigating catastrophic forgetting in a decentralized setting. Studies like Li et al. (2024a;c) adopt experience replay by storing samples with higher importance calculated based on their behaviors during training, but the overhead of calculating and storing sufficient previous samples makes these methods eclipsed compared with generating synthetic data (Babakniya et al., 2023c; Qi et al., 2023; Wuerkaixi et al., 2024), such as TARGET (Zhang et al., 2023), MFCL(Babakniya et al., 2023b) and AF-FCL(Wuerkaixi et al., 2024). Other methods, such as FedWeIT (Yoon et al., 2021) and FLwF-2T (Usmanova et al., 2021), achieve knowledge transfer between tasks via parameter partitioning and knowledge distillation which can be regarded as two main directions of regularization-based methods. However, most FCL methods lack theoretical guarantees. Keshri et al. (2025) extends the proof in CL setting proposed by Han et al. (2023b) to FCL setting, but they research on the convergence rate of loss function only on the last task, while the target of FCL is to ensure convergence across all tasks. In addition, Marfoq et al. (2023) studies convergence under streaming data and provides the theoretical support, but their assumptions are too restrictive. In our work, we establish a solid theoretical guarantee on the convergence rate and performance improvement under reasonable and practically applicable assumptions.

## A.3. Federated Domain-Incremental Learning

Federated Domain-Incremental Learning (FDIL) aims to learn from a sequence of tasks with different domains, which differs from Federated Class-Incremental Learning (FCIL). Existing FDIL methods focus on extracting inter-domain similarities or caching important previous data to mitigate the catastrophic forgetting. Li et al. (2024b) proposed pFedDIL, which selects previous models to be the initial model for the new task based on knowledge intensity and performs knowledge migration by adding the Knowledge Migration (KM) loss. RefFiL (Sun et al., 2025) addresses FDIL by learning domain-invariant knowledge and incorporates domain-specific prompts to alleviate catastrophic forgetting. Similar to ReFed (Li et al., 2024a), SR-FDIL (Li et al., 2024c) enables clients to cache the important samples which can be reuse in the further tasks, and the importance is calculated based on the domain-representative score and cross-client collaborative score. Unlike prior FDIL methods that require significant computational time to prepare for the next arrival task between tasks, our approach transfers knowledge by storing the previous global model, reducing preparation time and preserving privacy.

# B. Detailed Notation

The detailed notation is listed in Table 5.

# C. Lemma

In the followings, we list a series of lemmas that we use in the theoretical analysis.

**Lemma C.1.** *Suppose function $f_1, \ldots, f_K$ are L-smooth, then $f_{1:K} = \sum_{i=1}^{K} f_i$ is $KL$-smooth.*

*Table 5.* Notations and Terminologies.

| Notation | Description |
|---|---|
| $M$ | Number of all clients |
| $N$ | Number of participated clients per round |
| $T$ | Number of global epochs |
| $E$ | Number of local updates |
| $K$ | Number of tasks |
| $[n]$ | Set of integers $\{1, \ldots, n\}$ |
| $\theta_i$ | Global model after completing training on task $i$ |
| $\theta_i^t$ | Global model at round $t$ in task $i$ |
| $\theta_{i,m}^t$ | Local model on client $m$ at round $t$ in task $i$ |
| $\theta_{i,m}^{t,e}$ | Local model on client $m$ at epoch $e$ of round $t$ in task $i$ |
| $\Delta_{i,m}^t$ | Update vector on client $m$ of round $t$ in task $i$ |
| $f_i$ | Loss function of task $i$ |
| $f_{1:K}$ | Loss function over task 1 to task $K$ |
| $\nabla f_i$ | the gradient of $f_i$ |
| $g_{i,m}^{t,e}$ | Gradient calculated on client m at epoch $e$ of round $t$ in task $i$ |
| $\mathcal{T}$ | Set of all tasks, i.e., $\{\mathcal{T}_1, \ldots, \mathcal{T}_K\}$ |
| $\mathcal{T}_i$ | Dataset of the task $i$ |
| $\mathcal{T}_{i,m}$ | Dataset of client $m$ on task $i$ |
| $\lambda$ | Regularization coefficient |
| $\|\cdot\|^2$ | $\ell_2$-norms |

**Lemma C.2.** *For $x_1, \ldots, x_n \in \mathbb{R}^d$ and $a_1, \ldots, a_n \geq 0$, we have:*

$$\|a_1 x_1 + \cdots + a_n x_n\| \leq \sum_{i=1}^{n} \|a_i x_i\| = \sum_{i=1}^{n} a_i \|x_i\|. \tag{11}$$

**Lemma C.3.** *For random variables $x_1, \ldots, x_n$, we have:*

$$\mathbb{E} \|x_1 + \cdots + x_n\|^2 \leq n \sum_{i=1}^{n} \mathbb{E} \|x_i\|^2. \tag{12}$$

**Lemma C.4.** *For random variables $x_1, \ldots, x_n$ with $\mathbb{E}[x_i] = 0, \forall i \in [n]$, we have:*

$$\mathbb{E} \|x_1 + \cdots + x_n\|^2 \leq \sum_{i=1}^{n} \mathbb{E} \|x_i\|^2. \tag{13}$$

**Lemma C.5** (Lemma 2 in (Yang et al., 2021)). *For any local learning rate satisfying $\gamma_L \leq \frac{1}{8LE}$, we can have the following results:*

$$\frac{1}{M} \sum_{i=1}^{M} \mathbb{E} \left[ \left\| \theta_{i,m}^{t,e} - \theta_i^t \right\|^2 \right] \leq 5\gamma_L^2 E \left( \sigma_L^2 + 6E\sigma_G^2 \right) + 30\gamma_L^2 E^2 \left\| \nabla f_i \left( \theta_i^t \right) \right\|^2. \tag{14}$$

*Proof.* For any client $m \in [M]$, task $i \in [K]$, and $e \in \{0, \ldots, E-1\}$, we have:

$$\mathbb{E} \left[ \left\| \theta_{i,m}^{t,e} - \theta_i^t \right\|^2 \right] = \mathbb{E} \left[ \left\| \theta_{i,m}^{t,e-1} - \theta_i^t - \gamma_L g_{i,m}^{t,e-1} \right\|^2 \right]$$

$$= \mathbb{E} \left[ \left\| \theta_{i,m}^{t,e-1} - \theta_i^t - \gamma_L \left( g_{i,m}^{t,e-1} - \nabla f_{i,m} \left( \theta_{i,m}^{t,e-1} \right) \right. \right. \right.$$

$$\left. \left. \left. + \nabla f_{i,m} \left( \theta_{i,m}^{t,e-1} \right) - \nabla f_{i,m} \left( \theta_i^t \right) + \nabla f_{i,m} \left( \theta_i^t \right) - \nabla f_i \left( \theta_i^t \right) + \nabla f_i \left( \theta_i^t \right) \right) \right\|^2 \right]$$

$$\leq \left(1 + \frac{1}{2E-1}\right) \mathbb{E}\left[\left\|\theta_{i,m}^{t,e-1} - \theta_i^t\right\|^2\right] + \gamma_L^2 \mathbb{E}\left[\left\|g_{i,m}^{t,e-1} - \nabla f_{i,m}\left(\theta_{i,m}^{t,e-1}\right)\right\|^2\right]$$

$$+ 6\gamma_L^2 E \cdot \mathbb{E}\left[\left\|\nabla f_{i,m}\left(\theta_{i,m}^{t,e-1}\right) - \nabla f_{i,m}\left(\theta_i^t\right)\right\|^2 + \left\|\nabla f_{i,m}\left(\theta_i^t\right) - \nabla f_i\left(\theta_i^t\right)\right\|^2 + \left\|\nabla f_i\left(\theta_i^t\right)\right\|^2\right]$$

$$\overset{(l_1)}{\leq} \left(1 + \frac{1}{2E-1}\right) \mathbb{E}\left[\left\|\theta_{i,m}^{t,e-1} - \theta_i^t\right\|^2\right] + \gamma_L^2 \sigma_L^2$$

$$+ 6\gamma_L^2 E \left[L^2 \mathbb{E}\left[\left\|\theta_{i,m}^{t,e-1} - \theta_i^t\right\|^2\right] + \sigma_G^2 + \left\|\nabla f_i\left(\theta_i^t\right)\right\|^2\right]$$

$$\overset{(l_2)}{=} \left(1 + \frac{1}{2E-1} + 6\gamma_L^2 E L^2\right) \mathbb{E}\left[\left\|\theta_{i,m}^{t,e-1} - \theta_i^t\right\|^2\right] + \gamma_L^2 \sigma_L^2 + 6\gamma_L^2 E \sigma_G^2 + 6\gamma_L^2 E \left\|\nabla f_i\left(\theta_i^t\right)\right\|^2$$

$$\leq \left(1 + \frac{1}{E-1}\right) \mathbb{E}\left[\left\|\theta_{i,m}^{t,e-1} - \theta_i^t\right\|^2\right] + \gamma_L^2 \sigma_L^2 + 6\gamma_L^2 E \sigma_G^2 + 6\gamma_L^2 E \left\|\nabla f_i\left(\theta_i^t\right)\right\|^2, \tag{15}$$

where $(l_1)$ follows Assumption 4.4 and $(l_2)$ is due to Assumption 4.5.

Unrolling the recursion, we obtain:

$$\frac{1}{M}\sum_{m=1}^{M} \mathbb{E}\left[\left\|\theta_{i,m}^{t,e} - \theta_i^t\right\|^2\right] \leq \sum_{e=0}^{E-1}\left(1 + \frac{1}{E-1}\right)^e \left[\gamma_L^2 \sigma_L^2 + 6\gamma_L^2 E \sigma_G^2 + 6\gamma_L^2 E \left\|\nabla f_i\left(\theta_i^t\right)\right\|^2\right]$$

$$\leq (E-1)\left[\left(1 + \frac{1}{E-1}\right)^E - 1\right]\left[\gamma_L^2 \sigma_L^2 + 6\gamma_L^2 E \sigma_G^2 + 6\gamma_L^2 E \left\|\nabla f_i\left(\theta_i^t\right)\right\|^2\right] \tag{16}$$

$$\overset{(l_3)}{\leq} 5\gamma_L^2 E \left(\sigma_L^2 + 6E\sigma_G^2\right) + 30\gamma_L^2 E^2 \left\|\nabla f_i\left(\theta_i^t\right)\right\|^2,$$

where $(l_3)$ follows the fact that $\left(1 + \frac{1}{E-1}\right)^E \leq 5$ for $E > 1$. $\qquad\square$

**Lemma C.6.** *Assume $\boldsymbol{t}_m \in \mathbb{R}^d, m \in [M], \Lambda \in \mathbb{R}^d$, and $G$ is a constant, then we have:*

$$\left\|\sum_{m=1}^{M} G\boldsymbol{t}_m + \Lambda\right\|^2 = \left(MG^2 + G\right)\sum_{m=1}^{M}\|\boldsymbol{t}_m\|^2 - \frac{G^2}{2}\sum_{m\neq n}\|\boldsymbol{t}_m - \boldsymbol{t}_n\|^2$$

$$+ (1 + MG)\|\Lambda\|^2 - G\sum_{m=1}^{M}\|\boldsymbol{t}_m - \Lambda\|^2. \tag{17}$$

*Proof.*

$$\left\|\sum_{m=1}^{M} G\boldsymbol{t}_m + \Lambda\right\|^2 G^2\sum_{m=1}^{M}\|\boldsymbol{t}_m\|^2 + G^2\sum_{m\neq n}\langle \boldsymbol{t}_m, \boldsymbol{t}_n\rangle + \|\Lambda\|^2 + 2G\sum_{m=1}^{M}\langle \boldsymbol{t}_m, \Lambda\rangle$$

$$\overset{(l_4)}{=} G^2\sum_{m=1}^{M}\|\boldsymbol{t}_m\|^2 + G^2\sum_{m\neq n}\frac{1}{2}\left[\|\boldsymbol{t}_m\|^2 + \|\boldsymbol{t}_n\|^2 - \|\boldsymbol{t}_m - \boldsymbol{t}_n\|^2\right]$$

$$+ \|\Lambda\|^2 + G\sum_{m=1}^{M}\left[\|\boldsymbol{t}_m\|^2 + \|\Lambda\|^2 - \|\boldsymbol{t}_m - \Lambda\|^2\right]$$

$$= \left(MG^2 + G\right)\sum_{m=1}^{M}\|\boldsymbol{t}_m\|^2 - \frac{G^2}{2}\sum_{m\neq n}\|\boldsymbol{t}_m - \boldsymbol{t}_n\|^2$$

$$+ (1 + MG)\|\Lambda\|^2 - G\sum_{m=1}^{M}\|\boldsymbol{t}_m - \Lambda\|^2, \tag{18}$$

where $(l_4)$ follows that $\|\boldsymbol{x} - \boldsymbol{y}\|^2 = \frac{1}{2}\left[\|\boldsymbol{x}\|^2 + \|\boldsymbol{y}\|^2 - \|\boldsymbol{x} - \boldsymbol{y}\|^2\right]$. $\qquad\square$

**Lemma C.7.** *Assume $\boldsymbol{t}_m \in \mathbb{R}^d, m \in [M], \Lambda \in \mathbb{R}^d$, $G$ is a constant. If we randomly select $N$ clients from all $M$ clients, and the $\mathcal{S}$ is the set of the client index, then we have:*

$$\left\| \sum_{m=1}^M G\mathbb{P}\{m \in \mathcal{S}\}\boldsymbol{t}_m + \Lambda \right\|^2 = \left( \frac{N^2G^2}{M} + \frac{NG}{M} \right) \sum_{m=1}^M \|\boldsymbol{t}_m\|^2 - \frac{N(N-1)G^2}{2M(M-1)} \sum_{m \neq n} \|\boldsymbol{t}_m - \boldsymbol{t}_n\|^2$$

$$+ (1 + NG)\|\Lambda\|^2 - \frac{NG}{M} \sum_{m=1}^M \|\boldsymbol{t}_m - \Lambda\|^2. \tag{19}$$

*Proof.*

$$\left\| \sum_{m=1}^M G\mathbb{P}\{m \in \mathcal{S}\}\boldsymbol{t}_m + \Lambda \right\|^2$$

$$= G^2 \sum_{m=1}^M P\{m \in \mathcal{S}\} \|\boldsymbol{t}_m\|^2 + G^2 \sum_{m \neq n} P\{m, n \in \mathcal{S}\} \langle \boldsymbol{t}_m, \boldsymbol{t}_n \rangle + \|\Lambda\|^2$$

$$+ 2G \sum_{m=1}^M P\{m \in \mathcal{S}\} \langle \boldsymbol{t}_m, \Lambda \rangle$$

$$= \frac{NG^2}{M} \sum_{m=1}^M \|\boldsymbol{t}_m\|^2 + \frac{N(N-1)G^2}{M(M-1)} \sum_{m \neq n} \langle \boldsymbol{t}_m, \boldsymbol{t}_n \rangle + \|\Lambda\|^2 + \frac{2NG}{M} \sum_{m=1}^M \langle \boldsymbol{t}_m, \Lambda \rangle$$

$$\overset{(l_5)}{=} \frac{NG^2}{M} \sum_{m=1}^M \|\boldsymbol{t}_m\|^2 + \frac{N(N-1)G^2}{M(M-1)} \sum_{m \neq n} \frac{1}{2} \left[ \|\boldsymbol{t}_m\|^2 + \|\boldsymbol{t}_n\|^2 - \|\boldsymbol{t}_m - \boldsymbol{t}_n\|^2 \right] + \|\Lambda\|^2 \tag{20}$$

$$+ \frac{NG}{M} \sum_{m=1}^M \left[ \|\boldsymbol{t}_m\|^2 + \|\Lambda\|^2 - \|\boldsymbol{t}_m - \Lambda\|^2 \right]$$

$$= \left( \frac{N^2G^2}{M} + \frac{NG}{M} \right) \sum_{m=1}^M \|\boldsymbol{t}_m\|^2 - \frac{N(N-1)G^2}{2M(M-1)} \sum_{m \neq n} \|\boldsymbol{t}_m - \boldsymbol{t}_n\|^2$$

$$+ (1 + NG)\|\Lambda\|^2 - \frac{NG}{M} \sum_{m=1}^M \|\boldsymbol{t}_m - \Lambda\|^2,$$

where $(l_5)$ follows that $\|\boldsymbol{x} - \boldsymbol{y}\|^2 = \frac{1}{2} \left[ \|\boldsymbol{x}\|^2 + \|\boldsymbol{y}\|^2 - \|\boldsymbol{x} - \boldsymbol{y}\|^2 \right]$. $\qquad\square$

**Lemma C.8.** *Assume $\boldsymbol{t}_m, \boldsymbol{t}_n \in \mathbb{R}^d, m, n \in [M]$, then we have:*

$$\sum_{m \neq n} \|\boldsymbol{t}_m - \boldsymbol{t}_n\|^2 = 2M \sum_{m=1}^M \|\boldsymbol{t}_m\|^2 - 2 \left\| \sum_{m=1}^M \boldsymbol{t}_m \right\|^2. \tag{21}$$

*Proof.*

$$\sum_{m \neq n} \|\boldsymbol{t}_m - \boldsymbol{t}_n\|^2$$

$$= \sum_{m \neq n} \|\boldsymbol{t}_m\|^2 + \sum_{m \neq n} \|\boldsymbol{t}_n\|^2 - \sum_{m \neq n} \langle \boldsymbol{t}_m, \boldsymbol{t}_n \rangle$$

$$= 2(M-1) \sum_{m=1}^M \|\boldsymbol{t}_m\|^2 - 2 \left( \sum_{m,n} \langle \boldsymbol{t}_m, \boldsymbol{t}_n \rangle - \sum_{m=1}^M \langle \boldsymbol{t}_m, \boldsymbol{t}_m \rangle \right)$$

$$= 2(M-1) \sum_{m=1}^{M} \|\boldsymbol{t}_m\|^2 - 2 \left( \left\langle \sum_{m=1}^{M} \boldsymbol{t}_m, \sum_{n=1}^{M} \boldsymbol{t}_n \right\rangle - \sum_{m=1}^{M} \|\boldsymbol{t}_m\|^2 \right)$$

$$= 2(M-1) \sum_{m=1}^{M} \|\boldsymbol{t}_m\|^2 - 2 \left\| \sum_{m=1}^{M} \boldsymbol{t}_m \right\|^2 + 2 \sum_{m=1}^{M} \|\boldsymbol{t}_m\|^2$$

$$= 2M \sum_{m=1}^{M} \|\boldsymbol{t}_m\|^2 - 2 \left\| \sum_{m=1}^{M} \boldsymbol{t}_m \right\|^2. \tag{22}$$

$\square$

## D. Corollary of the Assumption

**Corollary D.1.** *Let Assumption 4.1 and 4.6 hold. Then, in the case of positive correlation, i.e., $\langle \nabla f_i(\theta), \nabla f_j(\theta) \rangle \geq \varepsilon \|\nabla f_i(\theta)\| \cdot \|\nabla f_j(\theta)\|$, for any $\varepsilon \in (0,1)$ it follows that $\sigma_T^2 \leq (3-2\varepsilon) B^2$.*

*Proof.* We have

$$\begin{aligned}
\sigma_T^2 &= \|\nabla f_i(\theta) - \nabla f_j(\theta)\|^2 \\
&= \|\nabla f_i(\theta)\|^2 + \|\nabla f_j(\theta)\|^2 - 2 \langle \nabla f_i(\theta), f_j(\theta) \rangle \\
&\leq \|\nabla f_i(\theta)\|^2 + \|\nabla f_j(\theta)\|^2 - 2\varepsilon \|\nabla f_i(\theta)\| \cdot \|\nabla f_j(\theta)\| \\
&= (\|\nabla f_i(\theta)\| - \varepsilon \|\nabla f_j(\theta)\|)^2 + (1 - \varepsilon^2) \|\nabla f_j(\theta)\|^2 \\
&\leq \max\{B^2, 2(1-\varepsilon) B^2\}.
\end{aligned}$$

$\square$

**Corollary D.2** (BKT under partial participation). *Let $\theta_K^t$ denote the global model after round $t$ of task $K$, and set $\theta_K^0 = \theta_{K-1}$. In each round $\tau$, the server samples a subset $\mathcal{S}_K^\tau \subseteq [M]$ of size $N$ uniformly without replacement and aggregates only those clients. Suppose that for every $\tau \in [t]$, local epoch $e \in [E]$, and client $m \in [M]$, the gradients are $\epsilon$-aligned with the earlier-task gradient at the task-$K$ start, i.e., $\left\langle \nabla f_{1:K-1}(\theta_K^0), \nabla f_{K,m}(\theta_{K,m}^{\tau,e}) \right\rangle \geq \epsilon \|\nabla f_{1:K-1}(\theta_K^0)\| \cdot \left\| \nabla f_{K,m}(\theta_{K,m}^{\tau,e}) \right\|, \epsilon \in (0,1)$, and the local and global learning rates satisfy $\gamma_L \leq \frac{2\epsilon \|\nabla f_{1:K-1}(\theta_K^0)\|}{BLEt \sqrt{ME \cdot \left(\frac{1}{\lambda^2 + 2\lambda}\right)}}$ and $\gamma_G \leq \frac{1}{M(K-1)}$. Under full participation, for every $t \geq 1$, the server-updated model $\theta_K^t$ generated by $\mathrm{SPECIAL}$ (Algorithm 1) satisfies:*

$$\mathbb{E}_t \left[ f_{1:K-1}(\theta_K^t) \right] \leq f_{1:K-1}(\theta_K^0) + \frac{2\epsilon^2 \sigma_L^2 \|\nabla f_{1:K-1}(\theta_K^0)\|^2}{(K-1) tEM^2 LB^2}, \tag{23}$$

*where the expectation is over client sampling and data stochasticity.*

**Corollary D.3** ( Task-uniform convergence of $\mathrm{SPECIAL}$ under full participation). *Let the local learning rate $\gamma_L$ and global learning rate $\gamma_G$ satisfy $\gamma_G \leq \frac{1}{K-1}$, $\gamma_L \leq \frac{1}{8EL}$ and $\gamma_G \gamma_L \leq \frac{1+\lambda}{3EL}$. Under full participation, the sequence $\{\theta_K^t\}$ generated by $\mathrm{SPECIAL}$ satisfies:*

$$\min_{t \in [T]} \mathbb{E} \|\nabla f_{1:K}(\theta_K^t)\|^2 \leq \frac{f_{1:K}^0 - f_{1:K}^*}{\frac{1 - \frac{1}{K}}{2(1+\lambda)} E \gamma_G \gamma_L T} + \Psi', \tag{24}$$

*where*

$$\begin{aligned}
\Psi' =& \frac{2}{1 - \frac{1}{K}} \left[ \left( \frac{\gamma_L^2 E^2 L^2}{\lambda^2} + K \right) B^2 + 5\gamma_L^2 KEL^2 \left( \sigma_L^2 + 6E\sigma_G^2 \right) \right. \\
&\left. + \frac{3\gamma_G \gamma_L L \sigma_L^2}{2M(1+\lambda)} + \frac{(K-1)^2 E}{K} \cdot \frac{3\gamma_G \gamma_L L \sigma_T^2}{2(1+\lambda)} + \|\nabla f_{1:K-1}(\theta_K^0)\|^2 \right].
\end{aligned}$$

# E. Complete Proofs

### E.1. Proof of Theorem 4.7

*Proof.* According to the server update rule, we have:

$$
\begin{aligned}
\theta_i^{t+1} - \theta_i^t &= \frac{\bar{\theta}_i^{t+1} + \lambda \theta_i^0}{1 + \lambda} - \theta_i^t \\
&= \frac{\theta_i^t + \gamma_G \Delta_i^t + \lambda \theta_i^0 - (1 + \lambda) \theta_i^t}{1 + \lambda} \\
&= \frac{\gamma_G \Delta_i^t}{1 + \lambda} + \frac{\lambda \left( \theta_i^0 - \theta_i^t \right)}{1 + \lambda}.
\end{aligned}
\tag{25}
$$

Then, we can obtain:

$$
\begin{aligned}
\theta_i^{t+1} - \theta_i^0 &= \theta_i^{t+1} - \theta_i^t - \left( \theta_i^0 - \theta_i^t \right) \\
&= \frac{\gamma_G \Delta_i^t}{1 + \lambda} + \frac{\lambda \left( \theta_i^0 - \theta_i^t \right) - (1 + \lambda) \left( \theta_i^0 - \theta_i^t \right)}{1 + \lambda} \\
&= \frac{\gamma_G \Delta_i^t}{1 + \lambda} + \frac{\left( \theta_i^t - \theta_i^0 \right)}{1 + \lambda}.
\end{aligned}
\tag{26}
$$

Unrolling the recursion, we get:

$$
\theta_i^t - \theta_i^0 = \sum_{\tau=0}^{t-1} \left( \frac{1}{1 + \lambda} \right)^{\tau+1} \gamma_G \Delta_i^\tau.
\tag{27}
$$

Then we can expand $f_{1:K-1}\left( \theta_K^0 \right)$ by Lemma C.1, we have:

$$
\begin{aligned}
&\mathbb{E}_t \left[ f_{1:K-1} \left( \theta_K^t \right) \right] \leq f_{1:K-1} \left( \theta_K^0 \right) + \left\langle \nabla f_{1:K-1} \left( \theta_K^0 \right), \mathbb{E}_t \left[ \theta_K^t - \theta_K^0 \right] \right\rangle \\
&\quad + \frac{(K-1)}{2} L \mathbb{E}_t \left\| \theta_K^t - \theta_K^0 \right\|^2 \\
&\leq f_{1:K-1} \left( \theta_K^0 \right) - \gamma_G \gamma_L \left\langle \nabla f_{1:K-1} \left( \theta_K^0 \right), \mathbb{E}_t \left[ \frac{1}{N} \sum_{\tau=0}^{t-1} \sum_{m \in \mathcal{S}_K^\tau} \sum_{e=0}^{E-1} \left( \frac{1}{1+\lambda} \right)^{\tau+1} \left[ g_{K,m}^{\tau,e} \right] \right] \right\rangle \\
&\quad + \frac{(K-1)}{2} L \cdot \mathbb{E}_t \left\| \frac{-\gamma_G \gamma_L}{N} \sum_{\tau=0}^{t-1} \left( \frac{1}{1+\lambda} \right)^{\tau+1} \sum_{m \in \mathcal{S}_K^\tau} \sum_{e=0}^{E-1} g_{K,m}^{\tau,e} \right\|^2 \\
&\leq f_{1:K-1} \left( \theta_K^0 \right) \\
&\quad - \frac{\gamma_G \gamma_L}{N} \left\langle \nabla f_{1:K-1} \left( \theta_K^0 \right), \sum_{\tau=0}^{t-1} \sum_{m=1}^{M} \mathbb{P}\{m \in \mathcal{S}_K^\tau\} \sum_{e=0}^{E-1} \left( \frac{1}{1+\lambda} \right)^{\tau+1} \nabla f_{K,m} \left( \theta_{K,m}^{\tau,e} \right) \right\rangle \\
&\quad + \frac{(K-1)}{2} L \cdot \mathbb{E}_t \left\| \frac{-\gamma_G \gamma_L}{N} \sum_{\tau=0}^{t-1} \left( \frac{1}{1+\lambda} \right)^{\tau+1} \sum_{m=1}^{M} \mathbb{P}\{m \in \mathcal{S}_K^\tau\} \sum_{e=0}^{E-1} g_{K,m}^{\tau,e} \right\|^2 \\
&\leq f_{1:K-1} \left( \theta_K^0 \right) - \frac{\gamma_G \gamma_L}{M} \left\langle \nabla f_{1:K-1} \left( \theta_K^0 \right), \sum_{\tau=0}^{t-1} \sum_{m=1}^{M} \sum_{e=0}^{E-1} \left( \frac{1}{1+\lambda} \right)^{\tau+1} \nabla f_{K,m} \left( \theta_{K,m}^{\tau,e} \right) \right\rangle \\
&\quad + \frac{(K-1) \gamma_G^2 \gamma_L^2}{2} L \cdot \mathbb{E}_t \underbrace{\left\| \frac{-1}{N} \sum_{\tau=0}^{t-1} \left( \frac{1}{1+\lambda} \right)^{\tau+1} \sum_{m=1}^{M} \mathbb{P}\{m \in \mathcal{S}_K^\tau\} \sum_{e=0}^{E-1} g_{K,m}^{\tau,e} \right\|^2}_{T_1}.
\end{aligned}
\tag{28}
$$

Based on Assumption 4.3 and the Lemma C.7 with $G = -\frac{1}{N}, \boldsymbol{t}_m = \sum_{\tau=0}^{t-1} \left( \frac{1}{1+\lambda} \right)^{\tau+1} \sum_{e=0}^{E-1} g_{K,m}^{\tau,e}$, and $\Lambda = 0$, we can

bound term $T_1$ as:

$$
\begin{aligned}
&\mathbb{E}_t \left\| -\frac{1}{N} \sum_{\tau=0}^{t-1} \left( \frac{1}{1+\lambda} \right)^{\tau+1} \sum_{m=1}^{M} \mathbb{P}\{m \in \mathcal{S}_K^\tau\} \sum_{e=0}^{E-1} g_{K,m}^{\tau,e} \right\|^2 \\
&= \mathbb{E}_t \left\| \sum_{m=1}^{M} -\frac{1}{N} \mathbb{P}\{m \in \mathcal{S}_K^\tau\} \boldsymbol{t}_m \right\|^2 \\
&= -\frac{(N-1)}{2MN(M-1)} \sum_{m \neq n} \mathbb{E}_t \left\| \boldsymbol{t}_m - \boldsymbol{t}_n \right\|^2 + \frac{1}{M} \sum_{m=1}^{M} \mathbb{E}_t \left\| \boldsymbol{t}_m \right\|^2 \\
&\overset{(a_1)}{=} \frac{1}{M} \sum_{m=1}^{M} \mathbb{E}_t \left\| \boldsymbol{t}_m \right\|^2 - \frac{(N-1)}{2MN(M-1)} \left( 2M \sum_{m=1}^{M} \mathbb{E}_t \left\| \boldsymbol{t}_m \right\|^2 - 2\mathbb{E}_t \left\| \sum_{m=1}^{M} \boldsymbol{t}_m \right\|^2 \right) \\
&= \left( \frac{1}{M} - \frac{(N-1)}{N(M-1)} \right) \sum_{m=1}^{M} \mathbb{E}_t \left\| \boldsymbol{t}_m \right\|^2 + \frac{(N-1)}{MN(M-1)} \mathbb{E}_t \left\| \sum_{m=1}^{M} \boldsymbol{t}_m \right\|^2 \\
&\overset{(a_2)}{\leq} \left( \frac{M-N}{MN(M-1)} + \frac{N-1}{N(M-1)} \right) \sum_{m=1}^{M} \mathbb{E}_t \left\| \boldsymbol{t}_m \right\|^2 \\
&= \frac{1}{M} \sum_{m=1}^{M} \mathbb{E}_t \left\| \boldsymbol{t}_m \right\|^2,
\end{aligned}
\tag{29}
$$

where $(a_1)$ is due to Lemma C.8 and $(a_2)$ follows Lemma C.3. Substituting (29) back to (28), we have:

$$
\begin{aligned}
\mathbb{E}_t \left[ f_{1:K-1} \left( \theta_K^t \right) \right] &\leq f_{1:K-1} \left( \theta_K^0 \right) \\
&\quad - \frac{\gamma_G \gamma_L}{M} \left\langle \nabla f_{1:K-1} \left( \theta_K^0 \right), \sum_{\tau=0}^{t-1} \sum_{m=1}^{M} \sum_{e=0}^{E-1} \left( \frac{1}{1+\lambda} \right)^{\tau+1} \nabla f_{K,m} \left( \theta_{K,m}^{\tau,e} \right) \right\rangle \\
&\quad + \frac{(K-1)}{2} L \cdot \frac{\gamma_G^2 \gamma_L^2 Et}{M} \left[ \sum_{\tau=0}^{t-1} \sum_{m=1}^{M} \sum_{e=0}^{E-1} \mathbb{E}_t \left\| \left( \frac{1}{1+\lambda} \right)^{\tau+1} g_{K,m}^{\tau,e} \right\|^2 \right] \\
&\overset{(a_3)}{\leq} f_{1:K-1} \left( \theta_K^0 \right) - \frac{\gamma_G \gamma_L}{M} \cdot \epsilon \left\| \nabla f_{1:K-1} \left( \theta_K^0 \right) \right\| \cdot \left[ \sum_{\tau=0}^{t-1} \sum_{m=1}^{M} \sum_{e=0}^{E-1} \left\| \left( \frac{1}{1+\lambda} \right)^{\tau+1} \nabla f_{K,m} \left( \theta_{K,m}^{\tau,e} \right) \right\| \right] \\
&\quad + \underbrace{\frac{(K-1)}{2} L \cdot \frac{\gamma_G^2 \gamma_L^2 Et}{M} \left[ \sum_{\tau=0}^{t-1} \sum_{m=1}^{M} \sum_{e=0}^{E-1} \mathbb{E}_t \left\| \left( \frac{1}{1+\lambda} \right)^{\tau+1} g_{K,m}^{\tau,e} \right\|^2 \right]}_{T_2},
\end{aligned}
\tag{30}
$$

where $(a_3)$ is due to $\left\langle \nabla f_{1:K-1} \left( \theta_K^0 \right), \nabla f_{K,m} \left( \theta_{K,m}^{i,e} \right) \right\rangle \geq \epsilon \left\| \nabla f_{1:K-1} \left( \theta_K^0 \right) \right\| \cdot \left\| \nabla f_{K,m} \left( \theta_{K,m}^{i,e} \right) \right\|$.

Term $T_2$ in (30) can be bounded as

$$
\begin{aligned}
T_2 &\leq \underbrace{\frac{(K-1)}{2} L \cdot \frac{\gamma_G^2 \gamma_L^2 Et}{M} \left[ \sum_{\tau=0}^{t-1} \sum_{m=1}^{M} \sum_{e=0}^{E-1} \mathbb{E}_t \left\| \left( \frac{1}{1+\lambda} \right)^{\tau+1} \nabla f_{K,m} \left( \theta_{K,m}^{\tau,e} \right) \right\|^2 \right]}_{T_{21}} \\
&\quad + \underbrace{\frac{(K-1)}{2} L \cdot \frac{\gamma_G^2 \gamma_L^2 Et}{M} \left[ \sum_{\tau=0}^{t-1} \sum_{m=1}^{M} \sum_{e=0}^{E-1} \mathbb{E}_t \left\| \left( \frac{1}{1+\lambda} \right)^{\tau+1} \left( g_{K,m}^{\tau,e} - \nabla f_{K,m} \left( \theta_{K,m}^{\tau,e} \right) \right) \right\|^2 \right]}_{T_{22}}.
\end{aligned}
\tag{31}
$$

Since $\gamma_L \leq \frac{2\epsilon \left\| \nabla f_{1:K-1}\left(\theta_K^0\right)\right\|}{BLEt \sqrt{ME \cdot \left(\frac{1}{\lambda^2+2\lambda}\right)}}$ and $\gamma_G \leq \frac{1}{\sqrt{N}(K-1)}$, we have:

$$
\begin{aligned}
T_{21} =& \frac{\gamma_G \gamma_L}{M\sqrt{N}} \cdot (K-1)\gamma_G \cdot \frac{\gamma_L LEt}{2} \left[\sum_{\tau=0}^{t-1}\sum_{m=1}^{M}\sum_{e=0}^{E-1}\left\|\left(\frac{1}{1+\lambda}\right)^{\tau+1}\nabla f_{K,m}\left(\theta_{K,m}^{\tau,e}\right)\right\|^2\right] \\
\leq& \frac{\gamma_G \gamma_L}{M\sqrt{N}} \cdot \frac{\epsilon \left\|\nabla f_{1:K-1}\left(\theta_K^0\right)\right\|}{B\sqrt{ME \cdot \left(\frac{1}{\lambda^2+2\lambda}\right)}}\left[\sum_{\tau=0}^{t-1}\sum_{m=1}^{M}\sum_{e=0}^{E-1}\left\|\left(\frac{1}{1+\lambda}\right)^{\tau+1}\nabla f_{K,m}\left(\theta_{K,m}^{\tau,e}\right)\right\|^2\right] \\
\overset{(a_4)}{\leq}& \frac{\gamma_G \gamma_L}{M\sqrt{N}} \cdot \frac{\epsilon \left\|\nabla f_{1:K-1}\left(\theta_K^0\right)\right\| \cdot \left[\sum_{\tau=0}^{t-1}\sum_{m=1}^{M}\sum_{e=0}^{E-1}\left\|\left(\frac{1}{1+\lambda}\right)^{\tau+1}\nabla f_{K,m}\left(\theta_{K,m}^{\tau,e}\right)\right\|^2\right]}{\sqrt{\sum_{\tau=0}^{t-1}\left\|\left(\frac{1}{(1+\lambda)^2}\right)^{\tau+1}\right\| \cdot \sum_{m=1}^{M}\sum_{e=0}^{E-1}\left\|\nabla f_{K,m}\left(\theta_{K,m}^{\tau,e}\right)\right\|^2}} \\
\leq& \frac{\gamma_G \gamma_L}{M\sqrt{N}} \cdot \frac{\epsilon \left\|\nabla f_{1:K-1}\left(\theta_K^0\right)\right\| \cdot \left[\sum_{\tau=0}^{t-1}\sum_{m=1}^{M}\sum_{e=0}^{E-1}\left\|\left(\frac{1}{1+\lambda}\right)^{\tau+1}\nabla f_{K,m}\left(\theta_{K,m}^{\tau,e}\right)\right\|^2\right]}{\sqrt{\sum_{\tau=0}^{t-1}\sum_{m=1}^{M}\sum_{e=0}^{E-1}\left\|\left(\frac{1}{1+\lambda}\right)^{\tau+1}\nabla f_{K,m}\left(\theta_{K,m}^{\tau,e}\right)\right\|^2}} \\
\leq& \frac{\gamma_G \gamma_L}{M\sqrt{N}} \cdot \epsilon \left\|\nabla f_{1:K-1}\left(\theta_K^0\right)\right\| \sqrt{\sum_{\tau=0}^{t-1}\sum_{m=1}^{M}\sum_{e=0}^{E-1}\left\|\left(\frac{1}{1+\lambda}\right)^{\tau+1}\nabla f_{K,m}\left(\theta_{K,m}^{\tau,e}\right)\right\|^2} \\
\leq& \frac{\gamma_G \gamma_L}{M\sqrt{N}} \cdot \epsilon \left\|\nabla f_{1:K-1}\left(\theta_K^0\right)\right\| \left[\sum_{\tau=0}^{t-1}\sum_{m=1}^{M}\sum_{e=0}^{E-1}\left\|\left(\frac{1}{1+\lambda}\right)^{\tau+1}\nabla f_{K,m}\left(\theta_{K,m}^{\tau,e}\right)\right\|\right] \\
\leq& \frac{\gamma_G \gamma_L}{M} \cdot \epsilon \left\|\nabla f_{1:K-1}\left(\theta_K^0\right)\right\| \left[\sum_{\tau=0}^{t-1}\sum_{m=1}^{M}\sum_{e=0}^{E-1}\left\|\left(\frac{1}{1+\lambda}\right)^{\tau+1}\nabla f_{K,m}\left(\theta_{K,m}^{\tau,e}\right)\right\|\right],
\end{aligned}
\tag{32}
$$

where $(a_4)$ follows that

$$
\frac{1}{\lambda^2+2\lambda} = \frac{1}{(1+\lambda)^2} \cdot \frac{1}{1-\frac{1}{(1+\lambda)^2}} = \sum_{\tau=0}^{\infty}\left(\frac{1}{(1+\lambda)^2}\right)^{\tau+1} \geq \sum_{\tau=0}^{t-1}\left(\frac{1}{(1+\lambda)^2}\right)^{\tau+1}.
\tag{33}
$$

Term $T_{22}$ in (31) can be bounded as

$$
\begin{aligned}
T_{22} =& \frac{(K-1)}{2} L \cdot \frac{\gamma_G^2 \gamma_L^2 Et}{M} \left[\sum_{\tau=0}^{t-1}\sum_{m=1}^{M}\sum_{e=0}^{E-1}\left(\frac{1}{(1+\lambda)^2}\right)^{\tau+1}\mathbb{E}_t \left\|g_{K,m}^{\tau,e} - \nabla f_{K,m}\left(\theta_{K,m}^{\tau,e}\right)\right\|^2\right] \\
\overset{(a_5)}{\leq}& \frac{(K-1)}{2} L \cdot \frac{\gamma_G^2 \gamma_L^2 Et}{M} \left[\sum_{\tau=0}^{t-1}\left(\frac{1}{(1+\lambda)^2}\right)^{\tau+1}\sum_{m=1}^{M}\sum_{e=0}^{E-1}\sigma_L^2\right] \\
\leq& \frac{(K-1)}{2} L \cdot \gamma_G^2 \gamma_L^2 E^2 \sigma_L^2 t \cdot \sum_{\tau=0}^{t-1}\left(\frac{1}{(1+\lambda)^2}\right)^{\tau+1} \\
\overset{(a_6)}{\leq}& \frac{(K-1)}{2} L \cdot \frac{\gamma_G^2 \gamma_L^2 E^2 \sigma_L^2 t}{\lambda^2+2\lambda} \\
\overset{(a_7)}{\leq}& \frac{2\epsilon^2 \sigma_L^2 \left\|\nabla f_{1:K-1}\left(\theta_K^0\right)\right\|^2}{(K-1)tEMNLB^2},
\end{aligned}
\tag{34}
$$

where $(a_5)$ is due to Assumption 4.4, $(a_6)$ follows from (33), and $(a_7)$ holds since $\gamma_L \leq \frac{2\epsilon \left\| \nabla f_{1:K-1}\left(\theta_K^0\right) \right\|}{BLEt\sqrt{ME \cdot \left(\frac{1}{\lambda^2 + 2\lambda}\right)}}$ and

$\gamma_G \leq \frac{1}{\sqrt{N}(K-1)}$.

Plugging (32) and (34) into (34), we have:

$$
\begin{aligned}
T_2 \leq & \frac{\gamma_G \gamma_L}{M} \cdot \epsilon \left\| \nabla f_{1:K-1}\left(\theta_K^0\right) \right\| \left[ \sum_{\tau=0}^{t-1} \sum_{m=1}^{M} \sum_{e=0}^{E-1} \left\| \left(\frac{1}{1+\lambda}\right)^{\tau+1} \nabla f_{K,m}\left(\theta_{K,m}^{\tau,e}\right) \right\| \right] \\
& + \frac{2\epsilon^2 \sigma_L^2 \left\| \nabla f_{1:K-1}\left(\theta_K^0\right) \right\|^2}{(K-1)\, tEMNLB^2}.
\end{aligned}
\tag{35}
$$

Plugging (35) back into (30), we have:

$$
\mathbb{E}_t \left[ f_{1:K-1}\left(\theta_K^t\right) \right] \leq f_{1:K-1}\left(\theta_K^0\right) + \frac{2\epsilon^2 \sigma_L^2 \left\| \nabla f_{1:K-1}\left(\theta_K^0\right) \right\|^2}{(K-1)\, tEMNLB^2}.
\tag{36}
$$

$\square$

### E.2. Proof of Lemma 4.9

*Proof.* According to Eq. 27 and Lemma C.2, we have:

$$
\begin{aligned}
\mathbb{E}_t \left\| \theta_i^t - \theta_i^0 \right\| =& \mathbb{E}_t \left\| \sum_{\tau=0}^{t-1} \left(\frac{1}{1+\lambda}\right)^{\tau+1} \gamma_G \Delta_i^\tau \right\| \\
\leq & \sum_{\tau=0}^{t-1} \left(\frac{1}{1+\lambda}\right)^{\tau+1} \mathbb{E}_t \left\| \frac{-\gamma_G \gamma_L}{N} \sum_{m \in \mathcal{S}_i^\tau} \sum_{e=0}^{E-1} \nabla f_{i,m}\left(\theta_{i,m}^{\tau,e}, \xi_{i,m}^{\tau,e}\right) \right\| \\
\leq & \left[ \sum_{\tau=0}^{t-1} \left(\frac{1}{1+\lambda}\right)^{\tau+1} \right] \cdot \mathbb{E}_t \left[ \frac{\gamma_G \gamma_L}{N} \sum_{m \in \mathcal{S}_i^\tau} \sum_{e=0}^{E-1} \left\| \nabla f_{i,m}\left(\theta_{i,m}^{\tau,e}, \xi_{i,m}^{\tau,e}\right) \right\| \right] \\
= & \left[ \sum_{\tau=0}^{t-1} \left(\frac{1}{1+\lambda}\right)^{\tau+1} \right] \cdot \frac{\gamma_G \gamma_L}{N} \sum_{m=1}^{M} \mathbb{P}\{m \in \mathcal{S}_i^\tau\} \left\| \nabla f_{i,m}\left(\theta_{i,m}^{\tau,e}, \xi_{i,m}^{\tau,e}\right) \right\| \\
\overset{(b_1)}{\leq} & \frac{1}{\lambda} \cdot \gamma_G \gamma_L EB,
\end{aligned}
\tag{37}
$$

where $(b_1)$ follows the truth that $\mathbb{P}\{m \in \mathcal{S}_i^\tau\} = \frac{N}{M}$ Assumption 4.1 and

$$
\sum_{\tau=0}^{t-1} \left(\frac{1}{1+\lambda}\right)^{\tau+1} \leq \sum_{\tau=0}^{\infty} \left(\frac{1}{1+\lambda}\right)^{\tau+1} = \left(\frac{1}{1+\lambda}\right) \cdot \frac{1}{1 - \frac{1}{1+\lambda}} = \frac{1}{\lambda}.
$$

Then, we can bound $\left\| \theta_i^t - \theta_i^0 \right\|^2$ as following:

$$
\mathbb{E}_t \left\| \theta_i^t - \theta_i^0 \right\|^2 \leq \frac{\gamma_G^2 \gamma_L^2 E^2 B^2}{\lambda^2}.
\tag{38}
$$

$\square$

### E.3. Proof of Theorem 4.10

*Proof.* According to Lemma C.1, $f_{1:K}$ is $KL$-smooth, and we have the following expansion by taking expectation over the randomness during the training process:

$$
\mathbb{E}_t \left[ f_{1:K}\left(\theta_K^{t+1}\right) \right] \leq f_{1:K}\left(\theta_K^t\right) + \left\langle \nabla f_{1:K}\left(\theta_K^t\right), \mathbb{E}_t \left[\theta_K^{t+1} - \theta_K^t\right] \right\rangle + \frac{KL}{2} \mathbb{E}_t \left\| \theta_K^{t+1} - \theta_K^t \right\|^2
$$

$$
= f_{1:K}\left(\theta_K^t\right) + \left\langle \nabla f_{1:K}\left(\theta_K^t\right), \mathbb{E}_t\left[\theta_K^{t+1} - \theta_K^t\right] - \frac{\gamma_G \gamma_L E}{1+\lambda} \nabla f_K\left(\theta_K^t\right) + \frac{\gamma_G \gamma_L E}{1+\lambda} \nabla f_K\left(\theta_K^t\right) \right\rangle
$$

$$
+ \frac{KL}{2} \mathbb{E}_t \left\| \theta_K^{t+1} - \theta_K^t \right\|^2
$$

$$
\overset{(c_1)}{=} f_{1:K}\left(\theta_K^t\right) - \frac{\gamma_G \gamma_L E}{1+\lambda} \left\| \nabla f_K\left(\theta_K^t\right) \right\|^2 - \frac{\gamma_G \gamma_L E}{1+\lambda} \underbrace{\left\langle \nabla f_{1:K-1}\left(\theta_K^t\right), \nabla f_K\left(\theta_K^t\right) \right\rangle}_{T_4}
$$

$$
+ \underbrace{\left\langle \nabla f_{1:K}\left(\theta_K^t\right), \mathbb{E}_t\left[\theta_K^{t+1} - \theta_K^t\right] + \frac{\gamma_G \gamma_L E}{1+\lambda} \nabla f_K\left(\theta_K^t\right) \right\rangle}_{T_5} + \frac{KL}{2} \underbrace{\mathbb{E}_t \left\| \theta_K^{t+1} - \theta_K^t \right\|^2}_{T_6}, \tag{39}
$$

where $(c_1)$ holds due to $f_{1:K}\left(\theta_K^t\right) = f_{1:K-1}\left(\theta_K^t\right) + f_K\left(\theta_K^t\right)$.

Then, term $T_4$ in (39) can be expanded as:

$$
T_4 \overset{(c_2)}{=} \frac{1}{2} \left[ \left\| \nabla f_{1:K}\left(\theta_K^t\right) \right\|^2 - \left\| \nabla f_{1:K-1}\left(\theta_K^t\right) \right\|^2 - \left\| \nabla f_K\left(\theta_K^t\right) \right\|^2 \right]
$$

$$
= \frac{1}{2} \left[ \left\| \nabla f_{1:K}\left(\theta_K^t\right) \right\|^2 - \left\| \nabla f_{1:K-1}\left(\theta_K^t\right) - \nabla f_{1:K-1}\left(\theta_K^0\right) + \nabla f_{1:K-1}\left(\theta_K^0\right) \right\|^2 \right.
$$

$$
\left. - \left\| \nabla f_K\left(\theta_K^t\right) \right\|^2 \right]
$$

$$
\geq \frac{1}{2} \left[ \left\| \nabla f_{1:K}\left(\theta_K^t\right) \right\|^2 - 2 \left\| \nabla f_{1:K-1}\left(\theta_K^t\right) - \nabla f_{1:K-1}\left(\theta_K^0\right) \right\|^2 - 2 \left\| \nabla f_{1:K-1}\left(\theta_K^0\right) \right\|^2 \right. \tag{40}
$$

$$
\left. - \left\| \nabla f_K\left(\theta_K^t\right) \right\|^2 \right]
$$

$$
\overset{(c_3)}{\geq} \frac{1}{2} \left\| \nabla f_{1:K}\left(\theta_K^t\right) \right\|^2 - \frac{1}{2} \left\| \nabla f_K\left(\theta_K^t\right) \right\|^2 - L^2 (K-1)^2 \left\| \theta_K^t - \theta_K^0 \right\|^2 - \left\| \nabla f_{1:K-1}\left(\theta_K^0\right) \right\|^2
$$

$$
\overset{(c_4)}{\geq} \frac{1}{2} \left\| \nabla f_{1:K}\left(\theta_K^t\right) \right\|^2 - \frac{1}{2} \left\| \nabla f_K\left(\theta_K^t\right) \right\|^2 - \frac{L^2 (K-1)^2 \gamma_G^2 \gamma_L^2 E^2 B^2}{\lambda^2} - \left\| \nabla f_{1:K-1}\left(\theta_K^0\right) \right\|^2,
$$

where $(c_2)$ holds because $\langle x, y \rangle = \frac{1}{2}\left[ \|x+y\|^2 - \|x\|^2 - \|y\|^2 \right]$, $(c_3)$ is due to the fact that $f_{1:K-1}\left(\theta_K^t\right)$ is $(L-1)$-smoothness and Assumption 4.2, and $(c_4)$ follows Lemma 4.9.

Since we can expand the right side of term $T_5$ as:

$$
\mathbb{E}_t\left[\theta_K^{t+1} - \theta_K^t\right] + \frac{\gamma_G \gamma_L E}{1+\lambda} \nabla f_K\left(\theta_K^t\right)
$$

$$
= \left(\frac{1}{1+\lambda}\right) \mathbb{E}_t \left[ \frac{-\gamma_G \gamma_L}{N} \sum_{m \in \mathcal{S}_K^t} \sum_{e=0}^{E-1} g_{K,m}^{t,e} + \lambda\left(\theta_K^0 - \theta_K^t\right) + \gamma_G \gamma_L E \nabla f_K\left(\theta_K^t\right) \right]
$$

$$
= \left(\frac{1}{1+\lambda}\right) \mathbb{E}_t \left[ \frac{-\gamma_G \gamma_L}{N} \cdot \frac{N}{M} \sum_{m=1}^{M} \sum_{e=0}^{E-1} \left( \nabla f_{K,m}\left(\theta_{K,m}^{t,e}\right) - \nabla f_{K,m}\left(\theta_{K,m}^{t,0}\right) \right) + \lambda\left(\theta_K^0 - \theta_K^t\right) \right]
$$

$$
= \left(\frac{1}{1+\lambda}\right) \mathbb{E}_t \left[ \frac{-\gamma_G \gamma_L}{M} \sum_{m=1}^{M} \sum_{e=0}^{E-1} \left( \nabla f_{K,m}\left(\theta_{K,m}^{t,e}\right) - \nabla f_{K,m}\left(\theta_{K,m}^{t,0}\right) \right) + \lambda\left(\theta_K^0 - \theta_K^t\right) \right]. \tag{41}
$$

Then, we can further expand $T_5$ as:

$$
T_5 = \left(\frac{1}{1+\lambda}\right) \left\langle \nabla f_{1:K}\left(\theta_K^t\right), \right.
$$

$$
\mathbb{E}_t \left[ \frac{-\gamma_G \gamma_L}{M} \sum_{m=1}^{M} \sum_{e=0}^{E-1} \left( \nabla f_{K,m}\left(\theta_{K,m}^{t,e}\right) - \nabla f_{K,m}\left(\theta_{K,m}^{t,0}\right) \right) + \lambda\left(\theta_K^0 - \theta_K^t\right) \right] \right\rangle
$$

$$
= \left(\frac{1}{1+\lambda}\right) \left\langle \sqrt{\frac{\gamma_G \gamma_L E}{K}} \nabla f_{1:K}\left(\theta_K^t\right), \right.
$$

$$\sqrt{\frac{K}{\gamma_G \gamma_L E}} \mathbb{E}_t \left[ \frac{-\gamma_G \gamma_L}{M} \sum_{m=1}^{M} \sum_{e=0}^{E-1} \left( \nabla f_{K,m} \left( \theta_{K,m}^{t,e} \right) - \nabla f_{K,m} \left( \theta_{K,m}^{t,0} \right) \right) + \lambda \left( \theta_K^0 - \theta_K^t \right) \right] \bigg\rangle$$

$$= \frac{1}{2(1+\lambda)} \left[ \frac{\gamma_G \gamma_L E}{K} \left\| \nabla f_{1:K} \left( \theta_K^t \right) \right\|^2 \right.$$

$$+ \frac{K}{\gamma_G \gamma_L E} \mathbb{E}_t \left\| \frac{-\gamma_G \gamma_L}{M} \sum_{m=1}^{M} \sum_{e=0}^{E-1} \left( \nabla f_{K,m} \left( \theta_{K,m}^{t,e} \right) - \nabla f_{K,m} \left( \theta_{K,m}^{t,0} \right) \right) + \lambda \left( \theta_K^0 - \theta_K^t \right) \right\|^2$$

$$- \mathbb{E}_t \left\| \sqrt{\frac{K}{\gamma_G \gamma_L E}} \left[ \frac{-\gamma_G \gamma_L}{M} \sum_{m=1}^{M} \sum_{e=0}^{E-1} \left( \nabla f_{K,m} \left( \theta_{K,m}^{t,e} \right) - \nabla f_{K,m} \left( \theta_{K,m}^{t,0} \right) \right) + \lambda \left( \theta_K^0 - \theta_K^t \right) \right] \right.$$

$$\left. \left. - \sqrt{\frac{\gamma_G \gamma_L E}{K}} \nabla f_{1:K} \left( \theta_K^t \right) \right\|^2 \right]$$

$$= \frac{1}{2(1+\lambda)} \left[ \frac{\gamma_G \gamma_L E}{K} \left\| \nabla f_{1:K} \left( \theta_K^t \right) \right\|^2 + T_{51} - T_{52} \right]. \tag{42}$$

Term $T_{51}$ in (42) can be bounded as:

$$T_{51} \leq 2 \mathbb{E}_t \left\| \frac{-\gamma_G \gamma_L}{M} \sum_{m=1}^{M} \sum_{e=0}^{E-1} \left( \nabla f_{K,m} \left( \theta_{K,m}^{t,e} \right) - \nabla f_{K,m} \left( \theta_{K,m}^{t,0} \right) \right) \right\|^2 + 2 \left\| \lambda \left( \theta_K^0 - \theta_K^t \right) \right\|^2$$

$$\overset{(c_5)}{\leq} \frac{2\gamma_G^2 \gamma_L^2 E}{M} \sum_{m=1}^{M} \sum_{e=0}^{E-1} \mathbb{E}_t \left\| \nabla f_{K,m} \left( \theta_{K,m}^{t,e} \right) - \nabla f_{K,m} \left( \theta_{K,m}^{t,0} \right) \right\|^2 + 2\lambda^2 \left\| \theta_K^0 - \theta_K^t \right\|^2$$

$$\overset{(c_6)}{\leq} \frac{2\gamma_G^2 \gamma_L^2 E L^2}{M} \sum_{m=1}^{M} \sum_{e=0}^{E-1} \mathbb{E}_t \left\| \theta_{K,m}^{t,e} - \theta_{K,m}^{t,0} \right\|^2 + 2\gamma_G^2 \gamma_L^2 E^2 B^2 \tag{43}$$

$$\overset{(c_7)}{\leq} 2\gamma_G^2 \gamma_L^2 E^2 L^2 \cdot \left( 5\gamma_L^2 E \left( \sigma_L^2 + 6E\sigma_G^2 \right) + 30\gamma_L^2 E^2 \left\| \nabla f_K \left( \theta_K^t \right) \right\|^2 \right) + 2\gamma_G^2 \gamma_L^2 E^2 B^2$$

$$= 10\gamma_G^2 \gamma_L^4 E^3 L^2 \sigma_L^2 + 60\gamma_G^2 \gamma_L^4 E^4 L^2 \left( \sigma_G^2 + \left\| \nabla f_K \left( \theta_K^t \right) \right\|^2 \right) + 2\gamma_G^2 \gamma_L^2 E^2 B^2,$$

where $(c_5)$ is due to Lemma C.3, $(c_6)$ holds due to Assumption 4.2 and Lemma 4.9, and $(c_7)$ follows Lemma C.5. Term $T_{52}$ in (42) can be expanded as:

$$T_{52} = \frac{1}{\gamma_G \gamma_L E K} \mathbb{E}_t \left\| \frac{-\gamma_G \gamma_L K}{M} \sum_{m=1}^{M} \sum_{e=0}^{E-1} \left( \nabla f_{K,m} \left( \theta_{K,m}^{t,e} \right) - \nabla f_{K,m} \left( \theta_{K,m}^{t,0} \right) \right) \right.$$

$$\left. + \lambda K \left( \theta_K^0 - \theta_K^t \right) - \gamma_G \gamma_L E \nabla f_{1:K} \left( \theta_K^t \right) \right\|^2$$

$$= \frac{\gamma_G \gamma_L}{E K} \mathbb{E}_t \left\| -\frac{1}{M} \sum_{i=1}^{K} \sum_{m=1}^{M} \sum_{e=0}^{E-1} \left( \nabla f_{K,m} \left( \theta_{K,m}^{t,e} \right) - \nabla f_{K,m} \left( \theta_{K,m}^{t,0} \right) \right) \right.$$

$$\left. + \frac{\lambda K}{\gamma_G \gamma_L} \left( \theta_K^0 - \theta_K^t \right) - E \sum_{i=1}^{K} f_i \left( \theta_K^t \right) \right\|^2$$

$$= \frac{\gamma_G \gamma_L}{E K} \mathbb{E}_t \left\| -\frac{1}{M} \sum_{i=1}^{K} \sum_{m=1}^{M} \sum_{e=0}^{E-1} \left( \nabla f_{K,m} \left( \theta_{K,m}^{t,e} \right) - \nabla f_K \left( \theta_K^t \right) + \nabla f_i \left( \theta_K^t \right) \right) \right.$$

$$\left. + \frac{\lambda K}{\gamma_G \gamma_L} \left( \theta_K^0 - \theta_K^t \right) \right\|^2. \tag{44}$$

Substituting terms $T_{51}$ and $T_{52}$ into (42), we have:

$$T_5 \leq \frac{\gamma_G \gamma_L E}{2K(1+\lambda)} \left\| \nabla f_{1:K} \left( \theta_K^t \right) \right\|^2 + \frac{5\gamma_G \gamma_L^3 K E^2 L^2}{(1+\lambda)} \left( \left( \sigma_L^2 + 6E\sigma_G^2 \right) + 6E \left\| \nabla f_K \left( \theta_K^t \right) \right\|^2 \right)$$

$$+ \frac{\gamma_G \gamma_L K E B^2}{1 + \lambda} - \frac{\gamma_G \gamma_L}{2 (1 + \lambda) E K}$$

$$\cdot \mathbb{E}_t \left\| -\frac{1}{M} \sum_{i=1}^{K} \sum_{m=1}^{M} \sum_{e=0}^{E-1} \left( \nabla f_{K,m} \left( \theta_{K,m}^{t,e} \right) - \nabla f_K \left( \theta_K^t \right) + \nabla f_i \left( \theta_K^t \right) \right) + \frac{\lambda K}{\gamma_G \gamma_L} \left( \theta_K^0 - \theta_K^t \right) \right\|^2. \tag{45}$$

Term $T_6$ in (39) can be expanded as:

$$T_6 = \mathbb{E}_t \left[ \left\| \left( \frac{-1}{1 + \lambda} \right) \left( \frac{\gamma_G \gamma_L}{N} \sum_{m \in \mathcal{S}_K^t} \sum_{e=0}^{E-1} g_{K,m}^{t,e} + \lambda \left( \theta_K^t - \theta_K^0 \right) \right) \right\|^2 \right]$$

$$= \frac{1}{(1 + \lambda)^2} \mathbb{E}_t \left\| \frac{1}{K} \left( \frac{-\gamma_G \gamma_L}{N} \sum_{i=1}^{K} \sum_{m \in \mathcal{S}_K^t} \sum_{e=0}^{E-1} g_{K,m}^{t,e} + \lambda K \left( \theta_K^0 - \theta_K^t \right) \right) \right\|^2$$

$$= \frac{1}{K^2 (1 + \lambda)^2} \mathbb{E}_t \left\| \frac{-\gamma_G \gamma_L}{N} \sum_{i=1}^{K} \sum_{m \in \mathcal{S}_K^t} \sum_{e=0}^{E-1} g_{K,m}^{t,e} + \lambda K \left( \theta_K^0 - \theta_K^t \right) \right\|^2$$

$$= \frac{1}{K^2 (1 + \lambda)^2} \mathbb{E}_t \left\| \frac{-\gamma_G \gamma_L}{N} \sum_{i=1}^{K} \sum_{m \in \mathcal{S}_K^t} \sum_{e=0}^{E-1} \left( g_{K,m}^{t,e} - \nabla f_{K,m} \left( \theta_{K,m}^{t,e} \right) \right. \right.$$

$$\left. \left. + \nabla f_{K,m} \left( \theta_{K,m}^{t,e} \right) - \nabla f_K \left( \theta_K^t \right) + \nabla f_K \left( \theta_K^t \right) - \nabla f_i \left( \theta_K^t \right) + \nabla f_i \left( \theta_K^t \right) \right) + \lambda K \left( \theta_K^0 - \theta_K^t \right) \right\|^2. \tag{46}$$

Then, term $T_6$ can be bounded by three terms as:

$$T_6 \leq \frac{3}{K^2 (1 + \lambda)^2} \left( \underbrace{\mathbb{E}_t \left\| -\frac{\gamma_G \gamma_L}{N} \sum_{i=1}^{K} \sum_{m \in \mathcal{S}_K^t} \sum_{e=0}^{E-1} \left( g_{K,m}^{t,e} - \nabla f_{K,m} \left( \theta_{K,m}^{t,e} \right) \right) \right\|^2}_{T_{61}} + \gamma_G^2 \gamma_L^2 \right.$$

$$\cdot \underbrace{\mathbb{E}_t \left\| -\frac{1}{N} \sum_{i=1}^{K} \sum_{m \in \mathcal{S}_K^t} \sum_{e=0}^{E-1} \left( \nabla f_{K,m} \left( \theta_{K,m}^{t,e} \right) - \nabla f_K \left( \theta_K^t \right) + \nabla f_i \left( \theta_K^t \right) \right) + \frac{\lambda K}{\gamma_G \gamma_L} \left( \theta_K^0 - \theta_K^t \right) \right\|^2}_{T_{62}}$$

$$+ \underbrace{\mathbb{E}_t \left\| -\frac{\gamma_G \gamma_L}{N} \sum_{i=1}^{K} \sum_{m \in \mathcal{S}_K^t} \sum_{e=0}^{E-1} \left( \nabla f_K \left( \theta_K^t \right) - \nabla f_i \left( \theta_K^t \right) \right) \right\|^2}_{T_{63}} \right). \tag{47}$$

Term $T_{61}$ in (47) can be bounded as:

$$T_{61} = \mathbb{E}_t \left\| -\frac{\gamma_G \gamma_L}{N} \sum_{i=1}^{K} \sum_{m=1}^{M} \mathbb{P}\{m \in \mathcal{S}_K^t\} \sum_{e=0}^{E-1} \left( g_{K,m}^{t,e} - \nabla f_{K,m} \left( \theta_{K,m}^{t,e} \right) \right) \right\|^2 \leq \frac{\gamma_G^2 \gamma_L^2 E K \sigma_L^2}{N}. \tag{48}$$

Term $T_{62}$ in (47) can be expanded as:

$$T_{62} = \mathbb{E}_t \left\| -\frac{1}{N} \sum_{i=1}^{K} \sum_{m=1}^{M} \mathbb{P}\{m \in \mathcal{S}_K^t\} \sum_{e=0}^{E-1} \left( \nabla f_{K,m} \left( \theta_{K,m}^{t,e} \right) - \nabla f_K \left( \theta_K^t \right) + \nabla f_i \left( \theta_K^t \right) \right) \right.$$

$$\left. + \frac{\lambda K}{\gamma_G \gamma_L} \left( \theta_K^0 - \theta_K^t \right) \right\|^2. \tag{49}$$

Term $T_{63}$ in (47) can be expanded as:

$$
\begin{aligned}
T_{63} &= \mathbb{E}_t \left\| -\frac{\gamma_G \gamma_L}{N} \sum_{i=1}^{K} \sum_{m=1}^{M} \mathbb{P}\{m \in \mathcal{S}_K^t\} \sum_{e=0}^{E-1} \left( \nabla f_K \left( \theta_K^t \right) - \nabla f_i \left( \theta_K^t \right) \right) \right\|^2 \\
&= \mathbb{E}_t \left\| -\frac{\gamma_G \gamma_L}{M} \sum_{i=1}^{K} \sum_{m=1}^{M} \sum_{e=0}^{E-1} \left( \nabla f_K \left( \theta_K^t \right) - \nabla f_i \left( \theta_K^t \right) \right) \right\|^2 \\
&= \mathbb{E}_t \left\| -\frac{\gamma_G \gamma_L}{M} \sum_{i=1}^{K-1} \sum_{m=1}^{M} \sum_{e=0}^{E-1} \left( \nabla f_K \left( \theta_K^t \right) - \nabla f_i \left( \theta_K^t \right) \right) \right\|^2 \\
&\leq \gamma_G^2 \gamma_L^2 \left( K - 1 \right)^2 E^2 \sigma_T^2.
\end{aligned}
\tag{50}
$$

Substituting (48), (49), and (50) back into (47), we have:

$$
\begin{aligned}
T_6 &\leq \frac{3\gamma_G^2 \gamma_L^2 E \sigma_L^2}{NK \left( 1 + \lambda \right)^2} + \frac{3\gamma_G^2 \gamma_L^2 \left( K - 1 \right)^2 E^2 \sigma_T^2}{K^2 \left( 1 + \lambda \right)^2} + \frac{3\gamma_G^2 \gamma_L^2}{K^2 \left( 1 + \lambda \right)^2} \\
&\quad \cdot \mathbb{E}_t \left\| -\frac{1}{N} \sum_{i=1}^{K} \sum_{m=1}^{M} \mathbb{P}\{m \in \mathcal{S}_K^t\} \sum_{e=0}^{E-1} \left( \nabla f_{K,m} \left( \theta_{K,m}^{t,e} \right) - \nabla f_K \left( \theta_K^t \right) + \nabla f_i \left( \theta_K^t \right) \right) \right. \\
&\quad \left. + \frac{\lambda K}{\gamma_G \gamma_L} \left( \theta_K^0 - \theta_K^t \right) \right\|^2.
\end{aligned}
\tag{51}
$$

Substituting (40), (45), and (51) back into (39), we have:

$$
\begin{aligned}
\mathbb{E}_t \left[ f_{1:K} \left( \theta_K^{t+1} \right) \right] &\leq f_{1:K} \left( \theta_K^t \right) - \frac{\gamma_G \gamma_L E}{1 + \lambda} \left( \frac{1}{2} - 30 K \gamma_L^2 L^2 E^2 \right) \left\| \nabla f_K \left( \theta_K^t \right) \right\|^2 \\
&\quad - \frac{1}{2} \left( \frac{\gamma_G \gamma_L E}{1 + \lambda} \right) \left( 1 - \frac{1}{K} \right) \left\| \nabla f_{1:K} \left( \theta_K^t \right) \right\|^2 + \frac{\gamma_G \gamma_L E}{1 + \lambda} \left( \frac{L^2 \left( K - 1 \right)^2 \gamma_G^2 \gamma_L^2 E^2}{\lambda^2} + K \right) B^2 \\
&\quad + \frac{\gamma_G \gamma_L E}{1 + \lambda} \left( 5 \gamma_L^2 K E L^2 \left( \sigma_L^2 + 6 E \sigma_G^2 \right) + \frac{3 \gamma_G \gamma_L L}{2 \left( 1 + \lambda \right)} \left( \frac{\sigma_L^2}{N} + \frac{\left( K - 1 \right)^2 E}{K} \sigma_T^2 \right) \right) \\
&\quad + \frac{\gamma_G \gamma_L E}{1 + \lambda} \left\| \nabla f_{1:K-1} \left( \theta_K^0 \right) \right\|^2 + \frac{3 \gamma_G^2 \gamma_L^2 L}{2K \left( 1 + \lambda \right)^2} \\
&\quad \cdot \underbrace{\mathbb{E}_t \left\| -\frac{1}{N} \sum_{i=1}^{K} \sum_{m=1}^{M} \mathbb{P}\{m \in \mathcal{S}_K^t\} \sum_{e=0}^{E-1} \left( \nabla f_{K,m} \left( \theta_{K,m}^{t,e} \right) - \nabla f_K \left( \theta_K^t \right) + \nabla f_i \left( \theta_K^t \right) \right) + \frac{\lambda K}{\gamma_G \gamma_L} \left( \theta_K^0 - \theta_K^t \right) \right\|^2}_{T_7} \\
&\quad - \frac{\gamma_G \gamma_L}{2 \left( 1 + \lambda \right) E K} \\
&\quad \cdot \underbrace{\mathbb{E}_t \left\| -\frac{1}{M} \sum_{i=1}^{K} \sum_{m=1}^{M} \sum_{e=0}^{E-1} \left( \nabla f_{K,m} \left( \theta_{K,m}^{t,e} \right) - \nabla f_K \left( \theta_K^t \right) + \nabla f_i \left( \theta_K^t \right) \right) + \frac{\lambda K}{\gamma_G \gamma_L} \left( \theta_K^0 - \theta_K^t \right) \right\|^2}_{T_8}.
\end{aligned}
\tag{52}
$$

According to Lemma C.7 with $\boldsymbol{t}_m = \sum_{i=1}^{K} \sum_{e=0}^{E-1} \left( \nabla f_{K,m} \left( \theta_{K,m}^{t,e} \right) - \nabla f_K \left( \theta_K^t \right) + \nabla f_i \left( \theta_K^t \right) \right)$, $G = -\frac{1}{N}$, and $\Lambda = \frac{\lambda K}{\gamma_G \gamma_L} \left( \theta_K^0 - \theta_K^t \right)$, we can expand term $T_7$ as:

$$
T_7 = \left( \frac{N^2}{MN^2} - \frac{N}{MN} \right) \sum_{m=1}^{M} \mathbb{E}_t \left\| \boldsymbol{t}_m \right\|^2 - \frac{N \left( N - 1 \right)}{2 M N^2 \left( M - 1 \right)} \sum_{m \neq n} \mathbb{E}_t \left\| \boldsymbol{t}_m - \boldsymbol{t}_n \right\|^2
$$

$$+ \left(1 - \frac{N}{N}\right) \mathbb{E}_t \|\Lambda\|^2 + \frac{N}{MN} \sum_{m=1}^{M} \mathbb{E}_t \|\boldsymbol{t}_m - \Lambda\|^2$$

$$= - \frac{(N-1)}{2MN(M-1)} \sum_{m \neq n} \mathbb{E}_t \|\boldsymbol{t}_m - \boldsymbol{t}_n\|^2 + \frac{1}{M} \sum_{m=1}^{M} \mathbb{E}_t \|\boldsymbol{t}_m - \Lambda\|^2 . \tag{53}$$

According to Lemma C.6 with $\boldsymbol{t}_m = \sum_{i=1}^{K} \sum_{e=0}^{E-1} \left(\nabla f_{K,m}\left(\theta_{K,m}^{t,e}\right) - \nabla f_K\left(\theta_K^t\right) + \nabla f_i\left(\theta_K^t\right)\right)$, $G = -\frac{1}{M}$, and $\Lambda = \frac{\lambda K}{\gamma_G \gamma_L}\left(\theta_K^0 - \theta_K^t\right)$, we can expand term $T_8$ as:

$$T_8 = \left(\frac{M}{M^2} - \frac{1}{M}\right) \sum_{m=1}^{M} \mathbb{E}_t \|\boldsymbol{t}_m\|^2 - \frac{1}{2M^2} \sum_{m \neq n} \mathbb{E}_t \|\boldsymbol{t}_m - \boldsymbol{t}_n\|^2$$

$$+ \left(1 - \frac{M}{M}\right) \|\Lambda\|^2 + \frac{1}{M} \sum_{m=1}^{M} \mathbb{E}_t \|\boldsymbol{t}_m - \Lambda\|^2 \tag{54}$$

$$= - \frac{1}{2M^2} \sum_{m \neq n} \mathbb{E}_t \|\boldsymbol{t}_m - \boldsymbol{t}_n\|^2 + \frac{1}{M} \sum_{m=1}^{M} \mathbb{E}_t \|\boldsymbol{t}_m - \Lambda\|^2 .$$

Since $T_7 \geq 0$ and $T_8 \geq 0$, we have:

$$\begin{cases} T_7 = -\frac{(N-1)}{2MN(M-1)} \sum_{m \neq n} \mathbb{E}_t \|\boldsymbol{t}_m - \boldsymbol{t}_n\|^2 + \frac{1}{M} \sum_{m=1}^{M} \mathbb{E}_t \|\boldsymbol{t}_m - \Lambda\|^2 \geq 0, \\[2mm] T_8 = -\frac{1}{2M^2} \sum_{m \neq n} \mathbb{E}_t \|\boldsymbol{t}_m - \boldsymbol{t}_n\|^2 + \frac{1}{M} \sum_{m=1}^{M} \mathbb{E}_t \|\boldsymbol{t}_m - \Lambda\|^2 \geq 0. \end{cases} \tag{55}$$

$$\Rightarrow \sum_{m \neq n} \mathbb{E}_t \|\boldsymbol{t}_m - \boldsymbol{t}_n\|^2 \leq 2M \sum_{m=1}^{M} \mathbb{E}_t \|\boldsymbol{t}_m - \Lambda\|^2 \leq 2M \cdot \frac{N(M-1)}{M(N-1)} \sum_{m=1}^{M} \mathbb{E}_t \|\boldsymbol{t}_m - \Lambda\|^2$$

For $\boldsymbol{t}_m$, we have:

$$\sum_{m=1}^{M} \mathbb{E}_t \|\boldsymbol{t}_m\|^2 = \sum_{m=1}^{M} \mathbb{E}_t \left\| \sum_{i=1}^{K} \sum_{e=0}^{E-1} \left(\nabla f_{K,m}\left(\theta_{K,m}^{t,e}\right) - \nabla f_K\left(\theta_K^t\right) + \nabla f_i\left(\theta_K^t\right)\right) \right\|^2$$

$$= \sum_{m=1}^{M} \mathbb{E}_t \left\| \sum_{i=1}^{K} \sum_{e=0}^{E-1} \left(\nabla f_{K,m}\left(\theta_{K,m}^{t,e}\right) - \nabla f_{K,m}\left(\theta_K^t\right) + \nabla f_{K,m}\left(\theta_K^t\right) - \nabla f_K\left(\theta_K^t\right)\right. \right.$$

$$\left. \left. + \nabla f_i\left(\theta_K^t\right) - \nabla f_K\left(\theta_K^t\right) + \nabla f_K\left(\theta_K^t\right)\right) \right\|^2$$

$$\leq 4 \sum_{m=1}^{M} \mathbb{E}_t \left\| \sum_{i=1}^{K} \sum_{e=0}^{E-1} \left(\nabla f_{K,m}\left(\theta_{K,m}^{t,e}\right) - \nabla f_{K,m}\left(\theta_K^t\right)\right) \right\|^2$$

$$+ 4 \sum_{m=1}^{M} \mathbb{E}_t \left\| \sum_{i=1}^{K} \sum_{e=0}^{E-1} \left(\nabla f_{K,m}\left(\theta_K^t\right) - \nabla f_K\left(\theta_K^t\right)\right) \right\|^2$$

$$+ 4 \sum_{m=1}^{M} \mathbb{E}_t \left\| \sum_{i=1}^{K} \sum_{e=0}^{E-1} \left(\nabla f_i\left(\theta_K^t\right) - \nabla f_K\left(\theta_K^t\right)\right) \right\|^2 + 4 \sum_{m=1}^{M} \mathbb{E}_t \left\| \sum_{i=1}^{K} \sum_{e=0}^{E-1} \nabla f_K\left(\theta_K^t\right) \right\|^2$$

$$\overset{(c_8)}{\leq} 4EK^2 \sum_{m=1}^{M} \sum_{e=0}^{E-1} \mathbb{E}_t \left\| \nabla f_{K,m}\left(\theta_{K,m}^{t,e}\right) - \nabla f_{K,m}\left(\theta_K^t\right) \right\|^2 + 4ME^2K^2 \left(\sigma_G^2 + \left\| \nabla f_K\left(\theta_K^t\right) \right\|^2\right)$$

$$+ 4ME^2(K-1)^2 \sigma_T^2$$

$$\overset{(c_9)}{\leq} 4EK^2L^2 \sum_{m=1}^{M} \mathbb{E}_t \left\| \theta_{K,m}^{t,e} - \theta_K^t \right\|^2 + 4ME^2K^2 \left(\sigma_G^2 + \left\| \nabla f_K\left(\theta_K^t\right) \right\|^2\right) + 4ME^2(K-1)^2 \sigma_T^2$$

$$\overset{(c_{10})}{\leq} 20ME^3K^2L^2\gamma_L^2\left(\sigma_L^2 + 6E\sigma_G^2\right) + \left(120ME^4L^2K^2\gamma_L^2 + 4ME^2K^2\right)\left\|\nabla f_K\left(\theta_K^t\right)\right\|^2$$
$$+ 4ME^2K^2\sigma_G^2 + 4ME^2\left(K-1\right)^2\sigma_T^2, \tag{56}$$

where $(c_8)$ follows Assumption 4.5 and Assumption 4.6, $(c_9)$ is due to Assumption 4.2, and $(c_{10})$ follows Lemma 4.9.

Based on (53), (54), (55), and (56), we have:

$$\frac{3\gamma_G^2\gamma_L^2 L}{2K\left(1+\lambda\right)^2}T_7 - \frac{\gamma_G\gamma_L}{2\left(1+\lambda\right)EK}T_8$$

$$= \frac{3\gamma_G^2\gamma_L^2 L}{2K\left(1+\lambda\right)^2}\left(-\frac{\left(N-1\right)}{2MN\left(M-1\right)}\sum_{m\neq n}\mathbb{E}_t\left\|\boldsymbol{t}_m - \boldsymbol{t}_n\right\|^2 + \frac{1}{M}\sum_{m=1}^M\mathbb{E}_t\left\|\boldsymbol{t}_m - \Lambda\right\|^2\right)$$

$$- \frac{\gamma_G\gamma_L}{2\left(1+\lambda\right)EK}\left(-\frac{1}{2M^2}\sum_{m\neq n}\mathbb{E}_t\left\|\boldsymbol{t}_m - \boldsymbol{t}_n\right\|^2 + \frac{1}{M}\sum_{m=1}^M\mathbb{E}_t\left\|\boldsymbol{t}_m - \Lambda\right\|^2\right) \tag{57}$$

$$= \left(\frac{\gamma_G\gamma_L}{4\left(1+\lambda\right)EM^2K} - \frac{3\left(N-1\right)\gamma_G^2\gamma_L^2 L}{4\left(1+\lambda\right)^2 KM\left(M-1\right)N}\right)\sum_{m\neq n}\mathbb{E}_t\left\|\boldsymbol{t}_m - \boldsymbol{t}_n\right\|^2$$

$$+ \left(\frac{3\gamma_G^2\gamma_L^2 L}{2MK\left(1+\lambda\right)^2} - \frac{\gamma_G\gamma_L}{2\left(1+\lambda\right)EKM}\right)\sum_{m=1}^M\mathbb{E}_t\left\|\boldsymbol{t}_m - \Lambda\right\|^2.$$

Since $\sum_{m\neq n}\left\|\boldsymbol{t}_m - \boldsymbol{t}_n\right\|^2 \leq 2M\sum_{m=1}^M\left\|\boldsymbol{t}_m - \Lambda\right\|^2$, we have:

$$\frac{3\gamma_G^2\gamma_L^2 L}{2K\left(1+\lambda\right)^2}T_7 - \frac{\gamma_G\gamma_L}{2\left(1+\lambda\right)EK}T_8$$

$$\leq \sum_{m=1}^M\mathbb{E}_t\left\|\boldsymbol{t}_m - \Lambda\right\|^2 \cdot \left(\frac{\gamma_G\gamma_L}{2\left(1+\lambda\right)EMK} - \frac{3\left(N-1\right)\gamma_G^2\gamma_L^2 L}{2\left(1+\lambda\right)^2 K\left(M-1\right)N}\right.$$

$$\left.+ \frac{3\gamma_G^2\gamma_L^2 L}{2MK\left(1+\lambda\right)^2} - \frac{\gamma_G\gamma_L}{2\left(1+\lambda\right)EKM}\right)$$

$$= \left(\frac{3\gamma_G^2\gamma_L^2 L}{2MK\left(1+\lambda\right)^2} - \frac{3\left(N-1\right)\gamma_G^2\gamma_L^2 L}{2\left(1+\lambda\right)^2 K\left(M-1\right)N}\right)\sum_{m=1}^M\mathbb{E}_t\left\|\boldsymbol{t}_m - \Lambda\right\|^2$$

$$\leq \frac{3\gamma_G^2\gamma_L^2}{2K\left(1+\lambda\right)^2}\cdot\frac{M-N}{MN\left(M-1\right)}\left(2\sum_{m=1}^M\mathbb{E}_t\left\|\boldsymbol{t}_m\right\|^2 + 2M\mathbb{E}_t\left\|\Lambda\right\|^2\right)$$

$$= \frac{3\gamma_G^2\gamma_L^2 L}{K\left(1+\lambda\right)^2}\cdot\frac{M-N}{MN\left(M-1\right)}\left(\sum_{m=1}^M\mathbb{E}_t\left\|\boldsymbol{t}_m\right\|^2 + M\mathbb{E}_t\left\|\Lambda\right\|^2\right)$$

$$\overset{(c_{11})}{\leq} \frac{\gamma_G\gamma_L E}{1+\lambda}\cdot\frac{3\gamma_G\gamma_L\left(M-N\right)L}{\left(1+\lambda\right)N\left(M-1\right)}\cdot\left(20E^2KL^2\gamma_L^2\left(\sigma_L^2 + 6E\sigma_G^2\right)\right. \tag{58}$$

$$\left.+ \left(120E^3KL^2\gamma_L^2 + 4EK\right)\left\|\nabla f_K\left(\theta_K^t\right)\right\|^2 + 4EK\sigma_G^2 + \frac{4E\left(K-1\right)^2}{K}\sigma_T^2 + EKB^2\right), \tag{59}$$

where $(c_{11})$ is due to (56) and Lemma 4.9.

Then we have:

$$\mathbb{E}_t\left[f_{1:K}\left(\theta_K^{t+1}\right)\right] \leq f_{1:K}\left(\theta_K^t\right) - \frac{1}{2}\left(\frac{\gamma_G\gamma_L E}{1+\lambda}\right)\left(1 - \frac{1}{K}\right)\left\|\nabla f_{1:K}\left(\theta_K^t\right)\right\|^2 - \frac{\gamma_G\gamma_L E}{1+\lambda}$$

$$\cdot\left(\frac{1}{2} - 30K\gamma_L^2 L^2 E^2 - \frac{3\gamma_G\gamma_L\left(M-N\right)L}{\left(1+\lambda\right)N\left(M-1\right)}\cdot\left(120E^3L^2K\gamma_L^2 + 4EK\right)\right)\left\|\nabla f_K\left(\theta_K^t\right)\right\|^2$$

$$
+ \frac{\gamma_G \gamma_L E}{1+\lambda} \left( \frac{L^2 (K-1)^2 \gamma_G^2 \gamma_L^2 E^2}{\lambda^2} + K + \frac{3\gamma_G \gamma_L (M-N) EKL}{(1+\lambda) N (M-1)} \right) B^2
$$

$$
+ \frac{\gamma_G \gamma_L E}{1+\lambda} \left( \left( 5\gamma_L^2 KEL^2 + \frac{60\gamma_G \gamma_L^3 (M-N) E^2 KL^3}{(1+\lambda) N (M-1)} \right) \left( \sigma_L^2 + 6E\sigma_G^2 \right) + \frac{3\gamma_G \gamma_L \sigma_L^2 L}{2N(1+\lambda)} \right.
$$

$$
+ \frac{(K-1)^2 E}{K} \left( \frac{3\gamma_G \gamma_L L}{2(1+\lambda)} + \frac{12\gamma_G \gamma_L (M-N) L}{(1+\lambda) N (M-1)} \right) \sigma_T^2 + \left. \frac{12\gamma_G \gamma_L (M-N) EKL\sigma_G^2}{(1+\lambda) N (M-1)} \right)
$$

$$
+ \frac{\gamma_G \gamma_L E}{1+\lambda} \cdot \left\| \nabla f_{1:K-1} \left( \theta_K^0 \right) \right\|^2
$$

$$
\overset{(c_{12})}{\leq} f_{1:K} \left( \theta_K^t \right) - \frac{1}{2} \left( \frac{\gamma_G \gamma_L E}{1+\lambda} \right) \left( 1 - \frac{1}{K} \right) \left\| \nabla f_{1:K} \left( \theta_K^t \right) \right\|^2
$$

$$
+ \frac{\gamma_G \gamma_L E}{1+\lambda} \left( \frac{\gamma_L^2 E^2 L^2}{\lambda^2} + K + \frac{3\gamma_G \gamma_L (M-N) EKL}{(1+\lambda) N (M-1)} \right) B^2
$$

$$
+ \frac{\gamma_G \gamma_L E}{1+\lambda} \left( \left( 5\gamma_L^2 KEL^2 + \frac{60\gamma_G \gamma_L^3 (M-N) E^2 KL^3}{(1+\lambda) N (M-1)} \right) \left( \sigma_L^2 + 6E\sigma_G^2 \right) + \frac{3\gamma_G \gamma_L \sigma_L^2 L}{2N(1+\lambda)} \right.
$$

$$
+ \frac{(K-1)^2 E}{K} \cdot \frac{3\gamma_G \gamma_L L}{1+\lambda} \left( \frac{1}{2} + \frac{4(M-N)}{N(M-1)} \right) \sigma_T^2 + \left. \frac{12\gamma_G \gamma_L (M-N) EKL\sigma_G^2}{(1+\lambda) N (M-1)} \right)
$$

$$
+ \frac{\gamma_G \gamma_L E}{1+\lambda} \cdot \left\| \nabla f_{1:K-1} \left( \theta_K^0 \right) \right\|^2, \tag{60}
$$

where $(c_{12})$ holds if $\frac{1}{2} - 30K\gamma_L^2 L^2 E^2 - \frac{3\gamma_G \gamma_L (M-N) L}{(1+\lambda) N (M-1)} \cdot \left( 120ME^3 L^2 K\gamma_L^2 + 4MEK \right) \geq 0$ and $\gamma_G \leq \frac{1}{K-1}$.

Rearranging (60) and summing it from $t = 0$ to $T - 1$, we have:

$$
\sum_{t=0}^{T-1} \left( 1 - \frac{1}{K} \right) \frac{\gamma_G \gamma_L E}{2(1+\lambda)} \mathbb{E} \left\| \nabla f_{1:K} \left( \theta_K^t \right) \right\|^2 \leq f_{1:K} \left( \theta_K^0 \right) - f_{1:K} \left( \theta_K^T \right)
$$

$$
+ \frac{\gamma_G \gamma_L E}{1+\lambda} \left( \frac{\gamma_L^2 E^2 L^2}{\lambda^2} + K + \frac{3\gamma_G \gamma_L (M-N) EKL}{(1+\lambda) N (M-1)} \right) B^2
$$

$$
+ \frac{\gamma_G \gamma_L E}{1+\lambda} \left( \left( 5\gamma_L^2 KEL^2 + \frac{60\gamma_G \gamma_L^3 (M-N) E^2 KL^3}{(1+\lambda) N (M-1)} \right) \left( \sigma_L^2 + 6E\sigma_G^2 \right) + \frac{3\gamma_G \gamma_L \sigma_L^2 L}{2N(1+\lambda)} \right. \tag{61}
$$

$$
+ \frac{(K-1)^2 E}{K} \cdot \frac{3\gamma_G \gamma_L L}{1+\lambda} \left( \frac{1}{2} + \frac{4(M-N)}{N(M-1)} \right) \sigma_T^2 + \left. \frac{12\gamma_G \gamma_L (M-N) EKL\sigma_G^2}{(1+\lambda) N (M-1)} \right)
$$

$$
+ \frac{\gamma_G \gamma_L E}{1+\lambda} \cdot \left\| \nabla f_{1:K-1} \left( \theta_K^0 \right) \right\|^2,
$$

which implies,

$$
\min_{t \in [T]} \mathbb{E} \left\| \nabla f_{1:K} \left( \theta_K^t \right) \right\|^2 \leq \frac{f_{1:K}^0 - f_{1:K}^*}{\frac{1 - \frac{1}{K}}{2(1+\lambda)} E\gamma_G \gamma_L T} + \Psi,
$$

where

$$
\Psi = \frac{2}{1 - \frac{1}{K}} \left[ \left( \frac{\gamma_L^2 E^2 L^2}{\lambda^2} + K + \frac{3\gamma_G \gamma_L (M-N) EKL}{(1+\lambda) N (M-1)} \right) B^2 \right.
$$

$$
+ \left( 5\gamma_L^2 KEL^2 + \frac{60\gamma_G \gamma_L^3 (M-N) E^2 KL^3}{(1+\lambda) N (M-1)} \right) \left( \sigma_L^2 + 6E\sigma_G^2 \right) + \frac{3\gamma_G \gamma_L \sigma_L^2 L}{2N(1+\lambda)}
$$

$$
+ \frac{(K-1)^2 E}{K} \cdot \frac{3\gamma_G \gamma_L L}{1+\lambda} \left( \frac{1}{2} + \frac{4(M-N)}{N(M-1)} \right) \sigma_T^2 + \frac{12\gamma_G \gamma_L (M-N) EKL\sigma_G^2}{(1+\lambda) N (M-1)}
$$

$$
+ \left. \left\| \nabla f_{1:K-1} \left( \theta_K^0 \right) \right\|^2 \right]. \tag{62}
$$

$\square$

# F. Additional Experimental Details

### F.1. Dataset.

We conduct our experiments on the popular domain shift datasets:

- **Digit-10**: It contains 10 digit categories and consists of 4 datasets: **MNIST** (LeCun et al., 2002), **USPS** (Hull, 2002), **SVHN** (Netzer et al., 2011), and **EMNIST** (Cohen et al., 2017). MNIST and EMNIST are handwritten style digits, but they are from different sources. SVHN is a real-world digit dataset from street view hour numbers, and the images in USPS are collected by the U.S. Postal Service. It contains 380,548 images in the training set and 78,039 images in the testing set.

- **VLCS** (Torralba & Efros, 2011): It contains 5 categories and consists of 4 datasets: VOC2007, LabelMe, Caltech-101, and Sun09. The domains differ in picture style, background and are taken with different shooting parameters. There are 7,486 images in all this dataset.

- **PACS** (Li et al., 2017): It contains 7 categories and the four domains are photo, art painting, cartoon, and sketch. The variation across domains lies in the image style. PACS contains a total of 9,991 images.

- **DN4IL** (Gowda et al., 2023): It is a is a subset of the DomainNet dataset, and it contains 100 categories and consists of 6 domains: clipart, infograph, painting, quickdraw, real, sketch. Since the original DomainNet dataset contains 345 redundancy classes, the DN4IL subtract the most representative and significant classes with size of 100 to be more effective for continual learning algorithms to train and test the capability.

### F.2. Hyperparameters.

Our experiments are consist of two parts: the main experiment and the ablation study experiment.

- The main experiment focuses on implementing SPECIAL in the three datasets and comparing it with baselines in two metrics, i.e., ACC and BWT. In the main experiment, the local learning rate $\gamma_L$ is initialized as 0.001 and decayed with 0.96 after each 5 epochs, the global learning rate $\gamma_G$ is initialized as 1 at the first task and decayed as $\frac{1}{i}$ at task $i$. In addition, we consider the total number of workers to be 8 and the number of participated clients to be 4, and we set communication rounds $T = \{20/20/30/30\}$ for {Digit-10/VLCS/PACS/DN4IL} and local epochs $E = 5$ for every dataset to guarantee the model converges in each task.

- In the ablation study experiment, the research focus has shifted to compare the performance of SPECIAL in different hyperparameter setting, we keep all settings consistent with the main experiment. We conduct ablation experiments with varying parameters in Digit-10 to observe the effect: we set communication round $T = \{1, 5, 10, 20, 30\}$, local epoch $E = \{1, 3, 5, 10\}$, and Dirichlet level $\alpha = \{0.05, 0.1, 0.3, 0.5, 1.0, 100\}$. We follow manual grid search to estimate the effects of different regularization coefficients $\lambda$ in all three datasets, using a step size of 0.2 in the range $[0, 1]$, and further narrow the step size between the two best-performing parameters to search the best coefficient. The result shows that the best coefficient for {Digit-10/VLCS/PACS} is between $\{[0.2, 0.4]/[0.4, 0.6]/[0, 0.2]\}$, and we further estimate the performance on $\{0.25, 0.30, 0.35\}/\{0.45, 0.50, 0.55\}/\{0.05, 0.10, 0.15\}/\{0.05, 0.10, 0.15\}$ to get the best coefficient as $\{0.25/0.40/0.05/0.10\}$.

In summary, the configuration details of all datasets is shown in Table 6 and the hyperparameters of baselines are listed in Table 7.

### F.3. Detailed Task Specific Performance Curves

In Table 9-12, we demonstrate the model performance on all tasks for the all 4 datasets. The performance of FedProx shows the same trend as FedAvg, indicating that the regularizer term in FedProx provides limited improvement in alleviating catastrophic forgetting.

### F.4. Additional Ablation Study

Figure 4 explores how communication rounds ($T$), local epochs ($E$), and data-heterogeneity level ($\alpha$) influence performance in Digit-10, thereby testing the claims of Section "Theoretical Analysis". (i) *Communication rounds $T$*. As T grows, ACC

*Table 6.* Configuration details

|  | Digit-10 | VLCS | PACS | DN4IL |
|---|---|---|---|---|
| Task size | 480M | 3.78G | 180M | 3.28G |
| Image number | 110k | 7.5k | 9.9k | 30k |
| Task number | 4 | 4 | 4 | 6 |
| Batch size | 32 | 32 | 32 | 32 |
| Local learning rate | 0.001 | 0.001 | 0.001 | 0.001 |
| Clients number | 8 | 8 | 8 | 8 |
| Epoch number | 5 | 5 | 5 | 5 |
| Round number | 20 | 20 | 20 | 30 |
| Dirichlet degree | 0.1 | 0.3 | 0.3 | 5.0 |

*Table 7.* Hyperparameters for baselines

| Baselines | Hyperparameters |
|---|---|
| FedCIL | $\lambda = 0.5$, image size: $32 \times 32$ for Digit-10, $224 * 224$ for others |
| MFCL | $w_{\text{div}} = 0.5, w_{\text{BN}} = 1, w_{\text{pr}} = 0.01, w_{\text{FT}} = 0.2, w_{\text{KD}} = 0.2$ |
| SR-FDIL | $\lambda = 0.5$ |
| pFedDIL | Threshold ($\lambda$): 0.8 for Digit-10, 0.5 for others |
| FLwF-2T | $\alpha = 0.5, \beta = 0.25$ |
| SPECIAL-C | $\lambda = 0.1$ for PACS, $\lambda = 0.2$ for others |

rises monotonically—more synchronous updates allow better optimization—while BWT exhibits a U-shape: it is weakest when $T$ is very small (the model leaves a task before converging), becomes increasingly negative as partial convergence accentuates forgetting, and finally improves once $T$ is large enough for the proximal anchor to propagate useful gradients back to earlier tasks. This behavior matches the trade-off predicted by Theorem 4.7. (ii) *Local epochs $E$.* Varying $E$ produces a pattern similar to that for $T$: ACC increases until large $E$ values yield diminishing returns, whereas BWT gradually declines. Longer local training stretches the distance between client updates and the anchor, slowing down the convergence rate in Corollary 4.12 and thereby reducing backward transfer. (iii) *Dirichlet parameter $\alpha$.* Smaller $\alpha$ (stronger non-IID partitions) degrades ACC, reflecting the additive variance terms $\sigma_L^2$ and $\sigma_G^2$ in Theorem 4.10. With highly heterogeneous clients, each round introduces greater noise, and more communication is required to attain the same accuracy. Overall, the empirical trends align closely with the theoretical rates: ACC improves with additional communication or computation, while BWT is sensitive to the balance between local plasticity and the stability enforced by the server-side anchor.

*Table 8.* The worst difference between the earlier-task and the final task

| Method | Digit-10 | VLCS | PACS | DN4IL |
|---|---|---|---|---|
| FedAvg | $-54.73 \pm 2.32$ | $-9.64 \pm 5.83$ | $-30.37 \pm 3.93$ | $-78.81 \pm 0.80$ |
| FedProx | $-52.60 \pm 1.49$ | $-12.22 \pm 4.81$ | $-26.38 \pm 2.44$ | $-30.49 \pm 1.27$ |
| FedCIL | $-41.14 \pm 7.39$ | $-8.54 \pm 6.94$ | $-16.35 \pm 3.80$ | $-38.38 \pm 0.18$ |
| MFCL | $-26.31 \pm 3.15$ | $-9.69 \pm 4.83$ | $-20.73 \pm 3.90$ | $-37.65 \pm 3.28$ |
| SR-FDIL | $-52.76 \pm 4.71$ | $-10.47 \pm 5.06$ | $-29.41 \pm 2.72$ | $-58.32 \pm 0.44$ |
| pFedDIL | $-33.86 \pm 5.47$ | $-12.13 \pm 3.74$ | $-27.93 \pm 0.68$ | $-36.76 \pm 1.01$ |
| FLwF-2T | $-38.60 \pm 4.37$ | $-7.59 \pm 2.83$ | $-26.29 \pm 8.47$ | $-36.87 \pm 0.05$ |
| SPECIAL-C | $-28.06 \pm 1.77$ | $-9.64 \pm 1.47$ | $-25.10 \pm 2.13$ | $-25.87 \pm 0.35$ |
| **SPECIAL (ours)** | $-34.37 \pm 6.65$ | $-5.84 \pm 5.26$ | $-20.38 \pm 3.83$ | $-37.46 \pm 1.28$ |

*Table 9.* Task specific performance curves on Digit-10 ($\alpha = 0.1$)

| Method | USPS | SVHN | EMNIST | MNIST | ACC |
|---|---|---|---|---|---|
| FedAvg | 0.8034 | 0.2839 | 0.4106 | 0.8586 | 0.5869 |
| FedProx | 0.8064 | 0.3069 | 0.3836 | 0.8626 | 0.5869 |
| FedCIL | 0.6795 | 0.1952 | 0.3057 | 0.8576 | 0.5095 |
| MFCL | 0.8266 | 0.3446 | 0.4142 | 0.8607 | 0.6115 |
| SR-FDIL | 0.8181 | **0.2559** | 0.3777 | 0.9335 | 0.6163 |
| pFedDIL | 0.7816 | 0.1552 | 0.1608 | 0.7491 | 0.4617 |
| FLwF-2T | 0.6630 | 0.1570 | 0.2321 | 0.8485 | 0.4739 |
| SPECIAL-C | 0.7873 | 0.1897 | 0.2605 | 0.7868 | 0.5061 |
| **SPECIAL (ours)** | **0.8618** | 0.2457 | **0.4797** | **0.8977** | **0.6212** |

*Table 10.* Task specific performance curves on VLCS ($\alpha = 0.3$)

| Method | Caltech101 | LabelMe | SUN09 | VOC2007 | ACC |
|---|---|---|---|---|---|
| FedAvg | 0.6126 | 0.4938 | 0.4472 | 0.4953 | 0.5122 |
| FedProx | 0.5679 | 0.4646 | 0.4215 | 0.4755 | 0.4824 |
| FedCIL | 0.5840 | 0.4887 | 0.4358 | 0.4449 | 0.4884 |
| MFCL | 0.5214 | 0.3932 | 0.3805 | 0.4208 | 0.4289 |
| SR-FDIL | 0.6192 | 0.4681 | 0.4305 | 0.4754 | 0.4938 |
| pFedDIL | 0.6971 | **0.5530** | 0.4215 | 0.4260 | 0.5244 |
| FLwF-2T | 0.6933 | 0.5229 | 0.4532 | 0.4604 | 0.5324 |
| SPECIAL-C | 0.6477 | 0.4721 | 0.4447 | **0.4822** | 0.5117 |
| **SPECIAL (ours)** | **0.7056** | 0.5385 | **0.4813** | 0.4715 | **0.5492** |

*Table 11.* Task specific performance curves on PACS ($\alpha = 0.3$)

| Method | Sketch | Cartoon | Photo | Art Painting | ACC |
|---|---|---|---|---|---|
| FedAvg | 0.2986 | 0.3293 | 0.4480 | 0.4357 | 0.3779 |
| FedProx | 0.3015 | 0.3370 | 0.4520 | 0.4292 | 0.3799 |
| FedCIL | 0.2372 | 0.3468 | 0.4539 | 0.3665 | 0.3511 |
| MFCL | 0.2345 | 0.2576 | 0.4474 | 0.4476 | 0.3468 |
| SR-FDIL | **0.3893** | 0.3935 | 0.5348 | 0.4791 | 0.4492 |
| pFedDIL | 0.3719 | 0.3878 | 0.4282 | 0.4024 | 0.3976 |
| FLwF-2T | 0.3401 | 0.3646 | **0.5718** | 0.4452 | 0.4304 |
| SPECIAL-C | 0.3624 | 0.4130 | 0.5114 | 0.4451 | 0.4330 |
| **SPECIAL (ours)** | 0.3761 | **0.4255** | 0.5253 | **0.4848** | **0.4529** |

*Table 12.* Task specific performance curves on DN4IL ($\alpha = 5$)

| Method | Sketch | Infograph | Painting | Quickdraw | Real | Clipart | ACC |
|---|---|---|---|---|---|---|---|
| FedAvg | 0.1158 | 0.0480 | 0.0611 | 0.0799 | 0.3579 | 0.3258 | 0.1647 |
| FedCIL | 0.0888 | 0.0375 | 0.0831 | 0.1305 | 0.2511 | 0.2663 | 0.1429 |
| MFCL | 0.1688 | 0.0588 | 0.1296 | 0.1045 | 0.3053 | 0.3729 | 0.1900 |
| SR-FDIL | 0.1609 | 0.0108 | 0.1110 | 0.1404 | **0.4655** | 0.3956 | 0.2140 |
| pFedDIL | **0.2902** | 0.0379 | 0.0818 | 0.0625 | 0.1165 | 0.1594 | 0.1247 |
| FLwF-2T | 0.2450 | 0.0377 | 0.1367 | **0.2211** | 0.3202 | **0.4319** | 0.2321 |
| SPECIAL-C | 0.2775 | 0.0622 | 0.1364 | 0.1188 | 0.2465 | 0.3412 | 0.1988 |
| **SPECIAL (ours)** | 0.2298 | **0.0681** | **0.1754** | **0.1724** | 0.4128 | 0.3997 | **0.2430** |

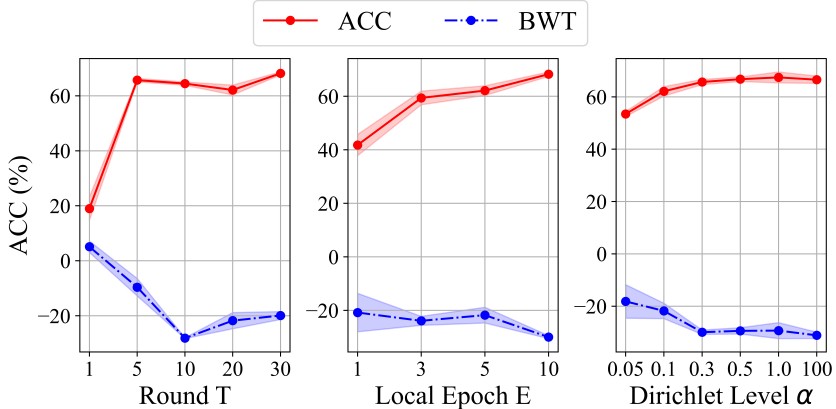

*Figure 4.* Visualized performance w.r.t $T$, $E$, $\alpha$.

## F.5. Experiment Details of "Client- vs. Server-side Proximal Terms"

In Section 2.2, we compare the performance of models with client-side and server-side proximal term to explore the effect of different locations of the proximal term, and the model with client-side proximal term is denoted as SPECIAL-C in the following sections. The observation is obtained from the model performance in the training process on the second task of Digit-10.

After obtaining the parameter result of each communication round, we estimate the result on three types of test datasets while training on task 2:

- the first task is recognized as the previous task, so the test dataset of task 1 is used to estimate the memorization ability of two models;

- the second task is recognized as the current task, so the test dataset of task 2 is used to measure the ability of learning new information with different settings;

- the combination of the two test datasets is used to estimate the overall performance.

The difference in the term location leads to the distinct update rules between SPECIAL and SPECIAL-C. In the task $i, i \geq 2$, the local update rule for client $m$ in each epoch is

$$\theta_{i,m}^{t,m} = \text{prox}_\lambda \left( \theta_{i,m}^{t,e} - \gamma_G g_{i,m}^{t,e} \right),$$

where $\text{prox}_\lambda(x) := \arg\min_{\theta \in \mathbb{R}^d} \{ \frac{1}{2} \|\theta - x\|^2 + \lambda \|\theta - \theta_{i-1}\|^2 \}$ is the proximal mapping (Xiao & Zhang, 2014). After collecting the difference vectors, the server of SPECIAL-C averages them without incorporating the previous global parameters. The complete algorithm of SPECIAL-C is summarized in Algorithm 2. In addition, we observe a similar $\lambda$-controlled stability-plasticity trade-off in SPECIAL-C. The results of the $\lambda$-sensitivity of SPECIAL-C on Digit-10 is shown in Table 13. Specifically, larger $\lambda$ values encourage the model to preserve previous knowledge more aggressively, thereby improving stability, while smaller $\lambda$ values allow the model to adapt more flexibly to new tasks, leading to stronger plasticity. The best performance is achieved when these two effects are properly balanced, which is consistent with the observation in SPECIAL.

*Table 13.* The $\lambda$-sensitivity of SPECIAL-C on Digit-10 ($\alpha = 0.1$).

| $\lambda$ | 0 | 0.2 | 0.4 | 0.6 | 0.8 | 1.0 |
|---|---|---|---|---|---|---|
| ACC (%) ↑ | 47.89 | 50.26 | 46.73 | 45.37 | 42.77 | 38.01 |
| BWT (%) ↑ | $-10.45$ | $-8.81$ | $-6.22$ | $-2.54$ | $-1.34$ | 2.56 |

---

**Algorithm 2** The `SPECIAL-C` Algorithm

---

1: Initialize $\theta_0$
2: **for** task $i = 1$ to $K$ **do**
3:     Set task initial point: $\theta_i^0 = \theta_{i-1}$
4:     **for** round $t = 0$ to $T - 1$ **do**
5:         The server randomly selects a set $\mathcal{S}_i^t$ of $N$ clients
6:         **for** client $m \in \mathcal{S}_i^t$ **in parallel do**
7:             Download from server: $\theta_{i,m}^{t,0} = \theta_i^t$
8:             **for** epoch $e = 0$ to $E - 1$ **do**
9:                 Calculate: $g_{i,m}^{t,e} = \nabla f_{i,m}\left(\theta_{i,m}^{t,e}, \xi_{i,m}^{t,e}\right)$
10:                 Local update: $\theta_{i,m}^{t,e+1} = \begin{cases} \theta_{i,m}^{t,e} - \gamma_L g_{i,m}^{t,e}, & i = 1 \\ \theta_{i,m}^{t,m} = \text{prox}_\lambda\left(\theta_{i,m}^{t,e} - \gamma_L g_{i,e}^{t,e}\right), & i \geq 2 \end{cases}$
11:             **end for**
12:             $\Delta_{i,m}^t = \theta_{i,m}^{t,E} - \theta_{i,m}^{t,0} = -\gamma_L \sum_{e=0}^{E-1} g_{i,m}^{t,e}$
13:             Send $\Delta_{i,m}^t$ to server
14:         **end for**
15:         Server receives all $\Delta_{i,m}^t$
16:         Compute: $\Delta_i^t = \frac{1}{M} \sum_{m=1}^M \Delta_{i,m}^t$
17:         Server update: $\theta_i^{t+1} = \theta_i^t + \gamma_G \Delta_i^t$
18:     **end for**
19:     Set final model of task $i$: $\theta_i = \theta_i^T$
20: **end for**

---

### F.6. The generalizability to class-incremental settings

Although the method and theoretical analysis of `SPECIAL` are developed under the domain-incremental setting, we further explore its feasibility in the class-incremental scenario. Compared with domain-incremental learning, class-incremental learning requires certain classifier layers to be dynamically expanded and adjusted according to the number of observed classes, so that the classification objective can evolve as the number of tasks increases. Empirically, we split CIFAR-100 into 5 tasks with 20 classes per task under Dirichlet parameter $\alpha = 1$. The result is shown in Table 14, and our `SPECIAL` achieves the best ACC and BWT among all compared methods.

*Table 14.* The results on Split CIFAR-100 under the class-incremental setting.

| Method | ACC (%) ↑ | BWT (%) ↑ |
|---|---|---|
| FedAvg | 37.92 | $-29.64$ |
| FedProx | 37.32 | $-27.51$ |
| FedCIL | 31.73 | $-21.19$ |
| MFCL | 28.37 | $-27.64$ |
| SR-FDIL | 35.43 | $-32.28$ |
| pFedDIL | 31.72 | $-12.61$ |
| FLwF-2T | 38.65 | $-23.76$ |
| SPECIAL-C | 34.60 | $-18.68$ |
| **SPECIAL (ours)** | **40.68** | **$-9.00$** |

### F.7. Additional numerical efficiency comparisons

We provide the direct numerical efficiency evidence on Digit-10 dataset in Table 15. *Memory overhead (Mem)* is measured by the total size of parameters involved in each client, including the trainable parameters and any dataset buffer. *Peak memory usage (Peak)* is measured by the maximum GPU memory consumption observed during training.

*Table 15.* The numerical efficiency comparisons on Digit-10

|  | memory overhead (Mb) | peak memory usage (Gb) |
|---|---|---|
| FedAvg | **42.69** | **1.54** |
| FedProx | **42.69** | 1.73 |
| FedCIL | 59.05 | 1.93 |
| MFCL | 117.8 | 3.28 |
| SR-FDIL | 55.82 | 2.29 |
| pFedDIL | 130.15 | 3.65 |
| FLwF-2T | 85.38 | 1.59 |
| SPECIAL-C | 85.38 | 1.58 |
| **SPECIAL (ours)** | **42.69** | 1.56 |

## F.8. Detailed results on different partial participation rates

### F.8.1. $N = 2$

*Table 16.* Main FDIL results under partial participation with $N = 2$.

| Type | Method | Digit-10 | | VLCS | | PACS | | DN4IL | |
|---|---|---|---|---|---|---|---|---|---|
| | | ACC (%) ↑ | BWT (%) ↑ | ACC (%) ↑ | BWT (%) ↑ | ACC (%) ↑ | BWT (%) ↑ | ACC (%) ↑ | BWT (%) ↑ |
| Standard-FL | FedAvg | $54.68 \pm 1.95$ | $-32.72 \pm 1.76$ | $34.58 \pm 4.26$ | $-16.58 \pm 7.02$ | $32.58 \pm 2.49$ | $-17.15 \pm 7.17$ | $18.07 \pm 0.29$ | $-12.78 \pm 0.53$ |
| | FedProx | $53.12 \pm 0.60$ | $-35.88 \pm 0.76$ | $47.97 \pm 0.93$ | $-34.38 \pm 3.44$ | $30.04 \pm 1.29$ | $-26.27 \pm 2.93$ | $19.52 \pm 0.16$ | $-12.32 \pm 0.46$ |
| Replay-based | FedCIL | $22.94 \pm 1.74$ | $-26.14 \pm 4.40$ | $38.40 \pm 2.62$ | $-1.45 \pm 3.70$ | $27.89 \pm 2.65$ | $-13.33 \pm 4.72$ | $13.18 \pm 0.51$ | $\mathbf{-8.30 \pm 0.43}$ |
| | MFCL | $46.68 \pm 3.98$ | $-19.35 \pm 4.79$ | $43.15 \pm 2.11$ | $-3.25 \pm 1.01$ | $23.31 \pm 1.97$ | $-10.58 \pm 5.08$ | $17.47 \pm 0.63$ | $-11.20 \pm 0.76$ |
| | SR-FDIL | $56.50 \pm 3.73$ | $-27.79 \pm 3.79$ | $45.64 \pm 3.30$ | $-14.31 \pm 5.22$ | $30.94 \pm 3.17$ | $-14.33 \pm 4.49$ | $19.04 \pm 0.13$ | $-12.63 \pm 0.71$ |
| Replay-free | pFedDIL | $42.15 \pm 0.71$ | $\mathbf{-14.72 \pm 1.52}$ | $43.84 \pm 2.82$ | $-3.85 \pm 3.56$ | $21.23 \pm 1.22$ | $-19.16 \pm 0.92$ | $12.41 \pm 0.23$ | $-11.85 \pm 0.05$ |
| | FLwF-2T | $38.27 \pm 1.70$ | $-17.51 \pm 1.28$ | $48.63 \pm 3.68$ | $-7.98 \pm 2.51$ | $25.15 \pm 4.44$ | $\mathbf{-10.14 \pm 2.45}$ | $19.72 \pm 0.18$ | $-14.99 \pm 0.45$ |
| | SPECIAL-C | $48.86 \pm 2.43$ | $-16.18 \pm 3.66$ | $43.93 \pm 4.99$ | $-14.94 \pm 3.21$ | $31.39 \pm 1.56$ | $-15.87 \pm 4.50$ | $18.18 \pm 0.27$ | $-15.93 \pm 0.60$ |
| | **SPECIAL (ours)** | $\mathbf{57.57 \pm 4.08}$ | $-16.13 \pm 2.60$ | $\mathbf{50.18 \pm 4.06}$ | $\mathbf{-0.09 \pm 0.81}$ | $\mathbf{33.50 \pm 2.41}$ | $-12.83 \pm 2.84$ | $\mathbf{21.20 \pm 0.27}$ | $-10.74 \pm 0.58$ |

*Table 17.* Task specific performance curves on Digit-10 with $N = 2$ ($\alpha = 0.1$).

| Method | USPS | SVHN | EMNIST | MNIST | ACC |
|---|---|---|---|---|---|
| FedAvg | 0.8219 | 0.2269 | 0.2325 | **0.9059** | 0.5468 |
| FedProx | **0.8283** | 0.2457 | 0.1618 | 0.8890 | 0.5312 |
| FedCIL | 0.3069 | 0.1083 | 0.1449 | 0.3573 | 0.2293 |
| MFCL | 0.6095 | **0.4552** | 0.2913 | 0.5113 | 0.4668 |
| SR-FDIL | 0.8065 | 0.2404 | 0.3121 | 0.9011 | 0.5650 |
| pFedDIL | 0.7579 | 0.1406 | 0.1322 | 0.6552 | 0.4215 |
| FLwF-2T | 0.6090 | 0.1243 | 0.1404 | 0.6572 | 0.3827 |
| SPECIAL-C | 0.7656 | 0.2010 | 0.2740 | 0.7137 | 0.4886 |
| **SPECIAL (ours)** | 0.8158 | 0.2198 | **0.4421** | 0.8252 | **0.5757** |

*Table 18.* Task specific performance curves on VLCS with $N = 2$ ($\alpha = 0.3$).

| Method | Caltech101 | LabelMe | SUN09 | VOC2007 | ACC |
|---|---|---|---|---|---|
| FedAvg | 0.2441 | 0.3037 | 0.4252 | 0.4102 | 0.3458 |
| FedProx | 0.3762 | 0.3713 | 0.4013 | **0.7699** | 0.4797 |
| FedCIL | 0.3884 | 0.4128 | 0.3756 | 0.3595 | 0.3841 |
| MFCL | 0.4042 | 0.3885 | 0.4638 | 0.4692 | 0.4314 |
| SR-FDIL | 0.3526 | 0.4157 | 0.5175 | 0.5398 | 0.4564 |
| pFedDIL | 0.5852 | 0.3734 | 0.4515 | 0.3434 | 0.4384 |
| FLwF-2T | 0.6673 | 0.4578 | 0.4074 | 0.4126 | 0.4863 |
| SPECIAL-C | 0.3622 | 0.4028 | **0.5272** | 0.4651 | 0.4393 |
| **SPECIAL (ours)** | **0.7066** | **0.4837** | 0.4386 | 0.3784 | **0.5018** |

*Table 19.* Task specific performance curves on PACS with $N = 2$ ($\alpha = 0.3$).

| Method | Sketch | Cartoon | Photo | Art Painting | ACC |
|---|---|---|---|---|---|
| FedAvg | 0.2448 | 0.2893 | 0.3994 | 0.3695 | 0.3257 |
| FedProx | 0.2993 | 0.2996 | 0.3013 | 0.3014 | 0.3004 |
| FedCIL | 0.2095 | 0.2385 | 0.3968 | 0.2708 | 0.2789 |
| MFCL | 0.2147 | 0.2529 | 0.2227 | 0.2423 | 0.2331 |
| SR-FDIL | 0.2051 | 0.2928 | **0.4045** | 0.3351 | 0.3094 |
| pFedDIL | 0.2470 | 0.1829 | 0.2078 | 0.2116 | 0.2123 |
| FLwF-2T | 0.2390 | 0.2328 | 0.2699 | 0.2643 | 0.2515 |
| SPECIAL-C | 0.2648 | 0.3014 | 0.3817 | 0.3078 | 0.3139 |
| **SPECIAL (ours)** | **0.3095** | **0.3324** | 0.3657 | **0.3322** | **0.3349** |

*Table 20.* Task specific performance curves on DN4IL with $N = 2$ ($\alpha = 5$).

| Method | Sketch | Infograph | Painting | Quickdraw | Real | Clipart | ACC |
|---|---|---|---|---|---|---|---|
| FedAvg | 0.1754 | 0.0161 | 0.1160 | 0.1075 | 0.2419 | 0.4271 | 0.1807 |
| FedProx | 0.1855 | 0.0380 | 0.1329 | 0.1237 | 0.2585 | **0.4329** | 0.1953 |
| FedCIL | 0.0812 | 0.0428 | 0.0871 | 0.1096 | 0.1795 | 0.2906 | 0.1318 |
| MFCL | 0.1627 | 0.0607 | 0.1221 | 0.0972 | 0.2419 | 0.3639 | 0.1747 |
| SR-FDIL | 0.1873 | 0.0300 | 0.1303 | 0.1121 | 0.2540 | 0.4285 | 0.1904 |
| pFedDIL | **0.2519** | 0.0435 | 0.0938 | 0.0689 | 0.1249 | 0.1620 | 0.1242 |
| FLwF-2T | 0.1840 | 0.0300 | 0.1275 | **0.1743** | 0.2630 | 0.4046 | 0.1972 |
| SPECIAL-C | 0.2010 | 0.0575 | 0.1378 | 0.1126 | 0.2363 | 0.3455 | 0.1818 |
| **SPECIAL (ours)** | 0.2138 | **0.0649** | **0.1620** | 0.1404 | **0.2858** | 0.4048 | **0.2119** |

### F.8.2. $N = 6$

*Table 21.* Main FDIL results under partial participation with $N = 6$.

| Type | Method | Digit-10 | | VLCS | | PACS | | DN4IL | |
|---|---|---|---|---|---|---|---|---|---|
| | | ACC (%) ↑ | BWT (%) ↑ | ACC (%) ↑ | BWT (%) ↑ | ACC (%) ↑ | BWT (%) ↑ | ACC (%) ↑ | BWT (%) ↑ |
| Standard-FL | FedAvg | $59.87 \pm 1.70$ | $-32.48 \pm 1.67$ | $44.16 \pm 2.28$ | $-17.08 \pm 3.74$ | $38.82 \pm 1.90$ | $-22.61 \pm 3.39$ | $20.90 \pm 0.24$ | $-12.00 \pm 0.13$ |
| | FedProx | $58.96 \pm 1.37$ | $-32.76 \pm 2.36$ | $47.49 \pm 1.14$ | $-34.49 \pm 1.32$ | $39.98 \pm 0.97$ | $-27.67 \pm 4.06$ | $20.52 \pm 0.22$ | $-12.42 \pm 0.36$ |
| Replay-based | FedCIL | $50.48 \pm 1.31$ | $-28.42 \pm 1.97$ | $51.70 \pm 1.66$ | $-1.99 \pm 1.45$ | $32.63 \pm 2.96$ | $-12.56 \pm 2.041$ | $11.82 \pm 0.53$ | $-13.96 \pm 0.25$ |
| | MFCL | $58.14 \pm 4.23$ | $-25.14 \pm 2.23$ | $29.24 \pm 3.37$ | $-17.52 \pm 2.45$ | $35.93 \pm 1.67$ | $-16.41 \pm 6.99$ | $20.59 \pm 0.36$ | $-11.56 \pm 1.97$ |
| | SR-FDIL | $59.37 \pm 0.43$ | $-28.96 \pm 2.93$ | $45.82 \pm 2.61$ | $-22.78 \pm 6.97$ | $35.78 \pm 0.60$ | $-23.09 \pm 2.61$ | $21.11 \pm 1.09$ | $-12.26 \pm 2.21$ |
| Replay-free | pFedDIL | $42.77 \pm 3.02$ | $\mathbf{-9.35 \pm 0.42}$ | $51.70 \pm 4.08$ | $-1.42 \pm 3.49$ | $23.03 \pm 2.86$ | $-19.35 \pm 3.97$ | $11.20 \pm 0.89$ | $\mathbf{-2.77 \pm 1.66}$ |
| | FLwF-2T | $45.09 \pm 0.71$ | $-15.34 \pm 1.87$ | $52.84 \pm 1.34$ | $-0.29 \pm 1.15$ | $27.88 \pm 3.12$ | $\mathbf{-8.87 \pm 4.30}$ | $20.20 \pm 0.88$ | $-12.24 \pm 0.33$ |
| | SPECIAL-C | $55.07 \pm 0.76$ | $-23.61 \pm 2.25$ | $44.46 \pm 7.47$ | $-14.41 \pm 5.32$ | $37.57 \pm 2.85$ | $-22.14 \pm 3.95$ | $20.07 \pm 1.32$ | $-11.82 \pm 2.43$ |
| | **SPECIAL (ours)** | $\mathbf{62.88 \pm 0.77}$ | $-25.15 \pm 0.41$ | $\mathbf{53.61 \pm 3.41}$ | $\mathbf{0.00 \pm 3.54}$ | $\mathbf{40.92 \pm 2.72}$ | $-12.41 \pm 1.41$ | $\mathbf{21.28 \pm 0.65}$ | $-10.76 \pm 0.91$ |

*Table 22.* Task specific performance curves on Digit-10 with $N = 6$ ($\alpha = 0.1$).

| Method | USPS | SVHN | EMNIST | MNIST | ACC |
|---|---|---|---|---|---|
| FedAvg | 0.8877 | 0.2792 | 0.2548 | 0.9731 | 0.5987 |
| FedProx | **0.8991** | 0.2740 | 0.2351 | 0.9501 | 0.5896 |
| FedCIL | 0.6878 | 0.1906 | 0.2777 | 0.8631 | 0.5048 |
| MFCL | 0.6944 | **0.5203** | 0.3627 | 0.7481 | 0.5814 |
| SR-FDIL | 0.8494 | 0.2385 | 0.3510 | 0.9358 | 0.5937 |
| pFedDIL | 0.7351 | 0.1289 | 0.1221 | 0.7248 | 0.4277 |
| FLwF-2T | 0.6504 | 0.1397 | 0.1863 | 0.8273 | 0.4509 |
| SPECIAL-C | 0.8352 | 0.1977 | 0.2793 | 0.8907 | 0.5507 |
| **SPECIAL (ours)** | 0.8650 | 0.1970 | **0.5137** | 0.9396 | **0.6288** |

*Table 23.* Task specific performance curves on VLCS with $N = 6$ ($\alpha = 0.3$).

| Method | Caltech101 | LabelMe | SUN09 | VOC2007 | ACC |
|---|---|---|---|---|---|
| FedAvg | 0.3837 | 0.3605 | 0.4378 | 0.5843 | 0.4416 |
| FedProx | 0.3831 | 0.3304 | 0.4252 | **0.7608** | 0.4749 |
| FedCIL | 0.6781 | 0.4847 | 0.4309 | 0.4743 | 0.5170 |
| MFCL | 0.2716 | 0.2529 | 0.3122 | 0.3329 | 0.2924 |
| SR-FDIL | 0.3647 | 0.3816 | 0.4569 | 0.6294 | 0.4582 |
| pFedDIL | **0.7702** | **0.5319** | 0.3829 | 0.3832 | 0.5171 |
| FLwF-2T | 0.7481 | 0.4441 | 0.4677 | 0.4535 | 0.5283 |
| SPECIAL-C | 0.3846 | 0.3801 | 0.4549 | 0.5586 | 0.4446 |
| **SPECIAL** (ours) | 0.7379 | 0.5214 | **0.4744** | 0.4106 | **0.5361** |

*Table 24.* Task specific performance curves on PACS with $N = 6$ ($\alpha = 0.3$).

| Method | Sketch | Cartoon | Photo | Art Painting | ACC |
|---|---|---|---|---|---|
| FedAvg | 0.2646 | 0.3178 | 0.5006 | 0.4697 | 0.3882 |
| FedProx | 0.2833 | 0.3167 | 0.4830 | **0.5162** | 0.3998 |
| FedCIL | 0.2555 | 0.2989 | 0.4318 | 0.3191 | 0.3263 |
| MFCL | 0.2578 | 0.3027 | 0.4480 | 0.4287 | 0.3593 |
| SR-FDIL | 0.2368 | 0.3205 | 0.4641 | 0.4099 | 0.3578 |
| pFedDIL | 0.2351 | 0.2184 | 0.2443 | 0.2237 | 0.2304 |
| FLwF-2T | 0.2499 | 0.2136 | 0.3657 | 0.2862 | 0.2789 |
| SPECIAL-C | 0.2777 | 0.3303 | 0.4655 | 0.4292 | 0.3757 |
| **SPECIAL** (ours) | **0.3146** | **0.3582** | **0.5088** | 0.4551 | **0.4092** |

*Table 25.* Task specific performance curves on DN4IL with $N = 6$ ($\alpha = 5$).

| Method | Sketch | Infograph | Painting | Quickdraw | Real | Clipart | ACC |
|---|---|---|---|---|---|---|---|
| FedAvg | 0.2012 | 0.0554 | 0.1481 | 0.1452 | 0.2719 | **0.4323** | 0.2090 |
| FedProx | 0.1891 | 0.0618 | 0.1487 | 0.1316 | 0.2771 | 0.4227 | 0.2052 |
| FedCIL | 0.0662 | 0.0380 | 0.0856 | 0.0800 | 0.1704 | 0.2690 | 0.1182 |
| MFCL | 0.1869 | 0.0590 | 0.1737 | 0.0903 | **0.3041** | 0.4214 | 0.2059 |
| SR-FDIL | 0.1961 | **0.0683** | 0.1716 | 0.1211 | 0.2822 | 0.4274 | 0.2111 |
| pFedDIL | **0.2352** | 0.0389 | 0.0833 | 0.0534 | 0.1175 | 0.1436 | 0.1120 |
| FLwF-2T | 0.1888 | 0.0299 | 0.1250 | **0.1846** | 0.2659 | 0.4176 | 0.2020 |
| SPECIAL-C | 0.2389 | 0.0625 | 0.1508 | 0.1177 | 0.2574 | 0.3767 | 0.2007 |
| **SPECIAL** (ours) | 0.2212 | 0.0556 | **0.1633** | 0.1414 | 0.2954 | 0.3999 | **0.2128** |

### F.8.3. $N = 8$

*Table 26.* Main FDIL results under partial participation with full participation.

| Type | Method | Digit-10 | | VLCS | | PACS | | DN4IL | |
|---|---|---|---|---|---|---|---|---|---|
| | | ACC (%) ↑ | BWT (%) ↑ | ACC (%) ↑ | BWT (%) ↑ | ACC (%) ↑ | BWT (%) ↑ | ACC (%) ↑ | BWT (%) ↑ |
| Standard-FL | FedAvg | $61.18 \pm 1.21$ | $-28.78 \pm 1.02$ | $51.91 \pm 3.17$ | $-26.58 \pm 4.78$ | $42.09 \pm 0.41$ | $-20.70 \pm 1.87$ | $21.04 \pm 0.04$ | $-12.15 \pm 0.34$ |
| | FedProx | $55.23 \pm 1.82$ | $-28.32 \pm 0.88$ | $53.36 \pm 1.04$ | $-34.45 \pm 1.13$ | $42.91 \pm 1.44$ | $-29.27 \pm 1.04$ | $20.32 \pm 2.31$ | $-10.03 \pm 1.25$ |
| Replay-based | FedCIL | $49.25 \pm 0.88$ | $-27.60 \pm 5.04$ | $46.31 \pm 3.73$ | $-7.81 \pm 1.66$ | $35.35 \pm 0.65$ | $-14.55 \pm 1.11$ | $14.33 \pm 0.22$ | $-10.23 \pm 0.65$ |
| | MFCL | $60.17 \pm 3.04$ | $-23.87 \pm 6.30$ | $47.30 \pm 3.50$ | $-6.40 \pm 0.05$ | $35.80 \pm 1.97$ | $-18.82 \pm 5.67$ | $19.21 \pm 0.97$ | $-9.19 \pm 0.99$ |
| | SR-FDIL | $63.57 \pm 0.61$ | $-28.42 \pm 0.39$ | $44.79 \pm 1.56$ | $-19.04 \pm 1.47$ | $42.90 \pm 1.81$ | $-23.69 \pm 1.64$ | $21.46 \pm 1.07$ | $-11.40 \pm 1.00$ |
| Replay-free | pFedDIL | $43.31 \pm 1.88$ | $\mathbf{-9.83 \pm 1.27}$ | $50.00 \pm 2.22$ | $-4.94 \pm 1.67$ | $24.11 \pm 0.95$ | $-19.10 \pm 1.97$ | $10.51 \pm 0.65$ | $\mathbf{-3.47 \pm 1.21}$ |
| | FLwF-2T | $61.33 \pm 1.15$ | $-24.10 \pm 6.31$ | $52.43 \pm 0.79$ | $-3.35 \pm 2.76$ | $27.47 \pm 0.79$ | $\mathbf{-9.28 \pm 1.62}$ | $20.00 \pm 0.78$ | $-14.51 \pm 1.65$ |
| | SPECIAL-C | $56.25 \pm 0.55$ | $-22.85 \pm 1.86$ | $44.14 \pm 2.59$ | $-16.00 \pm 2.38$ | $42.50 \pm 2.02$ | $-21.14 \pm 1.42$ | $20.14 \pm 0.93$ | $-11.71 \pm 0.22$ |
| | **SPECIAL** (ours) | $\mathbf{64.37 \pm 1.21}$ | $-24.14 \pm 2.36$ | $\mathbf{55.39 \pm 2.31}$ | $\mathbf{-2.22 \pm 3.49}$ | $\mathbf{44.63 \pm 0.81}$ | $-18.65 \pm 2.39$ | $\mathbf{21.70 \pm 0.74}$ | $-8.34 \pm 1.61$ |

*Table 27.* Task specific performance curves on Digit-10 with full participation ($\alpha = 0.1$).

| Method | USPS | SVHN | EMNIST | MNIST | ACC |
|---|---|---|---|---|---|
| FedAvg | 0.9113 | 0.2635 | 0.2928 | 0.9794 | 0.6118 |
| FedProx | 0.6800 | 0.1673 | 0.3891 | 0.9727 | 0.5523 |
| FedCIL | 0.6265 | 0.1876 | 0.2508 | 0.9049 | 0.4924 |
| MFCL | 0.7777 | 0.4016 | 0.3603 | 0.8671 | 0.6017 |
| SR-FDIL | **0.9189** | 0.2818 | 0.3607 | **0.9812** | 0.6357 |
| pFedDIL | 0.7535 | 0.1365 | 0.1300 | 0.7127 | 0.4332 |
| FLwF-2T | 0.7845 | **0.5586** | 0.3335 | 0.7765 | 0.6133 |
| SPECIAL-C | 0.8527 | 0.2006 | 0.2919 | 0.9050 | 0.5625 |
| **SPECIAL (ours)** | 0.8622 | 0.2171 | **0.5411** | 0.9544 | **0.6437** |

*Table 28.* Task specific performance curves on VLCS with full participation ($\alpha = 0.3$).

| Method | Caltech101 | LabelMe | SUN09 | VOC2007 | ACC |
|---|---|---|---|---|---|
| FedAvg | 0.4853 | 0.3917 | 0.5037 | 0.6956 | 0.5191 |
| FedProx | 0.4330 | 0.3949 | 0.4751 | **0.8313** | 0.5336 |
| FedCIL | 0.5347 | 0.4796 | 0.4171 | 0.4208 | 0.4631 |
| MFCL | 0.5536 | 0.4801 | 0.3841 | 0.4742 | 0.4730 |
| SR-FDIL | 0.4483 | 0.4731 | 0.4191 | 0.4509 | 0.4478 |
| pFedDIL | **0.7274** | 0.4932 | 0.3972 | 0.3824 | 0.5000 |
| FLwF-2T | 0.7483 | 0.5073 | 0.4159 | 0.4256 | 0.5243 |
| SPECIAL-C | 0.4701 | 0.4520 | 0.4057 | 0.4378 | 0.4414 |
| **SPECIAL (ours)** | 0.6990 | **0.5178** | **0.5362** | 0.4628 | **0.5539** |

*Table 29.* Task specific performance curves on PACS with full participation ($\alpha = 0.3$).

| Method | Sketch | Cartoon | Photo | Art Painting | ACC |
|---|---|---|---|---|---|
| FedAvg | 0.3361 | 0.3255 | 0.5074 | 0.5147 | 0.4209 |
| FedProx | 0.2838 | 0.3591 | 0.5238 | **0.5495** | 0.4291 |
| FedCIL | 0.2277 | 0.3487 | 0.4791 | 0.3586 | 0.3535 |
| MFCL | 0.1959 | 0.3008 | 0.4737 | 0.4616 | 0.3580 |
| SR-FDIL | 0.3102 | 0.3541 | **0.5597** | 0.4919 | 0.4290 |
| pFedDIL | 0.2527 | 0.2165 | 0.2604 | 0.2346 | 0.2410 |
| FLwF-2T | 0.2544 | 0.2107 | 0.3738 | 0.2599 | 0.2747 |
| SPECIAL-C | 0.3270 | **0.3670** | 0.5508 | 0.4953 | 0.4350 |
| **SPECIAL (ours)** | **0.3770** | 0.3554 | 0.5452 | 0.5077 | **0.4463** |

*Table 30.* Task specific performance curves on DN4IL with full participation ($\alpha = 5$).

| Method | Sketch | Infograph | Painting | Quickdraw | Real | Clipart | ACC |
|---|---|---|---|---|---|---|---|
| FedAvg | 0.1936 | 0.0655 | 0.1569 | 0.1412 | 0.2776 | 0.4270 | 0.2103 |
| FedProx | 0.1878 | 0.0621 | 0.1477 | 0.1451 | 0.2721 | 0.4042 | 0.2032 |
| FedCIL | 0.0872 | 0.0488 | 0.0928 | 0.1123 | 0.1834 | 0.3352 | 0.1433 |
| MFCL | 0.1679 | 0.0672 | 0.1581 | 0.0943 | 0.2828 | 0.3823 | 0.1921 |
| SR-FDIL | 0.2071 | 0.0614 | 0.1575 | 0.1486 | 0.2810 | **0.4317** | 0.2146 |
| pFedDIL | 0.1955 | 0.0406 | 0.0772 | 0.0580 | 0.1215 | 0.1376 | 0.1051 |
| FLwF-2T | 0.1858 | 0.0272 | 0.1209 | **0.1834** | 0.2690 | 0.4139 | 0.2000 |
| SPECIAL-C | **0.2325** | 0.0700 | 0.1471 | 0.1146 | 0.2557 | 0.3886 | 0.2014 |
| **SPECIAL (ours)** | 0.2071 | **0.0706** | **0.1688** | 0.1460 | **0.2997** | 0.4098 | **0.2170** |

