# OpenReview forum: "Server-Proximal Aggregation for Federated Domain-Incremental Learning under Partial Participation: Task-Uniform Convergence and Backward Transfer"
_ICML.cc/2026/Conference — ICML 2026 regular_

### Official Review · Reviewer_4h4p · 2026-03-04

**Soundness:** 2
**Presentation:** 3
**Significance:** 3
**Originality:** 2
**Overall Recommendation:** 3
**Confidence:** 3

**Summary:**

This paper studies **federated domain-incremental learning (FDIL)** under **partial participation**, where tasks arrive sequentially with changing domains but a fixed label space. The authors propose **SPECIAL**, a simple server-side proximal anchoring strategy: after standard aggregation, the server interpolates the current aggregated model with the previous task’s global model using a single hyperparameter $\lambda$, aiming to balance stability (retain past knowledge) and plasticity (learn new tasks). The paper provides convergence/backward-transfer style analysis and evaluates the method on several vision FDIL benchmarks.

**Compliance With Llm Reviewing Policy:**

Affirmed.

**Final Justification:**

#### 1. Regarding Equation (6) and Convergence Analysis
I appreciate the authors' response. I agree that **Eq. (6)** is supportive of **Federated Domain-Incremental Learning**. Its simplified form is indeed effective for facilitating more rigorous reasoning in **convergence analysis**, which is a clear theoretical advantage.

#### 2. Concerns on Methodological Superiority
However, I remain skeptical about the **methodological superiority** of this approach over other established solutions.
* The proposed **EMA-based scheme** appears functionally similar to existing strategies that retain a final model from previous tasks (e.g., weight consolidation or knowledge distillation variants).
* It remains unclear why this specific mechanism should inherently outperform these baseline methods, and my concerns regarding its comparative effectiveness persist.

#### 3. Measurement of Task/Domain Shift
Regarding my question on defining a measurable notion of task/domain change and its relation to the optimal $\alpha$:
> *“Since the theoretical contribution involves handling task/domain shifts, can you define (or adopt) a measurable notion of task/domain change and relate it to the optimal $\alpha$?”*

The authors' response provided an **intuitive explanation**, which unfortunately did not fully meet my expectations. I was looking for a more formal definition or a clear **empirical correlation** (e.g., how the magnitude of the shift dictates the value of $\alpha$) to provide stronger practical insights.

---

#### Final Decision
**That said**, considering the overall **workload** and the technical efforts presented in the current manuscript, I recognize the contribution of this work. I will **appropriately increase my score** to reflect this.

**Key Questions For Authors:**

- **On novelty / positioning:**
  The server-side proximal step has a closed-form solution and can be rewritten as an interpolation $\theta \leftarrow \alpha\,\theta_{i-1} + (1-\alpha)\,\bar{\theta}$ (for a re-parameterized $\alpha$). Could you clarify what the key novelty is beyond a standard smoothing/anchoring operation, and provide stronger positioning relative to common EMA-style stabilization (including asynchronous FL-style anchoring)?

- **On choosing $\lambda$ (or $\alpha$):**
  How is $\lambda$ selected in practice—fixed across all tasks, or task-dependent? Can you provide a clear guideline or default range that works robustly across datasets and task sequences, without extensive tuning?

- **Sensitivity analysis for $\lambda$:**
  Could you include a systematic sweep showing how ACC/BWT (and convergence speed) vary with $\lambda$? This would strengthen reproducibility and clarify how sensitive SPECIAL is to the stability--plasticity trade-off.

- **Task-shift-aware tuning:**
  Since the theoretical contribution involves handling task/domain shifts, can you define (or adopt) a measurable notion of task/domain change and relate it to the optimal $\lambda$ (e.g., larger shift $\Rightarrow$ smaller/larger $\lambda$)? Even a simple empirical correlation would improve practical insight.

- **Non-IID client heterogeneity:**
  FDIL already involves domain shift; in realistic FL, client data are also non-IID (label/feature skew). How does SPECIAL perform under explicit non-IID heterogeneity, and does $\lambda$ require substantially different tuning in that setting?

- **Beyond vision:**
  Do you expect similar behavior on non-vision tasks (e.g., text)? If not evaluated, could you comment on potential challenges or necessary adaptations?

**Limitations:**

yes

**Strengths And Weaknesses:**

#### Strengths
- **Very simple and engineering-friendly.** The method adds only a closed-form server-side anchoring step on top of FedAvg, with minimal changes to client training and communication.
- **Good design comparison:** The distinction between **server-side** vs **client-side** proximal regularization is clearly discussed and empirically motivated.
- **Relatively complete experimental pipeline (within vision FDIL).** The paper reports performance and efficiency metrics under partial participation, and includes theoretical analysis.

#### Weaknesses / Main Concerns
1) **Originality may be limited (EMA-style interpolation).**
   The server-side proximal update admits a closed-form solution and can be equivalently rewritten as an interpolation between the aggregated model and the previous task anchor, i.e., $\theta \leftarrow \alpha\,\theta_{i-1} + (1-\alpha)\,\bar{\theta}$ for some $\alpha\in(0,1)$ (a re-parameterization of the original proximal coefficient). This resembles a widely used smoothing/anchoring pattern (e.g., EMA-style stabilization commonly seen in optimization and asynchronous FL variants). The main novelty appears to be applying this idea to FDIL and analyzing backward transfer; the paper should better justify why this particular anchoring is uniquely suited to FDIL beyond being a generic smoothing step.

2) **Theory is largely basic; linking task shift to an optimal $\lambda$ would strengthen it.**
   While the convergence/backward-transfer analysis is useful, it follows standard non-convex FL analysis with extensions to incorporate task/domain shifts. A stronger contribution would be to quantify the **magnitude of task/domain change** and connect it to guidance for choosing $\lambda$ (e.g., how $\lambda$ should scale with task shift).

3) **No clear sensitivity analysis for the key hyperparameter $\lambda$.**
   $\lambda$ is central to the stability–plasticity trade-off, yet the paper does not provide a systematic sweep/sensitivity study for $\lambda$ or robust default recommendations across datasets/tasks. This weakens practicality and reproducibility.

4) **Evaluation limitations: vision-only and missing explicit non-IID heterogeneity study.**
   Experiments are restricted to CV datasets/backbones. Adding at least one non-vision task (e.g., text) would strengthen generality. In addition, the evaluation does not explicitly study **non-IID client heterogeneity**, which is a core FL challenge and may interact with the choice of $\lambda$.

---

> ### Author Rebuttal · Authors · 2026-03-30
>
> Thank you very much for the careful reading and detailed comments. We address the main concerns below.
> >**W1/Q1: Originality versus EMA-style interpolation**
>
> We agree that Eq.(6) can be *algebraically* rewritten in an EMA-like interpolation form, but the similarity is only formal. Eq.(6) is the minimizer of the proximal objective in Eq.(5), not a heuristic smoothing step. The different from generic EMA is that SPECIAL anchors each round of task $i$ to the *previous-task model* $\theta_{i-1}$, which contains past knowledge. This is suited to FDIL, where the key challenge is preserving earlier tasks under domain shift without access to past data.
>
> The closed form in Eq.(6) induces the recursion in Eq.(26), which yields a geometrically decaying dependence of $\theta_i^t-\theta_i^0$ on previous rounds. This structure enables Lemma 4.9 and our two theorems, but a generic smoothing step does not provide this drift control. Thus, the novelty is the *server-side proximal design*, and the *task-uniform theoretical* guarantees it hold for FDIL.
>
> >**W2/Q2/Q4: Novelty of the theory and interpretation of $\lambda$**
>
> We agree that our proofs start from standard smooth non-convex FL tools, but we respectfully disagree that the theory is merely routine. The key difficulty is that FDIL is *not* a single stationary optimization problem: the target is the cumulative objective $f_{1:K}$, under client heterogeneity and inter-task domain shift. Standard FL analyses typically treat a stationary objective, but our paper proves *task-uniform convergence* over the task sequence and a *backward-transfer guarantee*. For BKT, Remark 4.8 explains how the performance on earlier-tasks can be improved by learning new knowledge. For task-uniform convergence, Remark 4.11 show that the recursion (Eq.(26)/(27)) induces the closed form to yield the round-uniform drift bound in Lemma 4.9, which is the key to prove cumulative objective convergence.
>
> The $\lambda$ is fixed across all tasks in our setting. We agree that relating task shift to $\lambda$ more explicitly would improve practical insight, and the theory already partially does so: Theorem 4.10 contains the inter-task drift quantity $\sigma_T^2$, while Lemma 4.9 shows that larger $\lambda$ tightens the within-task drift bound. As a guideline, these results suggest the qualitative rule that larger task shift favors a larger $\lambda$, while milder shift allows a smaller $\lambda$ to preserve plasticity. We do not currently give a closed-form optimal $\lambda$ as it would require balancing unknown constants simultaneously.
>
>
> >**W3/Q3: Sensitivity analysis**
>
> We agree that $\lambda$ is central to the trade-off in SPECIAL, and that its sensitivity and practical selection should be made more explicit. We would like to clarify that the paper already includes a systematic $\lambda$ sweep in **Fig.3**. This sweep supports three observations: (i) BWT increases with $\lambda$, since larger $\lambda$ strengthens the anchor; (ii) ACC is unimodal, reflecting the expected trade-off; and (iii) best joint performance occurs at mid-range $\lambda$, indicating that performance does not rely on a narrow optimum. We also report results of rounds-to-best with vary values of $\lambda$ below, showing that the efficiency of SPECIAL is robust to $\lambda$. Overall, our empirical results indicate that SPECIAL is not brittle to $\lambda$: performance changes smoothly rather than collapsing outside a narrow optimum.
> ||0|0.2|0.4|0.6|0.8|1.0|
> |-|-|-|-|-|-|-|
> |Digit-10|83|63|56|67|49|47|
> |VLCS|77|55|51|54|52|47|
> |PACS|84|70|77|66|61|86|
> |DN4IL|171|145|164|162|154|153|
>
>
>
> >**W4/Q5/Q6: Beyond vision and non-IID heterogeneity**
>
> - **Beyond Vision**. The design and theory of SPECIAL are not tied to a specific modality, and we ran an additional *text* experiment using a GLUE task sequence ([SST-2, QQP, QNLI]) with a DistilBERT-base backbone. Under the same partial-participation setting, SPECIAL achieves the best ACC and the strongest BWT among baselines.
>     ||ACC|BWT|
>     |-|-|-|
>     |FedAvg|65.19|-7.00|
>     |FedProx|58.91|-16.80|
>     |pFedDIL|66.13|-6.80|
>     |FLwF-2T|69.15|-5.50|
>     |SPECIAL-C|61.37|-6.64|
>     |SPECIAL|**74.70**|**-4.03**|
> - **Explicit non-IID heterogeneity.** We also note that the main paper already includes an explicit study of non-IID client heterogeneity via vary $\alpha$ (Fig.4). Smaller $\alpha$ values induce stronger client-level skew, and we observe that stronger heterogeneity degrades ACC and affects BWT in a manner consistent with the variance terms in Theorem 4.10. Thus, non-IID heterogeneity is already part of both our setup and our empirical analysis. In addition, we add a FCIL experiment based on Split-CIFAR100 to show the efficiency under class-incremental setting, and the detailed results can be referred at W3 of Reviewer Ewok. In the revision we will add a discussion about the performance of SPECIAL under stronger heterogeneity setting.

---

> > ### Author Rebuttal · Reviewer_4h4p · 2026-04-02
> >
> > ### Reviewer Feedback on Rebuttal
> >
> > #### 1. Regarding Equation (6) and Convergence Analysis
> > I appreciate the authors' response. I agree that **Eq. (6)** is supportive of **Federated Domain-Incremental Learning**. Its simplified form is indeed effective for facilitating more rigorous reasoning in **convergence analysis**, which is a clear theoretical advantage.
> >
> > #### 2. Concerns on Methodological Superiority
> > However, I remain skeptical about the **methodological superiority** of this approach over other established solutions.
> > * The proposed **EMA-based scheme** appears functionally similar to existing strategies that retain a final model from previous tasks (e.g., weight consolidation or knowledge distillation variants).
> > * It remains unclear why this specific mechanism should inherently outperform these baseline methods, and my concerns regarding its comparative effectiveness persist.
> >
> > #### 3. Measurement of Task/Domain Shift
> > Regarding my question on defining a measurable notion of task/domain change and its relation to the optimal $\alpha$:
> > > *“Since the theoretical contribution involves handling task/domain shifts, can you define (or adopt) a measurable notion of task/domain change and relate it to the optimal $\alpha$?”*
> >
> > The authors' response provided an **intuitive explanation**, which unfortunately did not fully meet my expectations. I was looking for a more formal definition or a clear **empirical correlation** (e.g., how the magnitude of the shift dictates the value of $\alpha$) to provide stronger practical insights.
> >
> > ---
> >
> > #### Final Decision
> > **That said**, considering the overall **workload** and the technical efforts presented in the current manuscript, I recognize the contribution of this work. I will **appropriately increase my score** to reflect this.
> >
> >
> > Update:
> >
> > According to my understanding, your approach is not strictly 'memory-free' in a simultaneous sense. At a minimum, each client is required to maintain a  $\theta_{i-1}$, which constitutes a persistent memory overhead.
> >
> >
> > Update:
> > If  $\theta_{i-1}$ is stored in the server, it still constitutes a persistent memory overhead.

---

> > > ### Author Response · Authors · 2026-04-03
> > >
> > > Thank you for the thoughtful follow-up and for recognizing the theoretical advantage of Eq.(6) in enabling rigorous convergence analysis for FDIL. We are also grateful that you plan to increase your score.
> > >
> > > > **(1) On methodological superiority.**
> > >
> > > We would like to clarify our claim more precisely. We do **not** claim server-side anchor must universally outperform every retention mechanism such as consolidation- or distillation-based methods in every regime. Rather, our claim is it occupies a particularly favorable point in FDIL design space: it is *memory-free*, requires no replay buffer, no synthetic data, no auxiliary teacher/distillation targets, no model expansion, and no extra communication, while still admitting strong theory.
> > >
> > > Specifically, to our best knowledge, this paper provides the **first task-uniform convergence guarantee for FDIL under partial participation**, together with a **guaranteed BKT result**. This is the sense in which we view the mechanism as methodologically distinctive: not as an unconditional claim of empirical superiority over all retention strategies, but as a particularly simple, deployable, and analyzable design point that prior FDIL methods do not provide. **Among methods that are simultaneously memory-free, communication-neutral, and theoretically tractable under partial participation, we believe this is a meaningful advantage.**
> > >
> > > Empirically, SPECIAL is competitive with both memory-based and memory-free baselines. Theoretically, the key advantage is that server-side proximal recursion yields round-uniform drift bound in Lemma 4.9, which is precisely what makes cumulative-objective analysis possible. We will state this claim more carefully as a *theory/efficiency/implementability* advantage rather than as an unconditional superiority claim over all alternative retention strategies.
> > >
> > > > **(2) On measuring task/domain shift...relation to $\lambda$.**
> > >
> > > We agree our previous response was more qualitative than formal. In current paper, the formal quantity that captures task/domain change is the inter-task drift term $\sigma_T^2$ in Theorem 4.10. This is the object through which task shift enters our convergence result, and together with Lemma 4.9 it yields qualitative rule that larger inter-task drift should favor stronger anchoring (larger $\lambda$). In this sense, **current paper already takes a formal step toward a shift-aware interpretation of $\lambda$ by making the task-shift term explicit in the bound.**
> > >
> > > We agree, however, the paper does not yet instantiate a direct empirical estimator that maps observed shift magnitude to a recommended $\lambda$. We will revise the discussion to make clear $\sigma_T^2$ is the formal shift proxy in theory, and that a more explicit empirical calibration rule (for example, based on gradient discrepancy or distributional distance between adjacent tasks) is an important next step beyond current manuscript.
> > >
> > > A more explicit adaptive rule for selecting $\lambda$ is beyond current manuscript. More broadly, to our best knowledge, **a fully principled or adaptive rule for selecting this type of stability-plasticity coefficient remains unresolved not only for FDIL, but also more generally in CL and FCL.**
> > >
> > > We appreciate this clarification request. It helps us sharpen the positioning: the present contribution is a minimal server-side mechanism with strong analyzability and competitive empirical performance in FDIL, while a more explicit shift-to-$\lambda$ calibration rule remains an important direction for follow-up work.
> > >
> > > > **(3) The term "memory-free".**
> > >
> > > In CL/FCL, when we use memory-free, we mean the method does not rely on replay memory: it stores **no past raw samples, no exemplar buffer, no synthetic replay generator, and no task-specific memory** that grows with the number of tasks [1,2]. This is the sense in which we described SPECIAL as memory-free.
> > >
> > > More precisely, SPECIAL **does not require each client to maintain $\theta_{i-1}$**. Previous-task checkpoint $\theta_{i-1}$ is only used in server-side proximal aggregation step, while client-side training remains **the same as plain FedAvg**. Thus, the additional state is a single server-side model snapshot of constant size, rather than a replay buffer, generative memory, or per-task state that scales with task sequence.
> > >
> > > [1] Towards Robust Continual Learning With Bayesian Adaptive Moment Regularization.
> > >
> > > [2] No Forgetting Learning: Memory-free Continual Learning.
> > >
> > > >**Update:** Thanks for your updated comment. Again, our intended use of memory-free was **standard continual-learning sense**: no replay buffer, no exemplar storage, no synthetic replay generator, and no task-specific memory that grows with the number of tasks. To avoid ambiguity, we are happy to revise it to **replay-free**. The substantive point is SPECIAL adds only a single constant-size server-side checkpoint $O(|\theta|)$, while imposing no client-side replay memory, no per-task state growth, and no extra communication.

---

### Official Review · Reviewer_5ht3 · 2026-03-13

**Soundness:** 2
**Presentation:** 2
**Significance:** 2
**Originality:** 2
**Overall Recommendation:** 3
**Confidence:** 4

**Summary:**

This paper studies federated domain- incremental learning under partial participation and proposes SPECIAL, a lightweight server-side proximal aggregation method built on top of FedAvg. The problem setting is relevant, the paper is generally well written, and the method is simple

**Compliance With Llm Reviewing Policy:**

Affirmed.

**Final Justification:**

The authors have addressed my concerns partially but still, I remain skeptical with regards to the generalizability of the solution. For this I will maintain my score.

**Key Questions For Authors:**

Can the authors provide clearer numerical comparisons of memory overhead, peak memory usage, and extra computation cost?

Can the authors provide more detailed result and explaination on choosing \lambda, do we need to search for the optimal for each case, which sould less practical.Also clarify how robust the method is when \lambda is outside the selected range?

Can the authors report more complete results under different Dirichlet parameters, and clarify why the Dirichlet parameter was searched instead of being treated as a fixed heterogeneity setting(like typical 0.1 0.3 0.5)?

Can the authors clarify more explicitly how SPECIAL differs from FedProx-style proximal methods and other recent stability-oriented baselines(no experiment need)?

**Limitations:**

See above

**Strengths And Weaknesses:**

Strengths

The paper studies a meaningful and practical setting, namely federated domain-incremental learning under partial participation, which is relevant to realistic federated systems.

The proposed method is simple, lightweight, and easy to implement. It does not require replay buffers, synthetic data, model expansion, or extra communication.

The empirical results are generally encouraging, and the paper also attempts to support the method with theory tailored to this FDIL setting.

Weaknesses

The method's baseline is a bit old. In additional, SPECIAL may appear conceptually related to FedProx-style proximal training, while works such as MimiC(https://arxiv.org/abs/2306.12212) also study degradation under client partial participants could be discussed.

The empirical study remains limited in both experimental scope and hyperparameter transparency. Since the evaluation is mainly restricted to vision benchmarks with a ResNet backbone, it is difficult to assess generalization beyond this setup. Moreover, the large variation of

λ across datasets raises practical concerns about how robust the method is and how it should be tuned in real applications.

The efficiency claim is plausible but not yet quantitatively. As stated, SPECIAL reuses the previous global model/vector and does not introduce extra communication, the paper should provide direct numerical evidence on memory overhead, peak memory budget, and extra computation cost, instead of relying mainly on high-level efficiency arguments.

The discussion of the proximal coefficient

λ is not sufficiently detailed. The selected

λ values vary considerably across datasets, which raises practical concerns about whether the method only works well within a narrow range, and how performance degrades when

λ is not chosen properly.

The treatment of the Dirichlet parameter in the non-IID setting is not fully satisfactory. The paper mentions grid search over the Dirichlet parameter, but does not provide enough detailed results for different values. Since the Dirichlet parameter defines the heterogeneity setting rather than a method hyperparameter, it would be better discussed as an experimental condition.

---

> ### Author Rebuttal · Authors · 2026-03-30
>
> Thank you very much for your comments. We address the main concerns below.
>
> >**W1/Q4: Baseline choice and relation to FedProx/MimiC**
>
> Thank you for this comment. Our baseline set is designed to cover main families in FDIL, including replay/regularization-based and standard FL methods. In the revised version, we will make this categorization more explicit.
>
> Though SPECIAL and FedProx-style methods both use a proximal mechanism, the difference lies in where and to what it is applied. FedProx uses a client-side proximal term during SGD, anchoring each client to the global model to mitigate within-task client drift. In contrast, SPECIAL applies a server-side proximal anchor after aggregation, pulling the model toward the previous-task global model. This design targets across-task drift and enables the preservation of past knowledge.
>
> Empirically, comparisons with SPECIAL-C show that while client-side proximal can improve early-stage stability, SPECIAL achieves a better balance and stronger final performance. Theoretically, the server-side update in SPECIAL induces the recursion used in Lemma 4.9, which yields a geometrically decaying drift bound and underpins two theorems. This recursion does not arise in standard averaging or client-side proximal updates, so the same guarantees do not hold.
>
> We also thank the reviewer for pointing out MimiC. While MimiC studies partial participation and client drift, its objective is different: it focuses on correcting optimization bias in a single-task federated setting (similar to SCAFFOLD) without addressing continual task sequences or knowledge retention. In contrast, SPECIAL is specifically designed for FDIL, where across-task drift is the primary challenge. We will include a discussion about Mimic in the revision.
>
> >**W2: Limited empirical scope and transparency**
>
> This concern overlaps with W4/Q5/Q6 from Reviewer 4h4p, so we refer to our response there. In brief, we add a text experiment using a GLUE task sequence with a DistilBERT backbone, which shows that SPECIAL is not limited to vision models. In addition, we list the detailed configurations in Sec 5 and provide all hyperparameters in Appendix F.2 to improve transparency.
>
> >**W3/Q2: Robustness and tuning of $\lambda$**
>
> We agree that $\lambda$ is crucial in SPECIAL as it controls the stability-plasticity trade-off, and practical robustness is important. This concern overlaps with W3/Q3 from Reviewer 4h4p's, so we refer to our response there. To address your concern about robustness, we additionally evaluated values beyond the range $[0,1]$ used in the main paper. On Digit-10, we tested $\lambda \in \\{2,5,10,100\\}$ as shown below. The results show that even when $\lambda$ is chosen far outside the range, SPECIAL does not fail catastrophically: ACC decreases gradually while BWT keeps improving, which is the expected stability-plasticity trade-off. Thus, although the best $\lambda$ is dataset-dependent, the method does not only work in a narrow range, and a coarse sweep or a moderate default is often sufficient.
>
> ||2|5|10|100|
> |-|-|-|-|-|
> |ACC|48.23|45.04|42.90|40.40|
> |BWT|-3.73|-0.06|0.36|0.22|
>
> >**W4/Q1: Numerical efficiency comparisons**
>
> Thank you for this comment. We provide a numerical efficiency comparisons below based on Digit-10 dataset. **Memory overhead (Mem)** is measured by the total size of parameters involved in each client, including the trainable parameters and any dataset buffer. **Peak memory usage (Peak)** is measured by the maximum GPU memory consumption observed during training. We measure **computation cost** by the average training time per round. The result shows that the lightweight design of SPECIAL enables to achieve the training efficiency comparable to the standard FedAvg.
>
> ||Mem(MB)|Peak(GB)|Time(s)|
> |-|-|-|-|
> |FedAvg|42.69|1.54|191.27|
> |FedProx|42.69|1.73|510.82|
> |FedCIL|59.05|1.93|523.16|
> |MFCL|117.8|3.28|969.01|
> |SR-FDIL|55.82|2.29|357.34|
> |pFedDIL|130.15|3.65|699.43|
> |FLwF-2T|85.38|1.59|247.90|
> |SPECIAL-C|85.38|1.58|518.26|
> |SPECIAL|42.69|1.56|200.66|
>
> >**W5/Q3: Dirichlet parameter as an experimental condition**
>
> Thank you for this comment. We agree that the role of the Dirichlet parameter $\alpha$ should be stated more clearly. We do *not* grid-search $\alpha$ to optimize the model, since $\alpha$ is part of the *configuration*. Rather, We fix $\alpha$ on each dataset for all methods to keep fair comparisons.
>
> The wider grid $\alpha \in \\{0.05,\ldots,1,100\\}$ appears only in the ablation study, where we vary heterogeneity from highly non-IID to IID to understand its effect. We agree that this distinction could be more explicit. In the revision, we will state clearly that $\alpha$ is an fixed experimental parameter rather than a tuned hyperparameter, report the exact fixed $\alpha$ used for each benchmark, and expand the ablation discussion to make clear that Fig.4 is precisely a study under different non-IID conditions.

---

> > ### Author Rebuttal · Reviewer_5ht3 · 2026-04-01
> >
> > After looking at the rebuttal, I lean towards maintaining my score. The reason is that the sensitivity analysis of  $\lambda$ across datasets and settings remain unclear to me. It is also apparent that the performance of the method is heavily dependent on a  well-tuned  $\lambda$.

---

> > > ### Author Response · Authors · 2026-04-01
> > >
> > > ### Follow-up on $\lambda$ sensitivity across datasets/settings and the concern about heavy dependence on tuning. ###
> > >
> > > Thank you for this clarification. We understand that your remaining concern has two parts: (i) whether the sensitivity of $\lambda$ across datasets/settings is clear enough, and (ii) whether the method is heavily dependent on a finely tuned $\lambda$ in practice.
> > >
> > > >**(1) Sensitivity across datasets/settings.**
> > >
> > > Fig.3 already suggested that the ACC-BWT trade-off evolves smoothly with $\lambda$, but we agree that our previous rebuttal did not make this cross-dataset point explicit enough. Using the ACC values from our $\lambda$ sweep, we can now state the practical picture more clearly. On Digit-10, the best ACC is attained at $\lambda=0.25$, but $\lambda=0.2$ is only 0.06\% worse and $\lambda=0.3$ is only 0.60\% worse. On VLCS, the best ACC is at $\lambda=0.4$, while $\lambda=0.45$ and $\lambda=0.6$ are only 1.40\% and 1.79\% worse, respectively. On DN4IL, the best ACC is at $\lambda=0.1$, while $\lambda=0.15$ and $\lambda=0.05$ are only 0.82\% and 1.11\% worse. These results show that, for three of the four datasets, strong performance is attained over a *local band* of mid-range $\lambda$ values rather than at a knife-edge optimum.
> > >
> > > The main exception is PACS, where the best ACC is at $\lambda=0.05$ and nearby values degrade more quickly (e.g., $\lambda=0$ is 5.52\% worse and $\lambda=0.1$ is 7.84\% worse). We therefore do *not* claim that the degree of sensitivity is identical across datasets. Rather, **our point is that the sharpness of the stability-plasticity trade-off depends on the task/domain shift regime:** some datasets admit a broader near-optimal region, while others are sharper.
> > >
> > >
> > > >**(2) Does this mean the method is heavily dependent on a well-tuned $\lambda$?**
> > >
> > > We agree with the limited statement that $\lambda$ matters: SPECIAL is a regularization-based method, so some tuning is naturally required. However, the results do *not* support the stronger claim that the method only works with a single finely tuned value. On three of the four datasets, nearby choices remain within about **0.06\%-1.79\%** of the best ACC, and even on the sharper PACS case the degradation is still smooth rather than catastrophic. This is also consistent with the additional out-of-range Digit-10 experiment we reported earlier.
> > >
> > > So the practical takeaway is not that one can ignore $\lambda$, but that **exact optimal tuning is typically unnecessary**. In practice, a simple two-stage coarse-to-fine search is sufficient: we first evaluate a small coarse grid and then refine locally around the top candidates. This is enough to identify a robust value, with somewhat larger $\lambda$ generally preferred under stronger task/client heterogeneity and smaller $\lambda$ under milder shift.

---

### Official Review · Reviewer_Ewok · 2026-03-13

**Soundness:** 3
**Presentation:** 3
**Significance:** 4
**Originality:** 3
**Overall Recommendation:** 5
**Confidence:** 4

**Summary:**

This paper studies Federated Domain-Incremental Learning (FDIL), where heterogeneous clients receive tasks sequentially with shifting input domains while the label space is fixed. Two theoretical pillars are identified as missing: backward knowledge transfer (BKT) guarantees and convergence rates uniform across the task sequence under partial participation. SPECIAL is introduced: a memory-free algorithm extending FedAvg with a single server-side proximal anchor — blending the aggregated model with the previous global model after each round. Joint theoretical guarantees of BKT and task-uniform convergence are derived. No replay buffers, synthetic data, or task-specific heads are required.

**Compliance With Llm Reviewing Policy:**

Affirmed.

**Key Questions For Authors:**

1. How does SPECIAL compare empirically to GLFC, FOT, and FedWeIT on standard FDIL benchmarks?
2. What is the sensitivity to the proximal weight, and is there a data-driven method for selecting it?
3. Does the BKT guarantee hold under very sparse participation (e.g., 5–10% of clients per round)?
4. How does the method behave when domain shift between tasks is severe versus mild?

**Limitations:**

Not fully discussed; authors should add:
(1) proximal weight hyperparameter — sensitivity and principled selection;
(2) behavior at very low participation rates and whether a minimum participation fraction exists for guarantees to hold;
(3) impact of severe domain shift on smoothness assumptions;
(4) scope: results are for domain-incremental (fixed label space) only — generalizability to class-incremental is not addressed;
(5) privacy implications of the server blending with a previous global model;
(6) conditions under which the proximal anchor may slow convergence (e.g., early tasks with no forgetting risk).

**Strengths And Weaknesses:**

Strengths:
1. Clearly identified theoretical gap: BKT guarantees and task-uniform convergence under partial participation are both missing from prior FDIL work, and both are non-trivial to establish.
2. Algorithm is elegantly minimal: a single server-side proximal step requiring no change to client-side computation, communication protocol, or model architecture.
3. Memory-free design is practically significant in federated settings where client storage is severely constrained and replay buffers are infeasible.
4. Joint theoretical analysis (BKT + task-uniform convergence) is the paper's standout contribution and provides a theoretical foundation that has been absent from the field.
5. Problem formulation is precise and the setting is well-motivated against real federated deployment conditions.
6. Communication overhead is unchanged relative to FedAvg — an important practical property.

Weaknesses:
1. Empirical comparison with established FDIL and federated continual learning baselines (GLFC, FOT, FedWeIT) is absent, making it difficult to situate SPECIAL's practical performance in the literature.
2. Proximal weight hyperparameter sensitivity is not analyzed; practical selection guidance is not provided.
3. Ablations on participation rate are missing despite partial participation being central to the problem statement.
4. Standard federated continual learning benchmarks (Split-CIFAR with domain shifts, PACS) are not used, limiting comparability.
5. The domain-incremental setting (fixed label space) is somewhat restrictive; discussion of scope and generalizability to class-incremental settings would strengthen the contribution.

Soundness:
The theoretical analysis is the paper's greatest strength and appears rigorous. BKT guarantees and task-uniform convergence under partial participation are non-trivial results, and the proximal anchor mechanism is precisely described. Standard smoothness and bounded gradient assumptions are used; the degree to which these hold under severe domain shift should be acknowledged. The empirical evaluation is the weaker component, but the theoretical contribution is sound and independently valuable.

Presentation:
The paper is clearly written and the problem formulation is precise. The theoretical contributions are well-presented and accessible to specialists. The gap identification is particularly well-argued. The experimental section is the weakest in terms of presentation — standard benchmarks and established baselines are absent, which makes the empirical results hard to contextualize. A clearer discussion of how the theoretical assumptions map to realistic federated scenarios would strengthen the paper.

Significance:
Federated continual learning is a rapidly growing and practically important area (healthcare, mobile, edge AI). The combination of a minimal, memory-free algorithm with rigorous theoretical guarantees in an undertheorized setting is genuinely valuable. The theoretical contributions alone are a meaningful advancement. Communication efficiency and client-side simplicity make SPECIAL practically deployable in constrained federated environments.

Originality:
The server-side proximal anchor for continual drift prevention in FDIL is a simple but previously unexplored mechanism. The distinction from FedProx (client-side proximal for heterogeneity) is meaningful. The joint analysis of BKT and task-uniform convergence under partial participation is original and provides a theoretical foundation absent from prior work.

---

> ### Author Rebuttal · Authors · 2026-03-30
>
> Thank you very much for constructive comments and the positive assessment.
>
> >**W1/Q1: New baselines**
>
> Thank you for the comment. We evaluate GLFC, FOT, and FedWeIT under the same settings. The result shows that SPECIAL still maintains a leading position among them.
> ||Digit-10||VLCS||PACS||DN4IL||
> |-|-|-|-|-|-|-|-|-|
> ||ACC|BWT|ACC|BWT|ACC|BWT|ACC|BWT|
> |GLFC|61.48|-30.88|52.38|-2.70|38.68|-18.23|22.52|-15.03|
> |FOT|61.45|-37.73|42.27|-11.88|39.11|-27.33|21.81|**-13.43**|
> |FedWeIT|47.08|**-11.98**|45.85|-12.85|40.23|-17.64|19.59|-18.97|
> |SPECIAL|**62.12**|-21.78|**54.92**|**-0.11**|**45.29**|**-11.66**|**24.30**|-19.50|
>
> >**W2/Q2: $\lambda$ sensitivity and selection**
>
> This concern overlaps with W3/Q3 of Reviewer 4h4p, so we refer to our response there. Briefly, **Fig.3** provides a systematic $\lambda$ sweep, showing that SPECIAL is not brittle to precise tuning: BWT increases with $\lambda$, ACC is unimodal, and strong performance is achieved over a broad mid-range rather than at a narrow isolated optimum.
>
> As for data-driven selection, it is standard to choose $\lambda$ by grid-search in the current FCL work. We agree that an automatic rule would be valuable. Our theory provides guidance: Theorem 4.10 makes task shift explicit through $\sigma_T^2$, while Lemma 4.9 shows that larger $\lambda$ tightens the drift bound. Developing a self-tuning strategy for $\lambda$ is an important direction for future work.
>
> >**W3/Q3: Sparse participation**
>
> We agree that participation-rate ablations are important because partial rate is central to the problem. Theorems covers $1\le N\le M$: the BKT and convergence bounds make the dependence on participation explicit, so smaller $N$ worsens the relevant drift terms, but guarantees do not disappear.
>
> We ran a full study for SPECIAL with $M=8$ and $N\in\\{2,4,6,8\\}$. The results show that ACC improves as more clients participate, since more information is collected. Meanwhile, lower rate does not collapse performance: even with $N=2$, SPECIAL is competitive on all datasets.
>
> Our setup has $M=8$ clients, so the smallest $N$ is $1$ (12.5%). Thus, we do not yet claim direct empirical evidence for 5-10%, but the dependence on $N$ in theorems covers that regime, and the observed trend from $N=2$ to $8$ is consistent with it.
> |N|Digit-10||VLCS||PACS||DN4IL||
> |-|-|-|-|-|-|-|-|-|
> ||ACC|BWT|ACC|BWT|ACC|BWT|ACC|BWT|
> |2|57.57|-16.13|50.18|-0.09|33.50|-12.83|21.20|-10.74|
> |4|62.12|-21.78|54.92|-0.11|45.29|-11.66|24.30|-19.50|
> |6|62.88|-25.15|53.61|0.00|40.92|-12.41|21.28|-10.76|
> |8|64.37|-24.14|55.39|-2.22|44.63|-18.65|21.70|-8.34|
>
> >**W4: Benchmark choice**
>
> We would like to clarify two points.
> - PACS is already one of the four main datasets (Table 1) and also appears in the $\lambda$-sensitivity study (**Fig.3**).
> - our theory is based on FDIL, but Split-CIFAR is a class-incremental benchmark, and the samples share the same domain space, which makes it hard to be splitted in a domain-shift style.
> We report a Split-CIFAR experiment in our **W5** as a class-incremental dataset.
>
> >**W5: Class-Incremental**
>
> We agree that discussing scope beyond FDIL strengthens the paper.
> - SPECIAL can be extended to the class-incremental setting by expanding the classifier head as new classes arrive. The server-side anchor can be applied to the shared backbone and classifier parameters for seen classes, while newly introduced classifier parameters are free to adapt. Thus, the key mechanism is not limited to a fixed-label setting. We ran a FCIL experiment on CIFAR-100 split into $5$ tasks with $20$ classes per task. SPECIAL achieves the best ACC and BWT, suggesting its efficiency in this setting.
>     ||ACC|BWT|
>     |-|-|-|
>     |FedAvg|37.92|-29.64|
>     |FedProx|37.32|-27.51|
>     |FedCIL|31.73|-21.19|
>     |MFCL|28.37|-27.64|
>     |SR-FDIL|35.43|-32.28|
>     |GLFC|32.71|-9.82|
>     |FOT|34.33|-31.80|
>     |FedWeIT|30.08|-13.01|
>     |pFedDIL|31.72|-12.61|
>     |FLwF-2T|38.65|-23.76|
>     |SPECIAL-C|34.60|-18.68|
>     |SPECIAL|**40.68**|**-9.00**|
> - We want to be precise theoretically: current theorems are stated for FDIL, so we do *not* claim that they can cover FCIL, as the FCIL theory would require extending the setup to account for classifier expansion.
>
> >**Q4: Behavior under severe/mild domain shift**
>
> When the shift is *mild*, adaptation to the new task is more likely to preserve earlier tasks, so a smaller $\lambda$ is sufficient. When the shift is *severe*, updating toward the new domain increases forgetting, so a larger $\lambda$ is preferable to provide stronger anchoring to the previous-task model.
>
> This interpretation is consistent with both theory and experiment. Theorem 4.10 makes inter-task drift explicit through $\sigma_T^2$, while Lemma 4.9 shows that larger $\lambda$ tightens the drift bound. Empirically, **Fig.3** shows this pattern: increasing $\lambda$ tends to improve BWT but can eventually reduce ACC if the anchor becomes too strong.

---

> > ### Author Rebuttal · Reviewer_Ewok · 2026-04-02
> >
> > The rebuttal addresses my main empirical concern by adding comparisons to GLFC, FOT, and FedWeIT, and it also clarifies benchmark coverage and provides extra results on participation rate and class-incremental extension. The λ-sensitivity concern is partly addressed through the sweep and discussion, but there is still no principled selection rule beyond grid search. The severe-vs-mild domain shift question is answered mostly qualitatively, and the 5–10% participation regime remains without direct empirical evidence. Overall, the rebuttal is helpful and substantially strengthens the empirical positioning, but a few follow-up questions remain.

---

> > > ### Author Response · Authors · 2026-04-03
> > >
> > > Thank you for the careful follow-up. We are glad the additional comparisons to GLFC/FOT/FedWeIT, the participation-rate study, and the FCIL extension help clarify the empirical positioning.
> > >
> > > On the remaining points:
> > >
> > > >**(1) Principled $\lambda$ selection.**
> > >
> > > We agree that we do not yet provide a closed-form or fully automatic selection rule beyond grid search. Our claim is therefore not that $\lambda$ can be chosen without tuning, but that the method is *not brittle* to exact tuning and that the theory already provides a useful qualitative rule: Theorem 4.10 makes task shift explicit through $\sigma_T^2$, while Lemma 4.9 shows that larger $\lambda$ tightens the drift bound. In practice, this supports a small coarse-to-fine search rather than exhaustive tuning, with somewhat larger $\lambda$ generally preferred under stronger task/client heterogeneity and smaller $\lambda$ under milder shift.
> > >
> > > A more explicit adaptive rule for selecting $\lambda$ is beyond the current manuscript. More broadly, to the best of our knowledge, **adaptive or fully principled selection of such stability-plasticity trade-off coefficients remains unresolved not only for FDIL, but also in continual learning and federated continual learning more generally.** We will make this scope clear in the revision.
> > >
> > >
> > > >**(2) Severe versus mild domain shift.**
> > >
> > > We agree that our previous answer was mainly qualitative. The main point we intend to make is that the theory already identifies the relevant quantity, inter-task drift via $\sigma_T^2$, and that the empirical role of $\lambda$ is consistent with it: larger apparent shift calls for stronger anchoring, while milder shift allows more plasticity. We will make this connection more explicit in the revision.
> > >
> > >
> > > >**(3) 5--10\% participation.**
> > >
> > > Thank you for pointing this out. We agree that our previous rebuttal only addressed this regime theoretically. To respond more directly, we have now run additional experiments in a setup where such sparse participation is realizable. Specifically, we set $M=20$ and choose $N=1$ and $N=2$, corresponding to participation ratios of $5\%$ and $10\%$, respectively. The results are summarized below.
> > >
> > > | N/M | DIGIT10 | | VLCS || PACS | |
> > > |-|-|-|-|-|-|-|
> > > | | ACC      | BWT   | ACC  | BWT   | ACC  | BWT   |
> > > | 5%  | 62.52    | -21.63| 49.24| -12.77| 33.60| -22.27|
> > > | 10% | 64.18    | -23.49| 55.68| -5.65 | 37.44| -18.77|
> > >
> > > These results now provide direct empirical evidence that SPECIAL remains functional even in the very sparse participation regime. As expected, performance is lower than under higher participation, but it does not collapse: the degradation is gradual, which is consistent with the explicit dependence on $N$ in our theory. This strengthens our original claim that sparse participation affects the rate and constants, but does not invalidate the BKT/convergence behavior of the method.
> > >
> > >
> > >
> > > We hope this clarifies that the main empirical positioning is now substantially strengthened, while the remaining open points concern practical refinement rather than the soundness of the core contribution.

---

### Official Review · Reviewer_AxRa · 2026-03-14

**Soundness:** 3
**Presentation:** 3
**Significance:** 3
**Originality:** 3
**Overall Recommendation:** 5
**Confidence:** 3

**Summary:**

This paper investigates federated domain-incremental learning under partial participation. The authors propose SPECIAL, which augments FedAvg with a server-side proximal step to maintain prior task knowledge. Bounds for backwards transfer as well as task-uniform convergence on the joint objective, rather than the final objective, are presented. The performance of SPECIAL is validated on four benchmarks alongside prior FCL methods.

**Compliance With Llm Reviewing Policy:**

Affirmed.

**Final Justification:**

I thank the authors for their rebuttal. The additional ablations addressed my concerns and should be incorporated into the revised main text or appendix. I strongly suggest the authors integrate their explanation of W2 into the final main text. Given the clarifications provided, I will raise my score to Accept.

**Key Questions For Authors:**

1. Does the advantage of SPECIAL over SPECIAL-C hold in data-scarce settings? See W3.
2. Does client-side regularization have similar sensitivity to parameter lambda as server-side regularization? A study similar to that in Figure 3 could be beneficial.
3. The task-uniform convergence bound degrades with the number of tasks K. How does the performance of SPECIAL compare to prior works on longer task sequences?

**Limitations:**

Yes.

**Strengths And Weaknesses:**

**Strengths**

1.  The paper is well-written, and the theoretical contributions are contextualized in the literature.
2.  The proposed method is simple and obtains good performance with reasonable hyperparameter selection.

**Weaknesses**

1. The main drawback of this work is the reliance on the hyperparameter lambda.  The current hyperparameter selection is based on a grid search, which is not practical. The reported results may then be inflated; however, from Figure 3, it appears that SPECIAL is often competitive with sub-optimal lambda.
2. The original motivation for using server-side regularization in Section 2.2 is limited. The setting for Figure 2 is not self-contained and difficult to parse initially. This motivating section would benefit from either a deeper empirical comparison or a theoretical comparison of SPECIAL and SPECIAL-C.
3. The advantage of server-side over client-side regularization may not hold in data-scarce settings, where client-side anchoring could prevent local overfitting. Current experiments seem to have large local data volumes. Evaluation in a low-data setting would help solidify the advantage of SPECIAL over SPECIAL-C.

---

> ### Author Rebuttal · Authors · 2026-03-30
>
> Thank you for the careful reading, constructive comments, and positive overall assessment.
>
> >**W1/Q2: Reliance on $\lambda$ and its sensitivity**
>
> We agree that $\lambda$ is the key stability--plasticity knob in SPECIAL, so robustness to its choice is important. This concern overlaps with Reviewer 4h4p's comments (W3/Q3), so we refer to our response there; here we highlight the main points relevant to your question.
> - **Fig.3** provides a systematic $\lambda$ sweep, and the purpose of including it was to show that SPECIAL is not brittle to precise tuning. The figure shows three consistent patterns: (i) BWT increases with $\lambda$; (ii) ACC is unimodal, reflecting the stability-plasticity trade-off; and (iii) strong performance is achieved over a broad mid-range rather than at a narrow isolated range.
> - client-side and server-side regularization are sensitive to $\lambda$ in qualitatively different ways. In SPECIAL-C, the proximal term is applied at *every local step*, so increasing $\lambda$ suppresses local adaptation more aggressively. In contrast, SPECIAL applies the proximal interaction only once at the server after aggregation, which preserves local plasticity while still controlling inter-task drift. This is reflected in Fig.2 and Table1: SPECIAL-C can improve stability in highly correlated settings such as Digit-10, but when task correlation weakens it slows adaptation and trails SPECIAL in ACC and often also in BWT. We illustrate the $\lambda$-sensitivity of SPECIAL-C on Digit-10 below, and it exhibits the same trend as SPECIAL.
>     |$\lambda$|0|0.2|0.4|0.6|0.8|1.0|
>     |-|-|-|-|-|-|-|
>     | ACC|47.89|50.26|48.73|45.37|42.77|38.01|
>     | BWT|-10.45|-8.81|-6.22|-2.54|-1.34|2.56|
>
> >**W2: Motivation for server-side regularization and comparison to SPECIAL-C**
>
> We agree that Section2.2 can be more self-contained. Fig.2 compares SPECIAL and SPECIAL-C after switching from Task1 to Task2 by evaluating the evolving model on Task1 only (stability), Task2 only (plasticity), and their union (overall performance). Empirically, SPECIAL-C preserves Task1 slightly better in the first few rounds, but SPECIAL learns Task2 faster and achieves better combined performance throughout most of training. This is the trade-off that motivates placing the proximal term at the *server* rather than inside each local update.
>
> The difference is also structural. In SPECIAL, the difference between the server update and the initial point has the recurrence relation as $\theta_i^{t+1} - \theta_i^0=\frac{\gamma_G\Delta_i^t}{1+\lambda} + \frac{\theta_i^{t} - \theta_i^0}{1+\lambda}$, so unrolling the recursion yields geometrically decaying coefficients (Eq.(27)), which enables Lemma 4.9 to bound $\mathbb{E}_t\||\theta_i^t-\theta_i^0\||^2$ over rounds. This bound is then used in both the BKT and task-uniform convergence proofs. In SPECIAL-C, the proximal term is applied inside local training and the server still aggregates in FedAvg style, so the same decaying recursion does not hold. We will clarify this distinction more explicitly and place the empirical motivation closer to Lemma 4.9.
>
> >**W3/Q1: Data-scarce settings and comparison between SPECIAL and SPECIAL-C**
>
> We agree that low-data regimes are an interesting stress test. Our claim is not that SPECIAL universally dominates SPECIAL-C, but that the server-side anchor enables the drift bound and the resulting guarantees, while achieving a stronger overall stability-plasticity balance.
>
> We add a small low-data Digit-10 ablation by subsampling each client's local data with ratio of 0.1, 0.3, and 0.5. The result shows that the scarcity of local data affects both SPECIAL and SPECIAL-C, and the performance degrading as the ratio decreases, but SPECIAL still maintains a better performance than SPECIAL-C.
> |Method|ratio=0.1||ratio=0.3||ratio=0.5||ratio=1.0||
> |-|-|-|-|-|-|-|-|-|
> ||ACC|BWT|ACC| BWT|ACC|BWT|ACC|BWT|
> |SPECIAL-C|39.94|-12.94 |40.09|-6.29 |42.33|-15.29|50.61|-16.42|
> | SPECIAL|47.55|-15.29|50.75|-15.67|61.29| -18.80|62.12| -21.78|
>
> >**Q3: Dependence on the task length $K$**
>
> We agree that the dependence on $K$ is central to the practical meaning of our task-uniform guarantee. Theorem 4.10 shows that the residual term $\Psi$ grows with $K$ when $\gamma_L,\gamma_G$ are fixed. This is expected in FDIL: as more tasks are added, the cumulative objective $f_{1:K}=\sum_{i=1}^K f_i$ becomes harder to optimize because inter-task drift and accumulated variance both increase. The same theorem also suggests how to offset this degradation, namely $\gamma_L=\mathcal{O}(\frac{1}{\sqrt{K}E}),\gamma_G=\mathcal{O}(\frac{1}{K-1})$, which yields Corollary 4.11. Empirically, DN4IL is our longest sequence, and SPECIAL still achieves the best ACC with a larger margin than on the shorter task sequences. While one benchmark is not a complete scaling study over $K$, it provides initial evidence that controlling inter-task drift remains beneficial as the sequence grows.

---

> > ### Author Rebuttal · Reviewer_AxRa · 2026-04-03
> >
> > I thank the authors for their rebuttal. The additional ablations addressed my concerns and should be incorporated into the revised main text or appendix. I strongly suggest the authors integrate their explanation of W2 into the final main text. Given the clarifications provided, I will raise my score to Accept.

---

> > > ### Author Response · Authors · 2026-04-04
> > >
> > > Thank you very much for raising your score to Accept. We sincerely appreciate your constructive feedback throughout the discussion.
> > >
> > > We are glad that the additional ablations addressed your concerns. We also appreciate your suggestion regarding W2, and we will integrate this clarification into the revised main text rather than leaving it only in the rebuttal. We will also incorporate the new ablations into the revised paper and appendix in a more explicit and self-contained way.
> > >
> > > Thank you again for your helpful comments and support.

---

### Decision · Program_Chairs · 2026-04-30

**Decision:**

Accept (regular)

**Comment:**

This paper has two Accept and two Weak Reject after the effective rebuttal. The Reviewer 5ht3 concerns the method's generalizability regarding the tuned lambda. This point is also raised by all other reviewers. The authors provide additional experiments and detailed reply to selection of the parameter. Reviewers AxRa and Ewok generally accept the explanations, but Reviewer 5ht3 remains this concern which leads to a final recommendation of weak reject. The other negative Reviewer 4h4p challenges about the methodological superiority, but still acknowledge the amount of workload and technical contributions presented in this current work. With these information, this paper is qualified for acceptance to ICML. In the final version, the authors should ensure to include the rebuttal's explanation on selection of lambda, and clearly state the work's limitation regarding hyperparameter dependence.